# Statistical Inference Leveraging Synthetic Data with Distribution-Free Guarantees

## Abstract

The rapid proliferation of high-quality synthetic data—generated by advanced AI models or collected as auxiliary data from related tasks—presents both opportunities and challenges for statistical inference. This paper introduces a **GE**neral **S**ynthetic-**P**owered **I**nference (GESPI) framework that wraps around any statistical inference procedure to safely enhance sample efficiency by combining synthetic and real data. Our framework leverages high-quality synthetic data to boost statistical power, yet adaptively defaults to the standard inference method using only real data when synthetic data is of low quality. The error of our method remains below a user-specified bound without any distributional assumptions on the synthetic data, and decreases as the quality of the synthetic data improves. This flexibility enables seamless integration with conformal prediction, risk control, hypothesis testing, and multiple testing procedures, all without modifying the base inference method. We demonstrate the benefits of our method on challenging tasks with limited labeled data, including AlphaFold protein structure prediction, and comparing large reasoning models on complex math problems.[1]

## 1 Introduction

Statistical inference lies at the core of data-driven decision-making, enabling researchers and practitioners to draw conclusions while rigorously controlling error rates. The importance of such control cannot be overstated: uncontrolled errors can lead to misleading conclusions and costly mistakes. For example, in computational biology, researchers increasingly rely on AlphaFold's protein structure predictions, where substantial local errors can mislead downstream applications such as drug discovery or protein design. A/B testing can be used to assess whether a new large language model (LLM) outperforms the current one—yet falsely concluding that the new model is better can lead to revenue loss or customer dissatisfaction.

A fundamental limitation of statistical inference arises when data are scarce, leading to reduced statistical power and high variability in error rates. Yet limited data is almost unavoidable in domains where data acquisition is costly, difficult, or time-consuming. For example, in protein structure prediction, experimental validation is expensive and labor-intensive, resulting in relatively few labeled structures. Similarly, comparing LLMs on complex mathematical reasoning tasks requires curating high-quality problems with verified solutions: a process that is challenging and resource-intensive.

At the same time, the availability of high-quality synthetic data presents new opportunities to mitigate data scarcity and enhance statistical efficiency. For example, such data can be generated by LLMs or diffusion models, or retrieved from related auxiliary databases. However, it is challenging to construct procedures that leverage synthetic data with provable theoretical error rate control; because synthetic data may not reflect the real-world distribution. Naively pooling synthetic and real data in standard statistical inference methods can potentially result in poor performance that depends on the (unknown) quality of the synthetic data.

This tension between opportunities and risk motivates the goal of this paper: to design a general approach that "wraps" around any statistical inference method, providing it with the ability to harness synthetic data when beneficial while attaining rigorous, distribution-free error rate control.

---

[1]Software for reproducing the experiments is available at https://anonymous.4open.science/r/gespi.

Figure 1: **Overview of GESPI framework.** GESPI leverages a small real dataset and a large synthetic dataset. The procedure applies the base statistical method three times and aggregates the outputs in a way that guarantees error rate control while exploiting synthetic data when beneficial.

## 1.1 PREVIEW OF OUR METHOD AND KEY CONTRIBUTIONS

Our framework transforms any base statistical inference method to leverage synthetic data. As in Figure 1, let $\mathrm{Alg}$ denote the base inference method, which could represent conformal prediction, an A/B testing procedure, etc. Let $\mathcal{D}_n$ be the real observational dataset and $\tilde{\mathcal{D}}_N$ the abundant synthetic dataset, with sizes $n$ and $N \gg n$, respectively. Ideally, the synthetic data distribution would match the real one, but we make no such assumption.

With these notations in place, GESPI invokes the base inference algorithm three times:

1. **Base** $\mathrm{Alg}_{n,\alpha}$, uses the real data $\mathcal{D}_n$, targeting an error rate of $\alpha$ (e.g., 5%).
2. **Guardrail** $\mathrm{Alg}_{n,\alpha+\varepsilon}$, also uses only $\mathcal{D}_n$ but at a slightly higher target error rate level $\alpha + \varepsilon$ (e.g., 7%). This more relaxed level can increase the power compared to using $\alpha$.
3. **Synthetic-powered** $\mathrm{Alg}_{n,N,\alpha}$, which applies the base method to the pooled real and synthetic data, $\mathcal{D}_n \cup \tilde{\mathcal{D}}_N$, at level $\alpha$.

GESPI then aggregates (in a way we define) the outputs of these runs. We prove that this careful construction ensures the error rate never exceeds the guardrail bound $\alpha + \varepsilon$, regardless of the quality of the synthetic data (Theorems 3.2 and 3.3). At the same time, if the synthetic dataset is well aligned with the real distribution, GESPI adapts to use the synthetic-powered outcome. In this case, our theory shows that GESPI achieves tight $\alpha$ error rate control and mimics the effect of applying the base method to a larger real dataset. Our guarantees hold in finite samples and do not require any assumptions on the synthetic data distribution.

In Section 4 and Appendix C, we test the applicability of GESPI on a variety of tasks, including: (i) AlphaFold protein structure prediction, where we apply conformal risk control to bound the average fraction of residues with large prediction error; (ii) comparison of large reasoning models on math datasets, where we use hypothesis testing for win rate; (iii+iv) out-of-distribution detection, where we control the Type I error (in the single outlier case) and the family-wise error rate (for multiple outliers); (v) mechanistic interpretability of a Vision Transformer model, where we use a two-sample test to provide evidence for a property-specific role of an attention head. We also conduct ablation studies to evaluate the sensitivity of our method to $\varepsilon$ and to the synthetic data quality. These experiments support that GESPI adapts to the quality of synthetic data, providing meaningful improvements when possible while always maintaining error rate control.

## 2 RELATED WORK

Our proposed GESPI framework is inspired by Synthetic-Powered Predictive Inference (SPI) (Bashari et al., 2025a), a recent procedure for incorporating synthetic data into conformal prediction. SPI aims to construct prediction sets with distribution-free, finite-sample coverage guarantees that hold regardless of synthetic data quality. While we share the underlying motivation of leveraging synthetic data under rigorous error rate control, *our GESPI approach reformulates SPI, offering a new interpretation that serves as a foundation for addressing more general statistical inference problems.* Indeed, a key distinction between the two methods is that, instead of directly modifying the mechanism for constructing prediction sets (via transportation of non-conformity scores), GESPI treats conformal prediction as a "black-box" statistical method, without altering its inner workings.

Boyeau et al. (2025); Fisch et al. (2024); Oosterhuis et al. (2024) represent another related line of research, offering methods that use a large set of synthetically generated labels, together with a

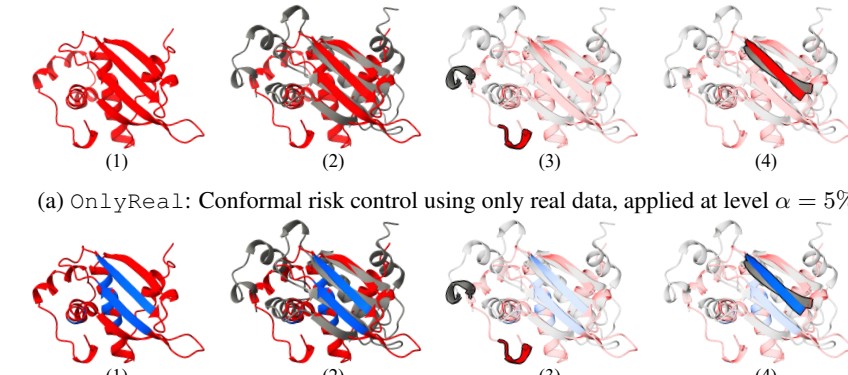

(a) `OnlyReal`: Conformal risk control using only real data, applied at level $\alpha = 5\%$

(b) `GESPI`: Our method applying conformal risk control at level $\alpha = 5\%$ and guardrail $\alpha + \varepsilon = 10\%$

Figure 2: **Visualization of protein structure prediction with error rate control.** Panels show protein T1029 predictions with residues abstained on by (a) `OnlyReal` and (b) `GESPI` methods. Red: residues abstained on; Blue: accepted residues. Gray: real experimental structure, aligned with AlphaFold2 predicted structure. Quantitative results {abstention ratio, risk}: `OnlyReal` − {100%, 0%}; `GESPI` − {85.6%, ≈ 7%}. See text in Section 4.1 for more details.

small set of human labels, for unbiased model evaluation—including the construction of confidence intervals for model performance. See Chatzi et al. (2024) for ranking, and Angelopoulos et al. (2023a;b) for constructing confidence bounds for parameters of interest. Notably, this line of work assumes that the covariates of the unlabeled and labeled data are i.i.d., in striking contrast to GESPI. We refer the reader to Appendix A for additional related work.

# 3 GENERAL SYNTHETIC-POWERED INFERENCE (GESPI)

For illustration, we first present applications of GESPI to different inference problems, and then introduce the general framework and theory.

## 3.1 GESPI FOR CONFORMAL PREDICTION AND RISK CONTROL

Consider a predictive inference task where we are given $n$ i.i.d. (real) data points $\mathcal{D}_n = (X_i, Y_i)_{i=1}^n \overset{\text{iid}}{\sim} P$, where $X_i \in \mathcal{X}$ and $Y_i \in \mathcal{Y}$ represent the features (e.g., an image) and labels (e.g., pedestrian), respectively. Given a new test input $X_{n+1}$, we aim to construct a prediction set $\widehat{C}_n(X_{n+1})$ that contains the unknown test label $Y_{n+1}$, such that for a user-specified level $\alpha \in (0, 1)$, e.g., 10%, the following holds: $\mathbb{P}_{\mathcal{D}_n \overset{\text{iid}}{\sim} P_{X,Y}, (X_{n+1}, Y_{n+1}) \overset{\text{iid}}{\sim} P_{X,Y}} \left\{ Y_{n+1} \notin \widehat{C}_n(X_{n+1}) \right\} \leq \alpha$.

Conformal prediction (Saunders et al., 1999; Vovk et al., 1999; 2005; Papadopoulos et al., 2002) is a framework that takes as input a holdout dataset $\mathcal{D}_n$ and transforms the output of any machine learning model into a prediction set $\widehat{C}_n(X_{n+1})$ for a new test point. Importantly, this prediction set satisfies the above finite-sample coverage guarantee, regardless of the underlying data distribution.

Conformal risk control extends conformal prediction beyond miscoverage rate control (0-1 loss) to any monotone risk function, such as the false negative rate or the F1 score. This framework constructs prediction sets $\widehat{C}_n$ with the following distribution-free risk control guarantee, for a target risk level $\alpha > 0$, e.g., 10% false negative rate, and $\ell(\cdot, \cdot)$ is the loss of interest:

$$\mathbb{E}_{\mathcal{D}_n \overset{\text{iid}}{\sim} P_{X,Y}, (X_{n+1}, Y_{n+1}) \overset{\text{iid}}{\sim} P_{X,Y}} \left[ \ell(\widehat{C}_n(X_{n+1}), Y_{n+1}) \right] \leq \alpha. \tag{1}$$

A fundamental limitation of these methods, however, is that when the sample size $n$ is small, they often produce excessively large and uninformative prediction sets, or exhibit unstable empirical risk with high variability—limiting their practical applicability.

Motivated by this small sample-size limitation, our GESPI procedure utilizes synthetic data to enhance sample efficiency. Let $\tilde{\mathcal{D}}_N = (\tilde{X}_j, \tilde{Y}_j)_{j=1}^N \overset{\text{iid}}{\sim} Q_{X,Y}$ denote the synthetic data. Let $\widehat{C}_{n,\alpha}(\cdot)$

denote the prediction set function obtained by applying conformal prediction to the real data $\mathcal{D}_n$ at level $\alpha$, and let $\widehat{C}_{n,N,\alpha}(\cdot)$ denote the corresponding function obtained from the pooled data $\mathcal{D}_n \cup \tilde{\mathcal{D}}_N$. Then, given an additional error tolerance level $\varepsilon > 0$, the GESPI prediction set is given as follows.

---

The GESPI conformal prediction set for a test point $X_{n+1} = x$ is given by

$$\hat{C}^{\text{GESPI}}(x) := \widehat{C}_{n,\alpha}(x) \cap \big( \widehat{C}_{n,N,\alpha}(x) \cup \widehat{C}_{n,\alpha+\varepsilon}(x) \big). \tag{2}$$

---

The intuition behind our GESPI construction is as follows. If the synthetic and real data distributions are identical, the second term $\widehat{C}_{n,N,\alpha}(x)$ amounts to applying conformal prediction on a larger real dataset, achieving the target risk level $\alpha$ while attaining tighter and more stable prediction sets.[2] At the same time, if the synthetic data is of poor quality, there are two guardrail bounds: (i) $\widehat{C}_{n,\alpha+\varepsilon}$ ensures the risk level of $\hat{C}^{\text{GESPI}}$ does not exceed $\alpha + \varepsilon$; and (ii) $\widehat{C}_{n,\alpha}$ prevents the GESPI prediction set from being even wider than the base method $\widehat{C}_{n,\alpha}$.

## 3.2 GESPI for one-sided hypothesis testing

We now turn to describe how GESPI can be used to enhance sample efficiency in the canonical problem of one-sided hypothesis testing; see e.g., Lehmann & Romano (2005b) for an overview of hypothesis testing. Consider a parameter $\theta$ of interest. Our goal is to test the following hypothesis

$$\mathcal{H}_0 : \theta = \theta_0 \text{ versus } \mathcal{H}_1 : \theta > \theta_0,$$

where $\mathcal{H}_0$ is the null hypothesis and $\mathcal{H}_1$ denotes the alternative. The parameter $\theta$ could represent, for example, the prediction error of a model, or the win rate of model A compared to model B. In the latter, rejecting the null that $\theta_0 = 0.5$ provides evidence that model A outperforms model B, an application we revisit later in the experiments (Section 4.2).

Let $\mathcal{D}_n = (X_i)_{i=1}^n$ denote the real dataset, with $\mathcal{D}_n \overset{\text{iid}}{\sim} P_\theta$, where $P_\theta$ is a distribution that depends on[3] $\theta \in \Theta$. When only a small dataset is available, statistical tests may suffer from low power; for example, the empirical mean estimate $\hat{\theta}$ can be noisy when evaluated on limited data, making it difficult to detect whether $\theta > \theta_0$ when using $\hat{\theta}$ as a test statistic.

Now suppose we also have access to a large synthetic dataset $\tilde{\mathcal{D}}_N$. To effectively leverage this additional data in a statistically valid way, we construct the GESPI testing procedure as follows. Given a testing procedure with Type I error rate control, let $\phi_{n,\alpha} \in \{0,1\}$ and $\phi_{n,N,\alpha} \in \{0,1\}$ denote the output of the tests applied to the real data $\mathcal{D}_n$ and the pooled data $\mathcal{D}_n \cup \tilde{\mathcal{D}}_N$, respectively, at level $\alpha$. By convention, we say that the test $\phi_{n,\alpha}$ rejects the null hypothesis if $\phi_{n,\alpha} = 1$, and fails to reject otherwise. Consider an additional error tolerance level $\varepsilon > 0$.

---

The GESPI hypothesis test $\phi^{\text{GESPI}} \in \{0,1\}$ is given by

$$\phi^{\text{GESPI}} := \phi_{n,\alpha} \text{ OR } \big( \phi_{n,N,\alpha} \text{ AND } \phi_{n,\alpha+\varepsilon} \big). \tag{3}$$

---

Intuitively, GESPI for hypothesis testing works as follows. We first apply the test $\phi_{n,N,\alpha}$ to the pooled dataset at level $\alpha$. If it rejects the null, we do not immediately reject, since the synthetic data may come from a distribution that differs significantly from the real one. To account for this, we also run the test on the real dataset $\phi_{n,\alpha+\varepsilon}$ at a slightly relaxed level $\alpha + \varepsilon$, and reject the null only if this test also rejects. In any case, if the base test $\phi_{n,\alpha}$ on the real dataset at level $\alpha$ rejects the null, we reject it immediately. This ensures that GESPI never loses power compared to the base test at level $\alpha$.

Importantly, for both Type I error rate control and power, the synthetic data do not need to follow the exact distribution of the real data. This flexibility stems from the structure of one-sided tests. To

---

[2]We usually have $\widehat{C}_{n,\alpha+\varepsilon} \subseteq \widehat{C}_{n,\alpha}$, as sets get wider when a tighter error guarantee (smaller $\alpha$) is required. In this case, we have $\widehat{C}_{n,\alpha+\varepsilon} \subseteq \hat{C}^{\text{GESPI}} \subseteq \widehat{C}_{n,\alpha}$. When the synthetic data has high quality, we expect that $\widehat{C}_{n,N,\alpha}$ is small, and does not increase $\widehat{C}_{n,\alpha+\varepsilon}$ by much. In such a setting, we will have that $\hat{C}^{\text{GESPI}}$ is close to $\widehat{C}_{n,\alpha+\varepsilon}$, which can be a much tighter set than the original set $\widehat{C}_{n,\alpha}$. This explains how GESPI can lead to tighter sets.

[3]This distribution could also depend on other parameters, which are omitted here for clarity.

see this, consider for illustration a simple setting where power is increasing[4] in the true parameter $\theta$ of the real distribution. Suppose that the pooled distribution can be described by the parameter $\theta^{\text{pool}}$. To control the Type I error when the null is true, it suffices that $\theta^{\text{pool}} \leq \theta = \theta_0$, even if the synthetic distribution differs from the real one (i.e., $\theta^{\text{pool}} \neq \theta$). Analogously, under the alternative $\theta > \theta_0$, it suffices that $\theta^{\text{pool}} > \theta_0$, without requiring $\theta^{\text{pool}} = \theta$. This property greatly expands the range of useful synthetic data that GESPI can leverage to improve power in one-sided hypothesis testing.

### 3.3 GESPI FOR ADDITIONAL TASKS: OUTLIER DETECTION AND MULTIPLE TESTING

GESPI can be used for a number of additional statistical inference problems, including outlier detection and multiple hypothesis testing. Due to space limitations, we are only able to present a high-level overview of these here, and defer details to the appendix (See Appendices D and H).

**Outlier detection.** Given a set of *inliers* $\mathcal{D}_n = (X_i)_{i=1}^n$, with $X_i \overset{\text{iid}}{\sim} P$, and a test point $X_{n+1}$, the goal is to determine whether $X_{n+1}$ is an inlier sampled from $P$—or an outlier sampled from a different distribution. Formally, this can be framed as testing the null hypothesis: $\mathcal{H}_0 : X_{n+1} \sim P$. Conformal outlier/anomaly detection Vovk et al. (2005); Bates et al. (2023); Laxhammar & Falkman (2011) provides a distribution-free test with finite-sample Type I error rate control. The associated test can then be used precisely as in Section 3.2 with GESPI to improve power with synthetic data.

**Multiple hypothesis testing.** Another important statistical inference problem is multiple hypothesis testing,[5] see e.g., Lehmann & Romano (2005b). Suppose we want to simultaneously test $m$ null hypotheses: $\mathcal{H}_{0,j}$, for $j = 1, \ldots, m$. When testing many hypotheses at once—such as identifying outliers among a batch of test points—the probability of incorrectly labeling at least one inlier as an outlier can increase rapidly if each hypothesis is tested separately at level $\alpha$. This motivates the goal of simultaneously testing all $m$ nulls while controlling the family-wise error rate (FWER), or more generally, the $k$-FWER (Lehmann & Romano, 2005a) at level $\alpha$: $\mathbb{P}\left\{\sum_{j=1}^m \mathbb{1}\left\{\mathcal{H}_{0,j} \text{ is true but rejected}\right\} \geq k\right\} \leq \alpha$, for some predetermined $k > 0$.

Suppose now that we have an FWER-controlling procedure which, given appropriate data, outputs a candidate set of rejections $\hat{\mathcal{S}}_{n,\alpha} = \{j : \mathcal{H}_{0,j} \text{ is rejected}\}$. Similarly, we define $\hat{\mathcal{S}}_{n,N,\alpha}$ as the rejection set obtained by applying the same FWER procedure using the real and synthetic data together. With this notation in place, we can now state the GESPI procedure.

> The GESPI rejection set $\hat{\mathcal{S}}^{\text{GESPI}} \subseteq \{1, \ldots, m\}$ for multiple testing is given by
> $$\hat{\mathcal{S}}^{\text{GESPI}} := \hat{\mathcal{S}}_{n,\alpha} \cup \left(\hat{\mathcal{S}}_{n,N,\alpha} \cap \hat{\mathcal{S}}_{n,\alpha+\varepsilon}\right). \tag{4}$$

### 3.4 THE PROPOSED GESPI FRAMEWORK

In this section, we describe our general framework for synthetic-powered inference that covers the applications in the previous section and extends beyond them.

**Problem setup.** Suppose we have a dataset $\mathcal{D}_n = (Z_1, Z_2, \ldots, Z_n) \in \mathcal{Z}^n$—e.g., in the setting of supervised learning, each $Z_i$ represents a (feature, outcome) pair $(X_i, Y_i)$. Consider a general statistical inference problem where the goal is to construct an algorithm[6] $\text{Alg} : \mathcal{Z}^\infty \to \mathcal{A}$ that maps the data to an action in the action space $\mathcal{A}$—with $\mathcal{Z}^\infty = \mathcal{Z} \cup \mathcal{Z}^2 \cup \mathcal{Z}^3 \cup \ldots$—and controls a risk:

$$\mathcal{R}(\text{Alg}, P) = \mathbb{E}_{\mathcal{D}_n \overset{\text{iid}}{\sim} P, V \sim \mathcal{T}(P)}\left[\ell(\text{Alg}(\mathcal{D}_n), V)\right] \leq \alpha, \text{ for all } P \in \mathcal{P} \text{ and } n \in \mathbb{N}. \tag{5}$$

or equivalently, $\sup_{P \in \mathcal{P}} \mathcal{R}(\text{Alg}, P) \leq \alpha$, for all $n \in \mathbb{N}$, for a predetermined target level $\alpha$.

Here, $\mathcal{P}$ is a set of distributions on $\mathcal{Z}$, $V \in \mathcal{V}$ denotes a quantity used for evaluating of the algorithm (e.g., a new test point in a predictive inference task, the target parameter in a confidence interval task, etc), and $\mathcal{T} : \mathcal{P} \to \mathcal{P}_\mathcal{V}$ is a function that maps the data distribution $P \in \mathcal{P}$ to a distribution on

---

[4]This holds generally, for one-dimensional families of probability distributions with the monotone likelihood ratio property, including exponential families such as the normal mean ; see Lehmann & Romano (2005b).

[5]Multiple hypothesis testing has a broad range of applications across science and engineering, see for instance Benjamini & Hochberg (1995); Efron (2012); Bretz et al. (2016), etc.

[6]We let the input of the algorithm be $\mathcal{Z}^\infty$ so that the same algorithm can be used for different sample sizes.

$\mathcal{V}$—where $\mathcal{P}_{\mathcal{V}}$ denotes the set of all distributions on $\mathcal{V}$—so that $\mathcal{T}(P)$ defines the distribution[7] of $V$. The function $\ell : \mathcal{A} \times \mathcal{V} \to \mathbb{R}^+$ is a loss function that evaluates the quality of the procedure $\text{Alg}(\mathcal{D}_n)$ with respect to $V$. See Table 1 for a non-exhaustive set of examples.

| Example | $\mathcal{Z}$ | $\text{Alg}(\mathcal{D}_n)$ | $V$ | Risk |
|---|---|---|---|---|
| Predictive inference | $\mathcal{X} \times \mathcal{Y}$ | Prediction set | New test point | Miscoverage rate |
| Hypothesis testing | $\mathcal{X}$ | Rejection indicator | None | Type I error |
| Multiple hypothesis testing | $\mathcal{X}$ | Rejection set | None | FWER |

Table 1: Examples of problems covered by our framework.

Now suppose we also have access to a synthetic/auxiliary dataset $\tilde{\mathcal{D}}_N = (\tilde{Z}_1, \ldots, \tilde{Z}_N) \in \mathcal{Z}^N$. Given a family of algorithms $\text{Alg}_\alpha$ for each $\alpha \in (0, 1)$ that attains the guarantee (5), we aim to construct an algorithm $\widetilde{\text{Alg}} : \mathcal{Z}^\infty \times \mathcal{Z}^\infty \to \mathcal{A}$ that takes both $\mathcal{D}_n$ and $\tilde{\mathcal{D}}_N$ as input, such that the synthetic-leveraging procedure $\widetilde{\text{Alg}}(\mathcal{D}_n, \tilde{\mathcal{D}}_N)$ improves upon the standard procedure $\text{Alg}_.(\mathcal{D}_n)$.

### 3.4.1 GENERAL ALGORITHM AND THEORETICAL GUARANTEES

To introduce our method, we begin with a simpler setting than in our examples, where we only aim to upper bound the risk, and not to lower bound it; we will consider this setting below.

**Condition 3.1** (informal). The action space $\mathcal{A}$ is partially ordered by $\preceq$, and for any $a_1, a_2 \in \mathcal{A}$ the minimum $a_1 \wedge a_2$ and the maximum $a_1 \vee a_2$ are well defined. In addition, the loss $\ell$ is bounded by a constant $c$, and monotone with respect to $\preceq$.

A formalized statement of Condition 3.1 is given in F.1. We note here that this is a mild condition, satisfied by all the inference problems discussed in this work; see Table 3. For example, in hypothesis testing, the action space consists of reject/accept $\{0, 1\}$ and the partial order $\preceq$ is defined by $\leq$. Recall that in the examples, our algorithm relies on taking unions and intersections or alternatively performing OR/AND operations. Generalizing our examples, our GESPI method takes the minimum ($\wedge$) of the output of two carefully chosen algorithms:

$$\widetilde{\text{Alg}}(\mathcal{D}_n, \tilde{\mathcal{D}}_N) = \text{Alg}_\alpha(\mathcal{D}_n \cup \tilde{\mathcal{D}}_N) \wedge \text{Alg}_{\alpha+\varepsilon}(\mathcal{D}_n), \tag{6}$$

where $\mathcal{D}_n \cup \tilde{\mathcal{D}}_N$ denotes the concatenated vector $(Z_1, \ldots, Z_n, \tilde{Z}_1, \ldots, \tilde{Z}_N)$.

Intuitively, the first component of the procedure (6), $\text{Alg}_\alpha(\mathcal{D}_n \cup \tilde{\mathcal{D}}_N)$, serves as the main part that incorporates the synthetic data, thereby producing a procedure based on a larger sample. The second component, $\text{Alg}_{\alpha+\varepsilon}(\mathcal{D}_n)$, at a relaxed level $\alpha + \varepsilon$, serves as a guardrail that does not depend on the synthetic data, and thus provides reliable statistical inference. The resulting procedure $\widetilde{\text{Alg}}(\mathcal{D}_n, \tilde{\mathcal{D}}_N)$ tightly controls the risk when the synthetic distribution closely resembles the real one, while still guaranteeing risk control at level $\alpha + \varepsilon$ even when the synthetic data is of low quality.

**Theorem 3.2.** *Given* $\alpha, \varepsilon > 0$*, suppose that algorithm* $\text{Alg}$ *satisfies* (5) *for* $\alpha$ *and* $\alpha + \varepsilon$*, and that Condition F.1 holds. Then the algorithm* $\widetilde{\text{Alg}}$ *defined in* (6) *satisfies*

$$\mathbb{E}_{\mathcal{D}_n \overset{\text{iid}}{\sim} P, \tilde{\mathcal{D}}_N \overset{\text{iid}}{\sim} Q, V \sim \mathcal{T}(P)} \left[ \ell(\widetilde{\text{Alg}}(\mathcal{D}_n, \tilde{\mathcal{D}}_N), V) \right] \leq \alpha + \min\{\varepsilon, c \cdot \text{d}_{\ell,\text{Alg}}(P, Q)\} \text{ for all } P, Q \in \mathcal{P},$$

*where* [8] $\text{d}_{\ell,\text{Alg}}(P, Q) = \text{d}_{\text{TV}}(P_{\ell,\text{Alg}}(P, Q), P_{\ell,\text{Alg}}(Q, Q))$*, and* $P_{\ell,\text{Alg}}(P, Q)$ *denotes the distribution of* $\ell(\text{Alg}_\alpha(\mathcal{D}_n \cup \tilde{\mathcal{D}}_N), V)$ *under* $\mathcal{D}_n \overset{\text{iid}}{\sim} P, \tilde{\mathcal{D}}_N \overset{\text{iid}}{\sim} Q, \mathcal{V} \sim \mathcal{T}(P)$.

The above result provides a general upper bound, which depends on the quality of the synthetic data as measured by $\text{d}_{\ell,\text{Alg}}(P, Q)$. If the synthetic data are of high quality, this term is small, and the resulting risk is controlled close to the level $\alpha$. However, even if the synthetic data are of arbitrary poor quality, and $\text{d}_{\ell,\text{Alg}}(P, Q) \to \infty$, the guardrail is active, and the risk is controlled at $\alpha + \varepsilon$. Tighter and more interpretable bounds for specific applications are detailed in Appendix H.

---

[7]In some cases, $V$ is deterministic, such as for confidence intervals, when it is the parameter of interest. In that case, the distribution of $V$ simplifies to a point mass.

[8]Here, $\text{d}_{\text{TV}}$ denotes the total variation distance.

**Inference with two-sided guardrails.** For the setting of two-sided guardrails, we have a similar version of GESPI, but one that also takes the maximum ($\vee$) with the output of the base algorithm:

$$\widetilde{\mathrm{Alg}}(\mathcal{D}_n, \tilde{\mathcal{D}}_N) = \mathrm{Alg}_\alpha(\mathcal{D}_n) \vee (\mathrm{Alg}_\alpha(\mathcal{D}_n \cup \tilde{\mathcal{D}}_N) \wedge \mathrm{Alg}_{\alpha+\varepsilon}(\mathcal{D}_n)), \qquad (7)$$

where $\varepsilon \geq 0$ is a predetermined level.

Similarly to procedure (6), a TV-distance-type bound can be derived for (7); we defer it to Appendix F due to space limitations. Here, we state a simpler observation that codifies that the two-sided guardrail version of GESPI is sandwiched between the base algorithm at levels $\alpha$ and $\alpha + \varepsilon$.

**Theorem 3.3.** *Suppose that Condition F.1 holds. Then given $\alpha, \varepsilon > 0$, the algorithm $\widetilde{\mathrm{Alg}}$ defined in* (7) *deterministically satisfies* $\mathrm{Alg}_\alpha(\mathcal{D}_n) \preceq \widetilde{\mathrm{Alg}}(\mathcal{D}_n, \tilde{\mathcal{D}}_N) \preceq \mathrm{Alg}_{\alpha+\varepsilon}(\mathcal{D}_n)$.

This result implies that the component $\mathrm{Alg}_{\alpha+\varepsilon}(\mathcal{D}_n)$ serves as a guardrail for the validity of the synthetic-leveraged procedure $\widetilde{\mathrm{Alg}}(\mathcal{D}_n, \tilde{\mathcal{D}}_N)$, while $\mathrm{Alg}_\alpha(\mathcal{D}_n)$ ensures that $\widetilde{\mathrm{Alg}}(\mathcal{D}_n, \tilde{\mathcal{D}}_N)$ is at least as useful as the base procedure $\mathrm{Alg}_\alpha(\mathcal{D}_n)$—in terms of prediction interval width, test power, etc.

In the case of the hypothesis testing example (3), Theorem 3.3 implies: $\widetilde{\mathrm{Alg}}(\mathcal{D}_n, \tilde{\mathcal{D}}_N) \in \{0, 1\}$ always rejects ($= 1$) whenever $\mathrm{Alg}_\alpha(\mathcal{D}_n)$ rejects (i.e., $\mathrm{Alg}_\alpha(\mathcal{D}_n) \preceq \widetilde{\mathrm{Alg}}(\mathcal{D}_n, \tilde{\mathcal{D}}_N)$), and never rejects unless $\mathrm{Alg}_{\alpha+\varepsilon}(\mathcal{D}_n)$ rejects (i.e., $\widetilde{\mathrm{Alg}}(\mathcal{D}_n, \tilde{\mathcal{D}}_N) \preceq \mathrm{Alg}_{\alpha+\varepsilon}(\mathcal{D}_n)$). This implies that $\widetilde{\mathrm{Alg}}(\mathcal{D}_n, \tilde{\mathcal{D}}_N)$'s Type I error is at most that of $\mathrm{Alg}_{\alpha+\varepsilon}(\mathcal{D}_n)$ ($\leq \alpha + \varepsilon$), while its power is at least that of $\mathrm{Alg}_\alpha(\mathcal{D}_n)$.

## 4 EXPERIMENTS

We now demonstrate the performance of GESPI across several applications. Additional results are provided in Appendix C, which also include controlled experiments on simulated data to provide further insight into GESPI 's performance. Full experimental details are provided in Appendix B.

**Methods.** For each applications, we compare the following methods: `OnlyReal`—the base inference method using only the real data. `OnlySynth`—the same inference method, but using only the synthetic data; this method does not have error rate control guarantees. `GESPI`—the proposed method that leverages both real and synthetic data, and supported by error rate control guarantees.

### 4.1 CONFORMAL RISK CONTROL FOR PROTEIN STRUCTURE PREDICTION

We consider the protein structure prediction problem, where the input $X$ is the amino-acid (residue) sequence and the target $Y$ is the corresponding 3D structure (coordinates per residue). The goal is to control the proportion of residues whose prediction error exceeds a threshold (e.g., 3Å). We achieve this by abstaining predicted coordinates of residues that are likely to have such large prediction errors. Formally, we define a prediction set $C_\lambda(X) \subseteq X$ as the subset of the residues abstained on. We employ the conformal risk control framework (Angelopoulos et al., 2024; Bates et al., 2021) and utilize real data to tune a threshold $\hat{\lambda}$ such that $\mathbb{E}\left[\frac{1}{|X|}\sum_{i \in X} \mathbb{I}\left\{\mathrm{err}_i > 3\text{Å}\right\} \cdot \mathbb{I}\left\{i \notin C_{\hat{\lambda}}(X)\right\}\right] \leq \alpha$. Here, $\mathrm{err}_i$ is the prediction error for residue $i$, which is formally defined in Appendix B.1. Note that the choice of 3Å is a standard scale for error, see, e.g., Jumper et al. (2021).

**Real data and prediction set formulation.** We use the CASP-14 dataset, focusing on monomer protein structure prediction using AlphaFold2 (Jumper et al., 2021). In addition to predicted structures, AlphaFold provides per-residue confidence scores (pLDDT, 0–100), which we use to construct the prediction set of residues abstained on: $C_\lambda(X) = \{i \in X : \mathrm{pLDDT}_i < \lambda\}$.

**Synthetic data.** A key component of AlphaFold2 is the use of multi-sequence alignments (MSAs), where the model searches a terabyte-scale database for related sequences to improve predictions. Inspired by this, we treat the same MSAs used by AlphaFold2 as synthetic data ($\tilde{X}$) and generate corresponding predicted structures to approximate the prediction error, since true structures for the synthetic data are unavailable. *The appeal of this construction is that we show how the powerful MSA component of AlphaFold2 can be utilized beyond its original purpose of improving predictions: we harness the MSA sequences to form high-quality synthetic data that boost statistical inference.* Further details on the construction of the synthetic data are provided in Appendix B.1.

**Experimental setup and metrics.** We use $n = 10$ out of 38 proteins to form the real dataset $\mathcal{D}_n$, and reserve the remaining proteins for the test set; the synthetic dataset contains $N = 1,000$ proteins.

We apply `GESPI` with $\varepsilon = 5\%$, chosen relative to the $\alpha$ levels ($5 - 15\%$) used in the experiments. Results are averaged over 10 repeated trials, including the average risk (average fraction of residues with error $> 3$Å), the average fraction of residues abstained on, and the selected pLDDT threshold $\hat{\lambda}$.

We begin by visualizing the differences between the base `OnlyReal` method and our `GESPI` procedure. To illustrate how our method performs, we select protein T1029, for which the AlphaFold prediction is only partly accurate, and show the resulting prediction sets obtained by `OnlyReal` (Figure 2a) and `GESPI` (Figure 2b). Panel (1) shows the predicted structure, with residues abstained on highlighted in red and accepted residues in blue. Observe how `OnlyReal` conservatively abstains from all residues; this is a consequence of the limited real data used to tune $\lambda$. By contrast, `GESPI` abstains less, demonstrating the advantage of using synthetic data. Panel (2) shows the predicted structure (red and blue) aligned with the true structure (gray). Panel (3) highlights a small subset of residues for which the prediction is clearly inaccurate and where both methods abstain. Lastly, panel (4) shows a well-aligned region where `OnlyReal` abstains unnecessarily, while `GESPI` correctly accepts these residues. For completeness, we also visualize protein T1078 in Appendix C.2, where AlphaFold achieves relatively high accuracy.

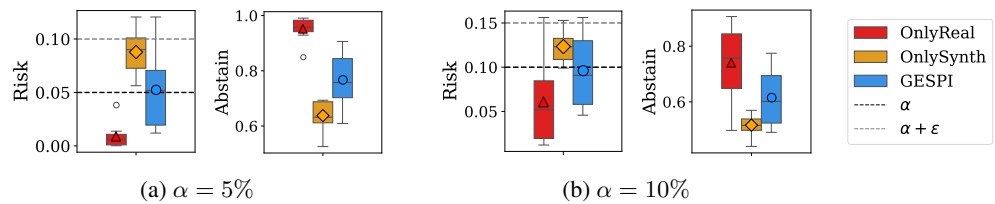

(a) $\alpha = 5\%$            (b) $\alpha = 10\%$

Figure 3: **Performance comparisons for protein structure prediction with error rate control**. Conformal risk control methods applied at target levels (a) $\alpha = 5\%$ and (b) $10\%$. Left: average risk (fraction of residues with error $> 3$Å). Right: average abstention rate.

Figure 3 presents quantitative results for two $\alpha$ levels, showing a consistent trend: `OnlyReal` conservatively controls the risk but at the cost of a high abstention rate. In contrast, `GESPI` achieves risk close to the nominal $\alpha$ level while reducing the abstention rate. Crucially, `OnlySynth` serves only as a heuristic baseline and does not provide risk control guarantees. Additional results for $\alpha = 15\%$ level, along with the selected thresholds for all $\alpha$ levels, are provided in Appendix C.2.

### 4.2 Hypothesis testing for win rate of large reasoning models

Given two large language models (LLMs), we aim to test whether `model A` outperforms `model B` on a specific type of task. Formally, we consider the hypotheses $\mathcal{H}_0 : p = 0.5$ vs. $\mathcal{H}_1 : p > 0.5$, where $p$ denotes the win rate of `model A` over `model B`.

**Data.** We evaluate the win rate on the AIME25 dataset, which consists of 30 challenging reasoning math questions. The goal is to pin down the ranking of the models on AIME25 as closely as possible, which is hindered by the very small number of questions for this competition. For synthetic data, we use a subset of the OlympiadBench (He et al., 2024) math competition questions, which resemble AIME problems but are drawn from a different distribution and therefore cannot be treated as real test data.[9] For each question and model, we record whether the model's answer is correct. Further details on the experimental setup are provided in Appendix B.2.

**Experimental setup and metrics.** We randomly choose $n = 15$ AIME25 math problems and $N = 100$ synthetic problems to from $\mathcal{D}_n$ and $\tilde{\mathcal{D}}_N$; the choice of using half of the real dataset allows us to run multiple repetitions and estimate the power and Type I error. In addition to this standard experiment, we evaluate the validity of our method by randomly shuffling the responses of the two models, which corresponds to the null hypothesis. This allows us to estimate the Type I error rate.

Figure 4 presents two model comparisons, each involving a distinct pair of LLMs. In both cases, the rejection rate (power) is well above the target level $\alpha$, indicating that the first model outperforms the second. Our proposed method, `GESPI`, achieves higher power than `OnlyReal`, while in the shuffled-answers setting, both methods achieve Type I error at the target level $\alpha$.

---

[9]Since OlympiadBench was released in 2024, before AIME25, these two datasets are non-overlapping.

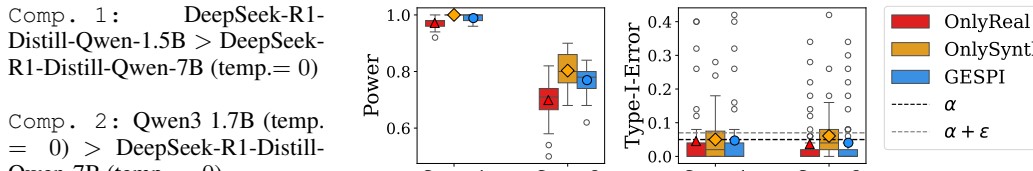

Comp. 1: DeepSeek-R1-Distill-Qwen-1.5B > DeepSeek-R1-Distill-Qwen-7B (temp.= 0)

Comp. 2: Qwen3 1.7B (temp. = 0) > DeepSeek-R1-Distill-Qwen-7B (temp. = 0)

Figure 4: **Performance comparisons for LLM win rate on AIME25 dataset**. Hypothesis testing methods applied at level $\alpha = 5\%$ and $\varepsilon = 2\%$. Left: Description of model comparisons. Middle: Power, comparing the rejection rate under the standard setting. Right: Type I error, measured in the shuffled-response setting where the null holds.

### 4.3 OUTLIER DETECTION WITH CONTAMINATED REFERENCE SET

We now consider the task of conformal outlier detection (Section 3.3) for both single and multiple testing. Conformal outlier detection guarantees error rate (Type I error/FWER) control given a reference set of pure inliers. In practice, however, one often has access to only a small inlier dataset $\mathcal{D}_n$—which can make conformal methods conservative (Bashari et al., 2025b)—as well as a larger, unlabeled dataset, $\tilde{\mathcal{D}}_N$, that is contaminated with a small percentage of outliers, say $q\%$. An ideal, but infeasible, Oracle would annotate $\tilde{\mathcal{D}}_N$ and use only the inliers from both datasets as reference data. As a cheap, annotation-free alternative, we use an ML model to trim the top $q\%$ of samples from $\tilde{\mathcal{D}}_N$ that are suspected to be outliers by the model, and treat the remainder as synthetic data consisting of "pseudo-inlier" points. Notably, this trimming can make OnlySynth less conservative, but does not guarantee error rate control at the desired level.

**Data.** We compare the performance of conformal outlier detection methods on three benchmark tabular datasets for outlier detection: *shuttle* (Catlett, 1992), *credit card* (Group, 2013), and *KDDCup99* (Stolfo et al., 1999). See Appendix B.3 for additional details on the experiments.

Figure 5 presents the performance for both single- and multiple-outlier testing, where both showing a similar trend. The OnlyReal method conservatively controls the error rate, but obtains low power due to the small sample size. The OnlySynth method fails to control the error rate, whereas the Oracle method achieves error rate tightly regulated around the target level, as expected. The error rate of our GESPI method is close to the nominal $\alpha$ level, while achieving substantially higher power than OnlyReal, approaching the performance of the Oracle. Additional results for both single and multiple hypothesis testing are provided in Appendix C.3.

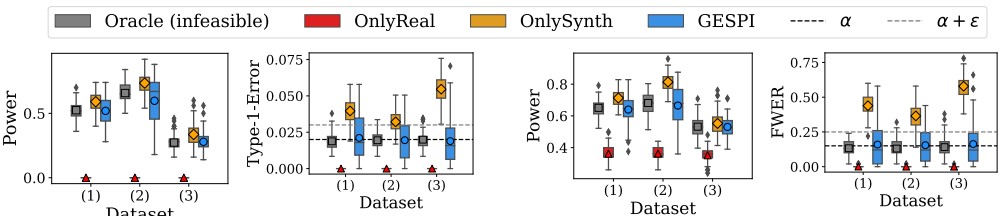

Figure 5: **Performance comparisons for outlier detection**. Evaluated on three datasets: (1) shuttle, (2) credit-card, (3) KDDCup99. Left two panels: single-outlier case ($\alpha = 2\%$, $\varepsilon = 1\%$). Right two panels: multiple-outlier case ($\alpha = 15\%$, $\varepsilon = 10\%$).

## 5 DISCUSSION

This work introduces GESPI, a general wrapper for statistical inference that safely leverages synthetic data while preserving distribution-free, finite-sample guarantees. Extensive experiments across different applications show that GESPI adapts automatically to data quality: it yields substantial gains when synthetic data are useful, but never underperforms the base method. One limitation is that the power gain depends on the quality of the synthetic data. A promising future direction is to develop adaptive methods for selecting synthetic data to further enhance statistical power.

## REPRODUCIBILITY STATEMENT

All experimental details, including dataset information, are provided in Section 4 and appendices B and C. An anonymous GitHub repository, containing the implementation of GESPI, baseline methods, and code to reproduce all experiments, is available at https://anonymous.4open.science/r/gespi. All assumptions and theoretical results stated in Section 3.4 and appendices E, F and H, with all proofs provided in Appendix I.

## LLM USAGE

LLMs did not play a significant role in this work and were only used for grammar polishing in writing.

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

## A  ADDITIONAL RELATED WORK

Our formulation can be viewed through the lens of statistical decision theory. In particular, our work is connected to robust statistics, statistical estimation under distribution shift. The guarantee (5) can be viewed as saying that the minimax optimal risk over the class $\mathcal{P}$ of probability distributions is upper bounded by $\alpha$. The algorithm $\mathrm{Alg}$ certifies this.

Then, Theorem 3.2 can be viewed as an upper bound on the minimax risk for the partial distribution shift problem where we observe $n$ datapoints from the original distribution, and $N$ datapoints from the shifted distribution. The algorithm $\widetilde{\mathrm{Alg}}$ certifies this. There has been work on statistical learning under a variety of distribution shifts, including the Huber contamination model (e.g., Huber, 1964; 1965; 2004; Hampel et al., 2005; Chen et al., 2016; 2018; Zhu et al., 2022, etc) and Wasserstein shifts Zhu et al. (2022); Liu & Loh (2022); Chao & Dobriban (2023). However, these works typically focus on the scenario where either (1) some random or adversarial subset of the data (that is not known to the analyst) is corrupted, or (2) all datapoints are potentially corrupted. We are not aware of studies of the scenario where a *known* subset is from the ground truth distribution, while the remaining subset is potentially shifted.

Our work is also related to transfer learning (Pan & Yang, 2010; Weiss et al., 2016; Zhuang et al., 2020), semi-supervised learning Blum & Mitchell (1998); Ben-David et al. (2008); Chapelle et al. (2006) and other forms of structured distribution shift, where a known part of the data is from the target distribution, while another part of the data shares some similarities with the target (see, e.g., Storkey, 2013; Shimodaira, 2000; Sugiyama & Kawanabe, 2012; Kouw & Loog, 2018; Qiu et al., 2024). For instance, in semi-supervised learning, we have additional unlabeled data from the target. The question is then how to use the additional data. However, most work in this area concerns certain known forms of relations between the auxiliary data and the original data (e.g., in semi-supervised learning, the distribution of the features is informative for the conditional distribution of the outcome given the features), and we are not aware of studies where using arbitrarily shifted data has been provably used to benefit in transfer learning.

Our work is also related to work in causal inference that develops methods to pool unbiased estimators from real (experimental) data with biased but more accurate estimators from another (usually observational) distribution (see, e.g., Cheng & Cai, 2021; Rosenman et al., 2023; Rosenman, 2025; De Bartolomeis et al., 2025, etc). Unlike these works, we do not focus on causal inference.

There has been a large amount of work on using prior data and information in hypothesis testing, see e.g., Spjøtvoll (1972); Roeder & Wasserman (2009); Bourgon et al. (2010); Dobriban et al. (2015), etc. In this line of work, the question is: How to use data from prior studies on the same hypotheses (at the simplest level, p-values for the same null hypotheses) to improve power in multiple testing. Strategies have been developed that rely on choosing a class of methods, such as based on p-value weighting (e.g., the weighted Bonferroni method), and then characterizing the optimal choice of weights as well as how to estimate them based on the available data. Our work is different because we do not assume an explicit statistical model that connects the prior and current data sets, but instead try to be adaptively useful when their distributions are close.

Other recent related works include Decruyenaere et al. (2023), who discuss how to use synthetic tabular data in statistical inference problems, arguing that such synthetic data cannot be used as if it were real data. Keret & Shojaie (2025) discuss using synthetic data in generalized linear models, proposing to use mis-specified linear regression estimators that they argue can have a faster speed of convergence. An important prior work is byMcCaw et al. (2024), which develops methods for improved confidence interval construction in mixed linear models using synthetic data. The crucial difference between this approach and ours is that we do not make any explicit modeling assumptions on the synthetic data.

More broadly, our methodology enables the application of statistical methods to the analysis of a variety of generative AI models. In our work, we illustrate this by studying the evaluation of large reasoning models as well as the identification of internal components of vision transformer models. Both of these areas of application (evaluation and identification of internal components of black-box models) has been discussed as promising avenues where statistical methods can be used (Dobriban, 2025), and our work supports that thesis.

# B EXPERIMENTAL DETAILS

## B.1 CONFORMAL RISK CONTROL FOR PROTEIN STRUCTURE PREDICTION

**Model.** We use AlphaFold2 (Jumper et al., 2021) through ColabFold (Mirdita et al., 2022) with the MMseqs2 search strategy over the UniRef and Environmental databases for MSA construction. Each prediction is run with five models, three recycles, and an early stopping criterion at a confidence score of 97.

**Data.** The real dataset is taken from CASP-14. For each CASP-14 protein, we retrieved the corresponding MSA files generated during the AlphaFold run. For every protein sequence appearing in the UniRef MSAs, we queried the AlphaFold Database (Varadi et al., 2024) to collect predicted structures, per-residue pLDDT scores, and PAE matrices (an additional AlphaFold output representing the predicted alignment error for each residue pair). CASP-14 proteins for which predictions were unavailable for any of their MSAs were excluded, leaving a total of 38 proteins. In each experiment, the CASP-14 proteins are split into real ($\mathcal{D}_n$) and test sets. For a given realization of the real dataset $\mathcal{D}_n$, the synthetic dataset is constructed using only the MSAs of proteins in this set. If more than 1,000 synthetic samples are available, we randomly sample a balanced subset of 1,000, ensuring roughly equal representation from each protein; otherwise, we include all available samples.

**Prediction error.** Let pred and real denote the predicted and real structures of a given sequence $X$, respectively, where pred$[i]$ and real$[i]$ are the 3D coordinates of the $i$-th residue. The per-residue prediction error is defined as the average absolute difference between the pairwise distances from residue $i$ to all other residues:

$$\text{err}_i = \frac{1}{|X|} \sum_{j=1}^{|X|} \big| \|\text{pred}[i] - \text{pred}[j]\|_2 - \|\text{real}[i] - \text{real}[j]\|_2 \big|.$$

Intuitively, err$_i$ quantifies how well the local spatial geometry relative to residue $i$ is preserved in the predicted structure compared to the real one.

Building on this, the risk from Section 4.1

$$\mathbb{E}\left[ \frac{1}{|X|} \sum_{i \in X} \mathbb{I}\left\{ \text{err}_i > 3\text{Å} \right\} \cdot \mathbb{I}\left\{ i \notin C_{\hat{\lambda}}(X) \right\} \right] \leq \alpha,$$

which is the proportion of residues with error greater than 3Å, is bounded by 1 by definition. In practice, however, AlphaFold2 predictions are fairly accurate, so the observed risk is far below this maximum. Conformal risk control uses an upper bound on the risk, denoted by $B$, to account for the unknown risk of a test point. Given AlphaFold2's performance, we set $B = 0.5$, meaning that in the worst case, at most half of the residues may have a prediction error exceeding 3Å.

For synthetic data, the true structures are unavailable. Instead, we approximate the per-residue error using the predicted alignment error (PAE) matrix. Specifically, the synthetic error for residue $i$ is taken as the mean of the $i$-th row of the PAE matrix, and we discard any synthetic protein for which more than 50% of residues exceed the 3Å threshold under this proxy.

**Visualization.** All protein visualizations were performed using UCSF ChimeraX (Pettersen et al., 2021), developed by the Resource for Biocomputing, Visualization, and Informatics at the University of California, San Francisco, with support from National Institutes of Health R01-GM129325 and the Office of Cyber Infrastructure and Computational Biology, National Institute of Allergy and Infectious Diseases.

## B.2 HYPOTHESIS TESTING FOR WIN RATE COMPARISON BETWEEN LLMS

**Testing the null hypothesis.** In this setting, we aim to test whether `model A` performs better than `model B` in terms of win rate. Formally, we test the null hypothesis that the win rate of `model A` over `model B` equal to 0.5, rejecting it when we have sufficient evidence that `model A` wins more frequently.

In more detail, one observation corresponds to a trinomial random variable $Z \sim$ Trinomial$(p_{\text{win}}, p_{\text{equal}}, p_{\text{loss}})$, where one coordinate of $Z = (Z_{\text{win}}, Z_{\text{equal}}, Z_{\text{loss}})$ is equal to unity, and all others are equal to zero. After observing $n$ trials/observations, we summarize them into $N \sim$ Trinomial$(n;\ p_{\text{win}}, p_{\text{equal}}, p_{\text{loss}})$, where $N = (N_{\text{win}}, N_{\text{equal}}, N_{\text{loss}})$ is the vector of corresponding counts.

Now, for any given observed value $N_{\text{equal}} = n_{\text{equal}}$, the conditional distribution of $N_{\text{win}}$ is

$$N_{\text{win}} \mid N_{\text{equal}} = n_{\text{equal}} \ \sim \ \text{Binomial}(n - n_{\text{equal}}, p_{\text{win}}/(p_{\text{win}} + p_{\text{loss}})).$$

Under the null hypothesis, this is a Binomial$\left(n - n_{\text{equal}}, \frac{1}{2}\right)$ distribution. Hence, we can apply a randomized binomial (or sign) test conditionally on $n_{\text{equal}}$, at level $\alpha$. This test will maintain level $\alpha$ even unconditionally.

**Models.** We use the following models from the vLLM library for the comparisons:

- `deepseek-ai/DeepSeek-R1-Distill-Qwen-1.5B`
- `deepseek-ai/DeepSeek-R1-Distill-Qwen-7B`
- `Qwen/Qwen3-1.7B`

Unless specified otherwise, models are run with temperature 0.6, top-p = 0.95, top-k = 20, and min-p = 0. For these runs, we generate 64 answers per question and take the majority vote as the final answer. The maximum token limit for all runs is 32,768.

**Data and answer verification.** We use datasets from the Hugging Face datasets library:

- Real data: AIME25 test split (`math-ai/aime25`), containing 30 challenging math reasoning questions in English.
- Synthetic data: a subset of OlympiadBench (`Hothan/OlympiadBench OE_TO_maths_en_COMP`), containing English-language math reasoning questions without full proofs.

For each question, we use the following system prompt:

```
You are a helpful AI Assistant. First, think through the reasoning
inside <think/>...</think>. Then, always present the final answer
in \boxed{}.
```

The question itself is provided as the user input.

To determine whether a model's answer is correct, we first extract the answer from the `\boxed{}` in the model's response and apply `math_verify.verify`. If the model fails to produce a complete answer within the token limit, the response is counted as incorrect and no further evaluation is performed. In cases where an answer is flagged as incorrect by `math_verify.verify` (e.g., due to notation mismatches), we re-evaluate it using the LLM `deepseek-ai/deepseek-math-7b-instruct`. This LLM returns "Yes" if it considers the answer correct and "No" otherwise, and this output is used as the final evaluation. Specifically, we use the following prompt for the LLM:

```
You are given a math problem, a reference solution, and a generated
answer. Determine if the generated answer is equivalent to the so-
lution.  Answer "Yes" or "No".

Problem:
{prompt}

Reference Solution:
{solution}

Generated Answer:
{generated_answer}

Are they equivalent? Answer Yes or No:
```

**Experimental setup and metrics.**    As described in the main manuscript, we consider two experimental schemes:

- **Original answers:** We compare the two models using their original responses. If `model A` outperforms `model B`, the null hypothesis should be false, and—given sufficient evidence against the null—we expect the rejection rate to exceed the nominal level $\alpha$.
- **Shuffled answers:** As a complementary baseline, we randomly shuffle the responses of the two models. In this setting, the null hypothesis holds by design. This scheme allows us to estimate the Type I error rate.

For each scheme, we estimate the power and Type I error by running 50 independent trials. In each trial, we randomly sample the real and synthetic datasets and test the null hypothesis on the resulting subset. To quantify variability, we repeat this entire procedure independently 50 times. For the shuffled-answers scheme, the random reassignment of responses is performed once at the start of each outer replicate and kept fixed across its 50 inner trials.

### B.3    OUTLIER DETECTION WITH CONTAMINATED REFERENCE SET

**Experimental setup and metrics.**    For single hypothesis testing (Type I error rate control), we use the following setup: For Shuttle and KDDCup99, we use 5,000 training datapoints; for Credit Card, 2,000. Both training and reference sets are contaminated at a 5% rate. The contaminated reference set contains 2,500 datapoints, while the clean reference set, $\mathcal{D}_n$, contains 40 inlier points. As the outlier detection model used by the conformal outlier detection framework, we use Isolation Forest (Liu et al., 2008), implemented using `scikit-learn` with 100 estimators. Test sets contain 950 inliers and 50 outliers. We report the average detection power and the average Type I error over 100 independent trials.

For multiple hypothesis testing (FWER control), the setup is similar with two key differences. First, the clean reference set consists of 100 inlier points. Second, the test set contains 1,000 datapoints with 5% outliers, randomly partitioned into 50 batches. For each batch, every method produces a rejection set; we record whether it contains at least one false rejection (indicator 0/1). Averaging these indicators across all batches yields the empirical FWER, computed over 100 independent trials. All methods use the Simes-Hochberg procedure (Hochberg, 1988) for testing.

# C   ADDITIONAL EXPERIMENTS

## C.1   HYPOTHESIS TESTING WITH SIMULATED DATA

In this section, we present controlled experiments on simulated data to systematically study the performance of GESPI.

We focus on hypothesis testing for a single parameter. Let $\mathcal{D}_n = (X_i)_{i=1}^n$ denote real datapoints drawn i.i.d. from $\text{Binomial}(n, \rho)$, and let $\tilde{\mathcal{D}}_N = (\tilde{X}_i)_{i=1}^N$ denote synthetic (auxiliary) datapoints drawn from a related but potentially different distribution $\text{Binomial}(N, \rho_{\text{synt}})$. We test the null hypothesis

$$\mathcal{H}_0 : \rho = 0.5 \text{ versus } \mathcal{H}_1 : \rho > 0.5.$$

We use the randomized binomial test and report both power (under the alternative $\mathcal{H}_1$) and Type I error (under the null $\mathcal{H}_0$).

**Experimental setup and metrics.**   Unless stated otherwise, the real dataset contains 50 datapoints and the synthetic dataset 500 datapoints. The target Type I error level is $\alpha = 5\%$, and GESPI is applied with $\varepsilon = 2\%$. Under the alternative, the real data parameter is set to $\rho = 0.6$, while the synthetic data parameter is $\rho_{\text{synt}} = 0.55$. Each experiment is repeated 100 times to estimate the power and Type I error, and the entire procedure is repeated 100 times to evaluate variability.

**The effect of the distance between real and synthetic distributions.**   We begin by examining how performance varies with different choices of $\rho_{\text{synt}}$. Figure 6 summarizes results for two regimes: $\rho = 0.6$ (alternative, first row) and $\rho = 0.5$ (null, bottom row).

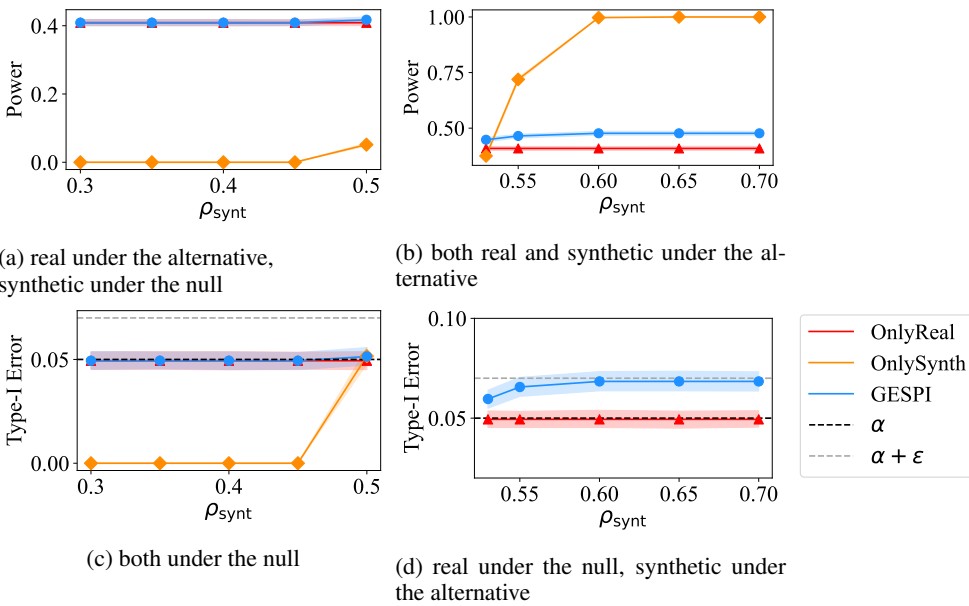

(a) real under the alternative, synthetic under the null

(b) both real and synthetic under the alternative

(c) both under the null

(d) real under the null, synthetic under the alternative

Figure 6: **Performance comparison as a function of $\rho_{\text{synt}}$.** Hypothesis testing methods across different values of $\rho$ applied at level $\alpha = 5\%$ and $\varepsilon = 2\%$. Top row: $\rho = 0.6$ (alternative). Bottom row: $\rho = 0.5$ (null).

Figure 6a considers a setting where the alternative holds for the real data, while the synthetic data correspond to the null. Here, the synthetic data do not provide useful information for inference, and as a result, both OnlyReal and GESPI achieve comparable power.

In contrast, Figure 6b presents the case where the alternative holds for both the real and synthetic data. In this setting, GESPI achieves higher power than OnlyReal, while OnlySynth attains even higher power. This is because OnlySynth naively treats the synthetic data as if it were real—a strategy that is invalid in this distribution-free setting, where the synthetic data may differ arbitrarily

from the real distribution. Importantly, even when $\rho_{\mathrm{synt}} \neq \rho$, we still observe a clear gain in power. This highlights that synthetic data do not need to perfectly match the real distribution; it suffices that the synthetic data support the same hypothesis as the real one (the alternative, in this case).

Figure 6c presents the setting where the null holds for both the real and the synthetic data. All methods control the Type I error at level $\alpha$, while `OnlySynth` exhibits Type I error approaching zero when $\rho_{\mathrm{synt}} < 0.5$. Importantly, `GESPI` still benefits from the synthetic data in this setting, which highlights the point made above: the synthetic data need not perfectly match the real distribution, as long as they support the same hypothesis (here, the null).

Finally, Figure 6d shows the case where the null holds for the real data, but the synthetic data follow the alternative. As in Figure 6a, the synthetic data are uninformative for inference. `OnlySynth` obtains very high Type I error and is therefore omitted from the plot (its Type I error matches the power reported in Figure 6b). In contrast, `GESPI` controls the Type I error at most $\alpha + \varepsilon$, as guaranteed by Theorem F.2.

**The effect of $\varepsilon$.** Figure 7 investigates how different values of $\varepsilon$ affects the performance of `GESPI`. The experiments follow a similar setup to the one in Figure 6, with results shown as a function of $\varepsilon$ across four different scenarios: (a) the alternative holds for both real and synthetic data ($\rho = 0.6$, $\rho_{\mathrm{synt}} = 0.55$), where the synthetic data provide a weaker signal against the null; (b) the null holds for both datasets ($\rho = \rho_{\mathrm{synt}} = 0.5$); (c) the alternative holds for the real data ($\rho = 0.6$) and the null for the synthetic data ($\rho_{\mathrm{synt}} = 0.5$); (d) the null holds for the real data ($\rho = 0.5$) and the alternative for the synthetic data ($\rho_{\mathrm{synt}} = 0.55$).

In scenario (a), shown in Figure 7a, `GESPI` achieves higher power than `OnlyReal`, and its power increases with $\varepsilon$. Intuitively, this occurs because larger values of $\varepsilon$ make the test on the real dataset at level $\alpha + \varepsilon$ more liberal, which in turn allows `GESPI` to rely more on the pooled-data decision: the method rejects the null if both the pooled-data test at level $\alpha$ and the real-data test at level $\alpha + \varepsilon$ reject. In scenario (b) from Figure 7b, where both datasets follow the null, all methods achieve Type I error close to the nominal level $\alpha$.

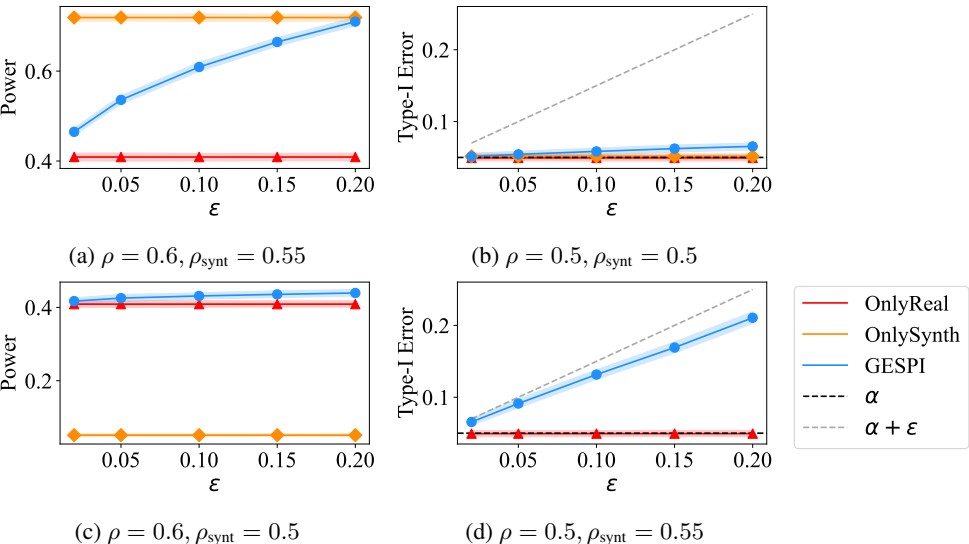

(a) $\rho = 0.6, \rho_{\mathrm{synt}} = 0.55$      (b) $\rho = 0.5, \rho_{\mathrm{synt}} = 0.5$

(c) $\rho = 0.6, \rho_{\mathrm{synt}} = 0.5$      (d) $\rho = 0.5, \rho_{\mathrm{synt}} = 0.55$

Figure 7: **Performance comparison as a function of $\varepsilon$.** Hypothesis testing methods across different values of $\rho$ and $\rho_{\mathrm{synt}}$ applied at level $\alpha = 5\%$.

For scenarios (c) and (d), where the real and synthetic datasets correspond to opposing hypotheses, the synthetic data do not provide useful information for inference. In scenario (c), shown in Figure 7c, `GESPI` and `OnlyReal` achieve comparable power, as expected. In scenario (d), Figure 7d, `OnlyReal` controls the Type I error at level $\alpha$, while `GESPI` exhibits a higher Type I error, but it remains controlled at level $\alpha + \varepsilon$, as guaranteed by Theorem F.2.

**The effect of sample size.** Figure 8 compares the performance of various hypothesis testing methods as a function of the real dataset size $n$ and the synthetic dataset size $N$, under the alternative hypothesis for both real and synthetic data. Following the left panel in that figure, we can see that GESPI consistently improves power compared to the baseline, OnlyReal, across all values of $n$. The right panel demonstrates a similar trend with respect to $N$: GESPI outperforms OnlyReal, and for sufficiently large synthetic datasets ($N \geq 500$), its power remains relatively unchanged under these conditions.

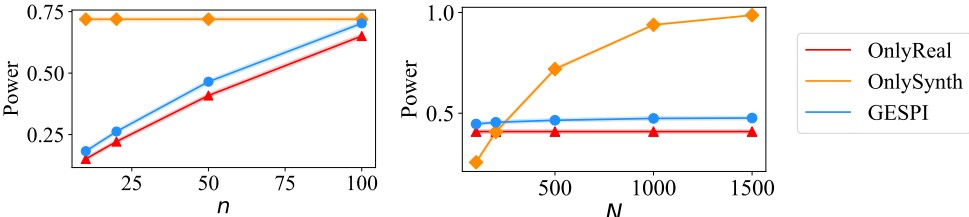

Figure 8: **Performance comparison as a function of the real dataset size $n$ and the synthetic dataset size $N$.** Hypothesis testing methods under the alternative ($\rho = 0.6$ and $\rho_{\text{synt}} = 0.55$) applied at level $\alpha = 5\%$ and $\varepsilon = 2\%$.

**The effect of the target error rate $\alpha$.** Figure 9 reports performance as a function of the target Type I error $\alpha$ across the four settings considered in Figure 7, with GESPI applied using $\varepsilon = 5\%$.

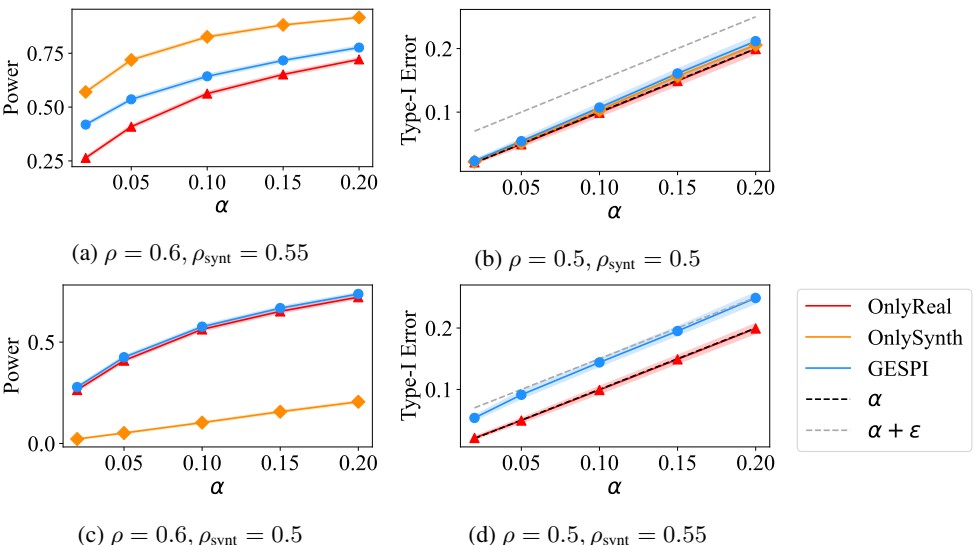

(a) $\rho = 0.6, \rho_{\text{synt}} = 0.55$

(b) $\rho = 0.5, \rho_{\text{synt}} = 0.5$

(c) $\rho = 0.6, \rho_{\text{synt}} = 0.5$

(d) $\rho = 0.5, \rho_{\text{synt}} = 0.55$

Figure 9: **Performance comparison as a function of the target Type I error level $\alpha$.** Hypothesis testing methods across different values of $\rho$ and $\rho_{\text{synt}}$. GESPI applied with $\varepsilon = 5\%$.

In Figure 9a, where the alternative holds for both real and synthetic data, GESPI consistently achieves higher power than OnlyReal across all values of $\alpha$. In Figure 9b, where the null holds for both datasets, all methods control the Type I error close to the nominal level. As before, when the real and synthetic data correspond to opposing hypotheses (Figures 9c and 9d), the synthetic data provide no useful information for inference. In the former case, GESPI and OnlyReal attain comparable power, while in the latter, OnlyReal controls the Type I error at level $\alpha$ and GESPI at level $\alpha + \varepsilon$, as guaranteed.

Finally, Figure 10 presents the same analysis for $\varepsilon = 2\%$, showing the same overall trends.

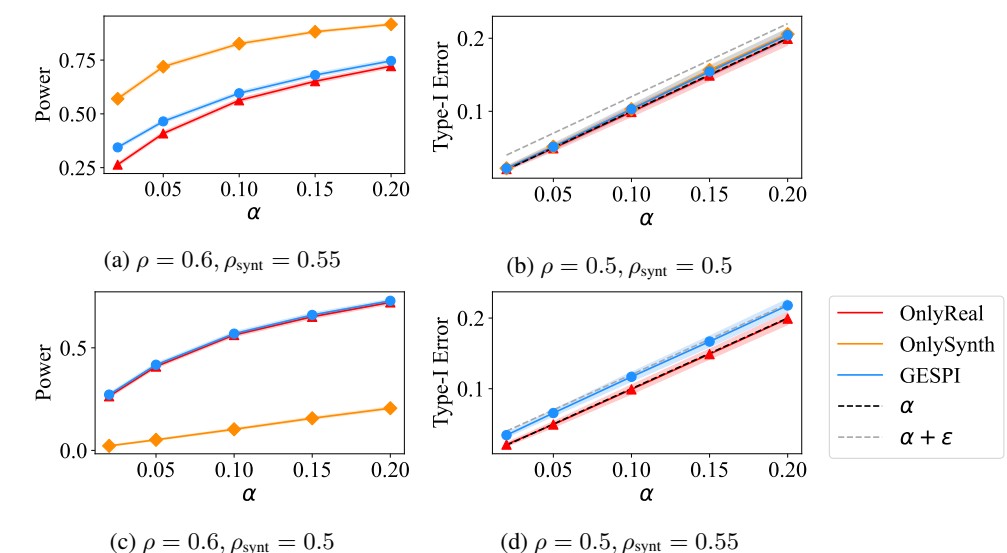

(a) $\rho = 0.6, \rho_{\text{synt}} = 0.55$

(b) $\rho = 0.5, \rho_{\text{synt}} = 0.5$

(c) $\rho = 0.6, \rho_{\text{synt}} = 0.5$

(d) $\rho = 0.5, \rho_{\text{synt}} = 0.55$

Figure 10: **Performance comparison as a function of the target Type I error level** $\alpha$. Hypothesis testing methods across different values of $\rho$ and $\rho_{\text{synt}}$. GESPI applied with $\varepsilon = 2\%$.

## C.2 CONFORMAL RISK CONTROL FOR PROTEIN STRUCTURE PREDICTION

Figure 13 visualizes protein T1078, for which AlphaFold produces a highly accurate predicted structure. Figures 11a and 11b present the resulting prediction sets obtained by `OnlyReal` and `GESPI`, respectively. Panel (1) presents the predicted structure, with residues abstained on marked in red and accepted residues in blue. Both methods obtain risk equal to 0; however, `OnlyReal` conservatively abstains from most residues ($\sim 91\%$), while `GESPI` abstains from significantly fewer ($\sim 48\%$), demonstrating the benefit of leveraging synthetic data. Panel (2) shows the predicted structure (red and blue) aligned with the true structure (gray), illustrating that most of the protein is accurately predicted. Panel (3) highlights a small subset of residues whose predicted structure is inaccurate; both methods abstain. Lastly, panel (4) shows a well-predicted region where `OnlyReal` unnecessarily abstains, while `GESPI` correctly accepts the residues.

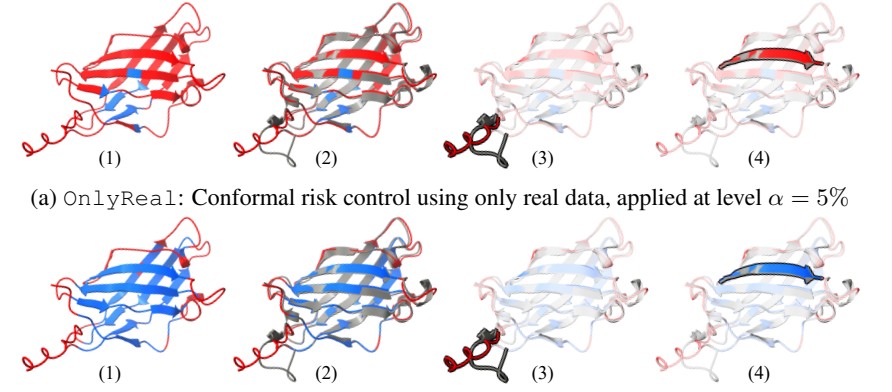

(a) `OnlyReal`: Conformal risk control using only real data, applied at level $\alpha = 5\%$

(b) `GESPI`: our method applying conformal risk control at level $\alpha = 5\%$ and guardrail $\alpha + \varepsilon = 10\%$

Figure 11: **Visualization of protein structure prediction with error rate control.** Panels show protein T1078 predictions with residues abstained on by (a) `OnlyReal` and (b) `GESPI` methods. Red: residues abstained on; Blue: accepted residues. Gray: real experimental structure, aligned with AlphaFold2 predicted structure. Quantitative results {abstention ratio, risk}: `OnlyReal` − {$\approx 91\%, 0\%$}; `GESPI` − {$\approx 48\%, 0\%$}.

Figure 12 complements Figure 3 in the main manuscript by showing the chosen pLDDT thresholds, $\hat{\lambda}$, for $\alpha = 5\%$ and $10\%$. The figure demonstrates that the selected thresholds vary with $\alpha$ and differ across methods: `OnlyReal` conservatively chooses higher thresholds, while our `GESPI` procedure selects lower thresholds, which, as shown in Figure 3, result in risk closer to the nominal $\alpha$ level.

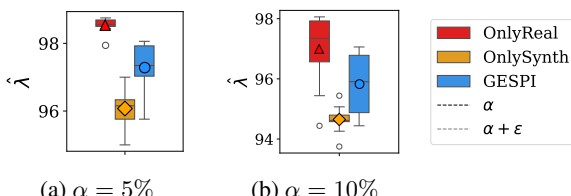

(a) $\alpha = 5\%$        (b) $\alpha = 10\%$

Figure 12: Chosen pLDDT thresholds, $\hat{\lambda}$, for $\alpha = 5\%$ and $10\%$, complementing the results shown in Figure 3.

Figure 13 presents the performance of the conformal risk control methods at target level $\alpha = 15\%$. The results show a similar trend to that in Figure 3. In particular, `OnlyReal` conservatively controls the risk and leads to a higher and more variable abstention rate. `OnlySynth`, which relies on approximate prediction errors, does not provide valid risk control guarantees. In contrast, our proposed method, `GESPI`, achieves risk close to the nominal $\alpha$ level with a lower abstention rate.

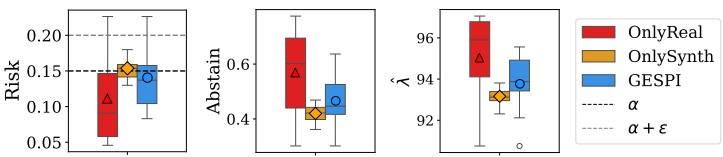

Figure 13: **Performance comparisons for protein structure prediction with error rate control.** Conformal risk control methods applied at target level $\alpha = 15\%$. Left: average risk (fraction of residues with error $> 3$Å). Middle: average abstention rate. Right: selected pLDDT threshold $\hat{\lambda}$.

## C.3 SINGLE AND MULTIPLE HYPOTHESIS TESTING FOR OUTLIER DETECTION

We extend the experiments from Section 4.3 by rerunning them with a different trimming proportion. Given that the true fraction of outliers in the contaminated reference set is generally unknown, we evaluate the performance under a smaller trimming rate, reducing it from $q = 5\%$ (main manuscript) to $q = 2.5\%$, i.e., removing fewer datapoints from the contaminated reference set.

As shown in Figure 14, the results follow a similar trend as in the main manuscript. `OnlySynth` produces error rates both above and below the nominal level, as it does not provide any error rate control guarantees. `OnlyReal` conservatively controls the error metric at the target level, and `GESPI` achieves error rates close to the target while improving power, approaching the performance of the `Oracle`.

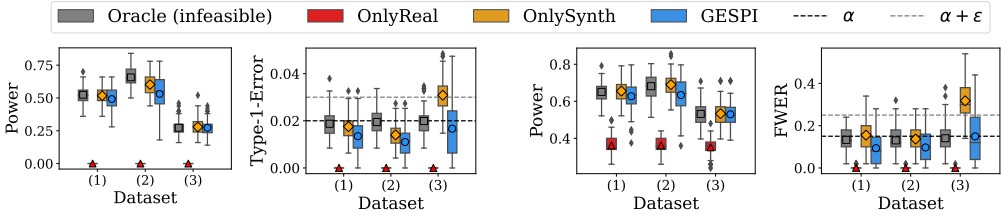

Figure 14: **Performance comparisons for outlier detection**. Evaluated on three datasets: (1) shuttle, (2) credit-card, (3) KDDCup99. Left two panels: single-outlier case ($\alpha = 2\%$, $\varepsilon = 1\%$). Right two panels: multiple-outlier case ($\alpha = 15\%$, $\varepsilon = 10\%$). Same setup as in Figure 5, but with $q = 2.5\%$ trimming proportion.

## C.4 HYPOTHESIS TESTING FOR MECHANISTIC INTERPRETABILITY OF A VISION TRANSFORMER MODEL

**Overview.** In this section, we consider a task related to the mechanistic interpretability of vision transformers. We aim to test whether a specific attention head exhibits a functional role—for example, detecting shapes, animals, or spatial locations. Concretely, we test whether the activation of a given head is stronger when the object it is hypothesized to detect is present in the image compared to when it is absent, which would provide evidence that the head indeed serves this role. As a case study, we examine attention head L22H6 in CLIP ViT-L/14 (Radford et al., 2021), which has been previously reported to strongly respond to images with animals (Gandelsman et al., 2024).

**Datasets.** We use ImageNet images as the real dataset, partitioned into *animals* and *non-animals* classes. For the synthetic dataset, we generate images using FLUX.1 (Labs, 2024), also separated into *animals* and *non-animals*. Details for each dataset are as follows:

- **ImageNet (real)** (Deng et al., 2009): We use the training split, restricted to nine selected classes (listed below); all available training images from these classes are included. Grouping is inherited directly from the ImageNet taxonomy: classes depicting animals are assigned to the *animals* group and the remainder to *non-animals*. The exact class lists, any filtering rules, and per-class counts are provided below.
- **FLUX.1 (synthetic)** (Labs, 2024): For each class, we generate 2,000 images using the `FluxPipeline` from `diffusers` with mixed precision (`float16`) and 50 inference steps. Prompts follow the CLIP-style template "A photo of a {class name}" (Radford et al., 2021), where {class name} is the corresponding ImageNet label. The resulting synthetic images are labeled *animals / non-animals* based on the originating class.
- **Class selection**: For the power experiment (animal vs. non-animal), we selected three animal categories—Labrador retriever (208), English springer spaniel (217), and kuvasz (222)—and three non-animal controls—lighter (626), tennis ball (852), and stinkhorn (994). For the Type I error experiment, both groups were drawn exclusively from non-animal categories. We used gyromitra (993), coral fungus (991), tandem bicycle (444), tennis ball (852), stinkhorn (994), and lighter (626). The numbers in parentheses indicate the corresponding ImageNet class indices.

**Model and architecture.** We use the CLIP configuration of Gandelsman et al. (2024) implemented via OpenCLIP with vision backbone `ViT-L-14` (24 transformer layers, 16 attention heads, patch size 14) and the standard CLIP text transformer (context length 77). We use the checkpoint `'laion2b_s32b_b82k'` of `'ViT-L-14'`, without any fine-tuning or architectural modification. Images are preprocessed using the model's default pipeline (resize and center-crop to $224 \times 224$, CLIP normalization).

**Experimental procedure.** To evaluate whether attention head L22H6 detects animals, we quantify its response to images containing animals versus images without animals. Specifically, we compute a per-image "activation score" that summarizes how strongly each image patch (token) at this head aligns with the animal concept in CLIP's embedding space, with larger scores indicating stronger evidence that the image contains an animal.

We first encode the prompt $t^\star =$ "a photo of an animal" with the CLIP text encoder model $M_{\text{text}}$. Then, for each input image $X$, we use the heads-and-tokens decomposition of the multi-head self attention CLIP vision encoder output. Let $M_{\text{Image}}^{l,h}(X; i)$ denote the direct contribution of token $i$ at layer $l$ and head $h$ of the input $X$. We define a per-head spatial heatmap by projecting each token's contribution onto the text direction, as follows:

$$H_i^{l,h}(X) = \left\langle M_{\text{Image}}^{l,h}(X; i), M_{\text{text}}(t^\star) \right\rangle.$$

Intuitively, $H_i^{l,h}(X)$ measures how strongly token $i$ aligns with the animal text direction. See an example visualized as a spatial heatmap in Figure 15.

We summarize $H_i^{l,h}(X)$ across all tokens $i$ into a scalar "activation" score given by the mean absolute value over tokens:

$$s^{l,h}(X) = \frac{1}{\#\text{tokens}} \sum_{\text{tokens } i} \left| H_i^{l,h}(X) \right|, \tag{8}$$

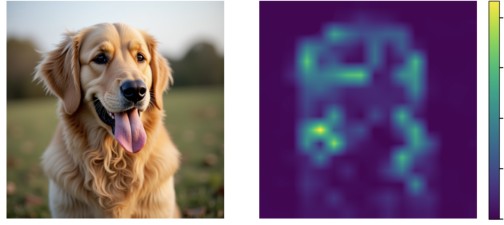

Figure 15: Left: input image $X$ of a dog. Right: spatial heatmap $H^{22,6}(X)$ from layer 6, head 22, obtained by projecting token contributions onto $M_{\text{text}}$("a photo of an animal").

where the number of tokens $i$ is a function of the image resolution and patch size.

**Formulation of the null and alternative.** We compute the activation score $s^{l,h}(X)$ (8) on two disjoint image sets, $\mathcal{D}_{\text{animals}}$ and $\mathcal{D}_{\text{non-animals}}$, and test whether head $(l,h)$ is more active on animals. Let $P^{(l,h)}_{\text{animals}}$ and $P^{(l,h)}_{\text{non-animals}}$ denote the population distributions of activation scores of head $(l,h)$ over animals and non-animals, respectively. Let $\mu^{(l,h)}_{\text{animals}}$ and $\mu^{(l,h)}_{\text{non-animals}}$ denote the expected values/means of these populations. We test the null hypothesis that these two distributions are equal against the one-sided alternative that the mean activation score over the animal distribution is larger:

$$\mathcal{H}_0 : P^{(l,h)}_{\text{animals}} = P^{(l,h)}_{\text{non-animals}} \text{ versus } \mathcal{H}_1 : \mu^{(l,h)}_{\text{animals}} - \mu^{(l,h)}_{\text{non-animals}} > 0.$$

As the test statistic, we use the standardized difference in mean activation scores:

$$\hat{\Delta}^{(l,h)} = \frac{\frac{1}{|\mathcal{D}_{\text{animals}}|}\sum_{X\in\mathcal{D}_{\text{animals}}} s^{l,h}(X) - \frac{1}{|\mathcal{D}_{\text{non-animals}}|}\sum_{X\in\mathcal{D}_{\text{non-animals}}} s^{l,h}(X).}{\sqrt{\hat{\sigma}^2_{\text{animals}}/|\mathcal{D}_{\text{animals}}| + \hat{\sigma}^2_{\text{non-animals}}/|\mathcal{D}_{\text{non-animals}}|)}}$$

where $\hat{\sigma}^2_{\text{animals}}$ and $\hat{\sigma}^2_{\text{non-animals}}$ are the empirical variances of the activation scores of $\mathcal{D}_{\text{animals}}$ and $\mathcal{D}_{\text{non-animals}}$, respectively.

To set the critical value, we use a standard permutation approach, which is guaranteed to control the Type I error under the null hypothesis when the two distributions are equal (see e.g., Lehmann & Romano, 2005b).

**Base statistical test.** We use a permutation test with 10,000 permutations per trial. For each setting (null for Type I error, alternative for power), we perform 100 independent trials. Within each trial, we compute the test statistic on the observed data and on the permuted datasets to obtain a $p$-value. Aggregating over the 100 trials yields our estimates of Type I error and power. We report the mean and standard deviation across trials to quantify variability.

**Results: Evaluating power.** Figure 16 presents the performance of different hypothesis testing methods as a function of the real sample size $n$, for $N = 100$, $\alpha = 10\%$, and $\varepsilon = 5\%$. The rejection rate (power) is well above the target level $\alpha$, indicating that the mean activation score for images containing animals is indeed higher than for non-animals, supporting the conclusion that attention head L22H6 detects animals.

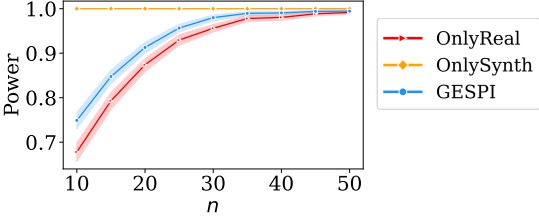

Figure 16: **Performance comparisons as a function of the real sample size $n$.** Hypothesis testing methods applied to animal versus non-animal groups, each containing three classes, at target level $\alpha = 10\%$ and $\varepsilon = 5\%$.

Following that figure, we observe that `GESPI` consistently achieves higher power than the base method `OnlyReal` across all values of $n$, with power increasing as $n$ grows, illustrating the benefit of leveraging additional synthetic data. For sufficiently large $n$ ($n \geq 45$), both methods achieve power close to 1 and obtain comparable performance. At the same time, `OnlySynth`, which relies only on synthetic images, achieves high power due to the large sample size; however, this approach does not provide valid error rate control, as the synthetic data may differ from the real data distribution.

**Results: Evaluating Type I error.** Next, we compare the performance of different hypothesis testing methods under the null hypothesis. To do so, we split the non-animal classes into two disjoint groups and repeat the experiment from Figure 16. Under the null, the activation score distributions of the two groups are expected to be equal. Indeed, Figure 17a shows the histograms for the real data, where the two groups exhibit similar distributions. In contrast, Figure 17b shows the histograms for the synthetic data, where the two groups display noticeable differences, highlighting that the synthetic distribution does not perfectly match the real one.

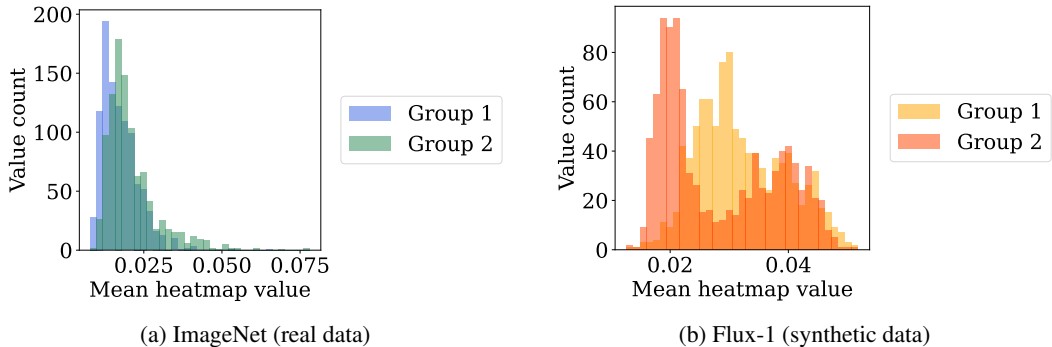

(a) ImageNet (real data)                    (b) Flux-1 (synthetic data)

Figure 17: **Histograms of the per-image activation scores $s^{l,h}(X)$ for two disjoin non-animal groups.** Each histogram is based on 1,000 samples drawn from the corresponding group; (a) shows results for real ImageNet images, and (b) for synthetic images generated with FLUX.1.

Figure 18 shows the Type I error for $n = 10$ and 100; other details are as in Figure 16. Both `OnlyReal` and `GESPI` control the Type I error at the target level $\alpha$. In contrast, `OnlySynth` obtains very high Type I error, illustrating that the synthetic data cannot be naively treated as if it were real.

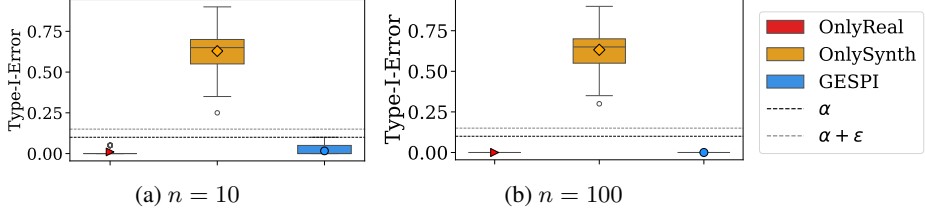

(a) $n = 10$                    (b) $n = 100$

Figure 18: **Performance comparisons under the null hypothesis.** Hypothesis testing methods applied to two disjoint groups of non-animal classes at target level $\alpha = 10\%$ and $\varepsilon = 5\%$; with (a) $n = 10$ and (b) $n = 100$.

## D   EXAMPLES OF STATISTICAL INFERENCE WITH RISK CONTROL

In this section, we provide the details of how a number of classical statistical inference problems can be formulated in the form from (5) required by our paper in order to apply the GESPI methodology. We aim to illustrate that the formulation (5) covers a wide range of problems.

1. *Distribution-free predictive inference*
   Suppose we are given i.i.d. data $(X_1, Y_1), \ldots, (X_n, Y_n)$ and a new input $X_{n+1}$, with the

goal of constructing a prediction set for the unobserved outcome $Y_{n+1}$, while controlling its coverage rate. This corresponds to the following setting, where $\mathcal{B}(\mathcal{Y})$ is the Borel sigma-algebra of $\mathcal{Y}$:

$$\mathcal{Z} = \mathcal{V} = \mathcal{X} \times \mathcal{Y}, \ \mathcal{A} = \{g : \mathcal{X} \to \mathcal{B}(\mathcal{Y}), \text{ measurable}\}, \ \mathcal{T}(P) = P = P_X \times P_{Y|X},$$

$$V = (X_{n+1}, Y_{n+1}) \sim P,$$

$$\mathcal{P} = \Delta(\mathcal{X} \times \mathcal{Y}) = \{\text{ set of all Borel probability distributions on } \mathcal{X} \times \mathcal{Y}\},$$

$$\ell(g, (x, y)) = \mathbb{1}\left\{y \notin g(x)\right\}.$$

Then the condition (5) is equivalent to the following distribution-free marginal coverage guarantee:

$$\mathbb{P}_{(X_1, Y_1), \ldots, (X_n, Y_n), (X_{n+1}, Y_{n+1}) \overset{\text{iid}}{\sim} P}\left\{Y_{n+1} \in \widehat{C}_n(X_{n+1})\right\} \geq 1 - \alpha, \qquad \text{for all distributions } P,$$

where $\widehat{C}_n = \text{Alg}(\mathcal{D}_n)$ denotes the output prediction set function from the algorithm Alg.

As a remark, if we instead set $\mathcal{T}(P) = Q_X \times P_{Y|X}$—assuming a known $Q_X$ or known likelihood ratio $dQ_X/dP_X$—then this setting corresponds to predictive inference under covariate shift.

2. *Hypothesis testing*

   Consider a hypothesis testing problem where one uses a sample $X_1, \ldots, X_n \overset{\text{iid}}{\sim} P$ on $\mathcal{X}$ to test a null hypothesis of the form
   $$\mathcal{H}_0 : P \in \mathcal{P}_0,$$
   for some set of (null) distributions $\mathcal{P}_0$, while controlling the Type I error. This problem can be viewed as a special case of the general target (5) under the following setup:

   $$\mathcal{Z} = \mathcal{X}, \ \mathcal{A} = [0, 1], \ \mathcal{V} = \{0\}, \ \mathcal{T}(P) \equiv \delta_0, \ V = 0, \ \mathcal{P} = \mathcal{P}_0,$$

   $$\ell(w, 0) = w.$$

   This results in the following Type I error/level control condition:

   $$\mathbb{E}_P[\phi] \leq \alpha, \qquad \text{for all distributions } P \in \mathcal{P}_0,$$

   where $\phi = \mathcal{A}(\mathcal{D}_n)$ denotes the output testing procedure from Alg.

3. *Confidence intervals*

   Fix a space $\mathcal{P}$ of distributions and a parameter-functional $\theta : \mathcal{P} \to \mathbb{R}$—e.g., that maps a distribution to its mean or median. Consider constructing a confidence interval for $\theta(P)$ using the sample $X_1, \ldots, X_n \overset{\text{iid}}{\sim} P$. We let

   $$\mathcal{Z} = \mathcal{X}, \ \mathcal{A} = \mathcal{B}(\mathbb{R}), \ \mathcal{V} = \mathbb{R}, \ \mathcal{T}(P) = \delta_{\theta(P)}, \ V = \theta(P)$$

   $$\ell(I, v) = \mathbb{1}\left\{v \notin I\right\},$$

   which corresponds to the coverage guarantee

   $$\mathbb{P}\left\{\theta(P) \in \widehat{C}\right\} \geq 1 - \alpha, \text{ for all } P \in \mathcal{P},$$

   where $\widehat{C} = \text{Alg}(\mathcal{D}_n)$.

4. *Conformal risk control*

   Given sample $(X_1, Y_1), \ldots, (X_n, Y_n) \overset{\text{iid}}{\sim} P$ on $\mathcal{X} \times \mathcal{Y}$ and $X_{n+1} \in \mathcal{X}$, suppose we now aim to construct a map $h : \mathcal{X} \to \mathcal{Y}'$, for some $\mathcal{Y}'$, with risk control, i.e.,

   $$\mathbb{E}\left[\tilde{\ell}(h(X_{n+1}), Y_{n+1})\right] \leq \alpha,$$

   for a loss function $\tilde{\ell} : \mathcal{Y}' \times \mathcal{Y} \to \mathbb{R}^+$, in the distribution-free sense. This corresponds to the following setup:

   $$\mathcal{Z} = \mathcal{V} = \mathcal{X} \times \mathcal{Y}, \ \mathcal{A} = \{h : \mathcal{X} \to \mathcal{Y}'\}, \ \mathcal{T}(P) = P, \ V = (X_{n+1}, Y_{n+1}) \sim P,$$

   $$\mathcal{P} = \Delta(\mathcal{X} \times \mathcal{Y}) = \{\text{ set of all Borel probability distributions on } \mathcal{X} \times \mathcal{Y}\},$$

   $$\ell(h, (x, y)) = \tilde{\ell}(h(x), y).$$

Angelopoulos et al. (2024) introduces a procedure—conformal risk control—that achieves this guarantee.

5. *Simultaneous predictive inference on multiple test points*
   Consider the setting

   $$\mathcal{Z} = \mathcal{X} \times \mathcal{Y}, \ \mathcal{A} = \{g : \mathcal{X} \to \mathcal{B}(\mathcal{Y})\}, \ \mathcal{V} = \mathcal{Z}^m, \ \mathcal{T}(P) = P^m,$$

   $$V = ((X_{n+1}, Y_{n+1}), \ldots, (X_{n+m}, Y_{n+m})) \overset{\text{iid}}{\sim} P$$

   $$\mathcal{P} = \Delta(\mathcal{X} \times \mathcal{Y}) = \{ \text{ set of all Borel probability distributions on } \mathcal{X} \times \mathcal{Y} \},$$

   $$\ell(g, (z_1, \ldots, z_m)) = \mathbb{1}\left\{ \frac{1}{m} \sum_{j=1}^{m} \mathbb{1}\{y_j \in g(x_j)\} < \delta \right\}, \text{ where } z_j = (x_j, y_j) \text{ for each } j \in [m].$$

   This setup leads to the following PAC-coverage guarantee

   $$\mathbb{P}_{(X_1,Y_1),\ldots,(X_{n+m},Y_{n+m})\overset{\text{iid}}{\sim}P}\left\{ \frac{1}{m} \sum_{j=1}^{m} \mathbb{1}\left\{ Y_{n+j} \in \widehat{C}(X_{n+j}) \right\} \geq 1 - \delta \right\} \geq 1-\alpha, \text{ for all } P,$$

   where $\widehat{C} = \text{Alg}(\mathcal{D}_n)$. Methods for achieving this guarantee are discussed in Gazin et al. (2024) and Lee et al. (2024).

| Example | $\mathcal{Z}$ | $\mathcal{A}$ | $\mathcal{V}$ | $\mathcal{T}(P)$ | $\mathcal{P}$ | $\ell$ |
|---|---|---|---|---|---|---|
| Predictive inference | $\mathcal{X} \times \mathcal{Y}$ | $\{g : \mathcal{X} \to \mathcal{B}(\mathcal{Y})\}$ | $\mathcal{X} \times \mathcal{Y}$ | $P$ | $\Delta(\mathcal{X} \times \mathcal{Y})$ | $\mathbb{1}\{y \notin g(x)\}$ |
| Hypothesis testing | $\mathcal{X}$ | $[0, 1]$ | $\{0\}$ | $\delta_0$ | $\mathcal{P}_0$ | $w$ |
| Confidence intervals | $\mathcal{X}$ | $\mathcal{B}(\mathbb{R})$ | $\mathbb{R}$ | $\delta_{\theta(P)}$ | General $\mathcal{P}$ | $\mathbb{1}\{v \notin I\}$ |
| Risk control | $\mathcal{X} \times \mathcal{Y}$ | $\{h : \mathcal{X} \to \mathcal{Y}'\}$ | $\mathcal{X} \times \mathcal{Y}$ | $P$ | $\Delta(\mathcal{X} \times \mathcal{Y})$ | $\tilde{\ell}(h(x), y)$ |
| Simult. pred. inf. | $\mathcal{X} \times \mathcal{Y}$ | $\{g : \mathcal{X} \to \mathcal{B}(\mathcal{Y})\}$ | $(\mathcal{X} \times \mathcal{Y})^m$ | $P^m$ | $\Delta(\mathcal{X} \times \mathcal{Y})$ | $\mathbb{1}\{\frac{1}{m} \sum_{j=1}^{m} \mathbb{1}\{y_j \in g(x_j)\} < \delta\}$ |

Table 2: Summary of examples of our framework.

# E  IMPOSSIBILITY RESULTS

One may consider whether it is possible to achieve a stronger target than what we aim for in the paper. Namely, that for any target and any synthetic data distribution, the risk is controlled at the level $\alpha$ instead of $\alpha + \varepsilon$. Specifically, one may consider the following guarantees:

$$\mathbb{E}_{\mathcal{D}_n\overset{\text{iid}}{\sim}P,\tilde{\mathcal{D}}_N\sim\bar{Q},V\sim\mathcal{T}(P)}\left[ \ell(\widetilde{\text{Alg}}(\mathcal{D}_n, \tilde{\mathcal{D}}_N), V) \right] \leq \alpha, \tag{9}$$

for all $P \in \mathcal{P}$ and any distribution $\bar{Q}$ on $\mathcal{Z}^N$, for all $n, N \in \mathbb{N}$.

However, it turns out that achieving the guarantee (9)—i.e., uniformly controlling the loss for all potential synthetic distributions $\bar{Q}$—is generally not a practical target. Indeed, as we show below, if we can attain the above condition, then we can also do it by ignoring the synthetic data.

**Proposition E.1.** *If an algorithm* $\widetilde{\text{Alg}} : Z^\infty \times Z^\infty \to \mathcal{A}$ *satisfies the condition* (9)*, then it can be dominated by an algorithm that takes only the real datapoints* $\mathcal{D}_n$ *as input, in the following sense: For any arbitrarily small* $\delta > 0$*, there exists an algorithm* $\text{Alg} : Z^\infty \to \mathcal{A}$ *such that*

$$\sup_{\substack{P\in\mathcal{P} \\ \bar{Q}}} \mathbb{E}_{\substack{\mathcal{D}_n\overset{\text{iid}}{\sim}P,\tilde{\mathcal{D}}_N\sim\bar{Q} \\ V\sim\mathcal{T}(P)}}\left[ \ell(\widetilde{\text{Alg}}(\mathcal{D}_n, \tilde{\mathcal{D}}_N), V) \right] \leq \sup_{P\in\mathcal{P}} \mathbb{E}_{\mathcal{D}_n\overset{\text{iid}}{\sim}P,V\sim\mathcal{T}(P)}\left[ \ell(\text{Alg}(\mathcal{D}_n), V) \right] \leq \alpha + \delta.$$

See Section I.1 for the proof. Proposition E.1 states that for any procedure leveraging synthetic data, there exists an algorithm that uses only the real data and achieves the target level with equal or better tightness—potentially leading to higher power, narrower confidence intervals or prediction sets, etc. In other words, if the goal is to uniformly control the loss over all theoretically possible

synthetic distributions, then ignoring the synthetic data may be the optimal choice.[10] However, in practical scenarios where some similarity between the true and synthetic distributions can reasonably be expected, ignoring the synthetic data based on the strong condition (9) would not be the preferred choice.

# F ADDITIONAL THEORETICAL RESULTS FOR GESPI

We first provide a formal statement of the conditions required by our algorithm.

**Condition F.1.** The following conditions hold.

1. The action space $\mathcal{A}$ is endowed with a partial order denoted by $\succeq$. For any pair of elements $a_1, a_2 \in \mathcal{A}$, there uniquely exists an element $a_1 \vee a_2$ in $\mathcal{A}$ such that the following hold:

    (i) $a_1 \vee a_2 \succeq a_1$ and $a_1 \vee a_2 \succeq a_2$. (ii) If $a \succeq a_1$ and $a \succeq a_2$, then $a \succeq a_1 \vee a_2$.

    Similarly, there uniquely exists an element $a_1 \wedge a_2$ in $\mathcal{A}$ such that

    (i) $a_1 \wedge a_2 \preceq a_1$ and $a_1 \vee a_2 \preceq a_2$. (ii) If $a \preceq a_1$ and $a \preceq a_2$, then $a \preceq a_1 \wedge a_2$.

2. For any $a_1, a_2, a_3 \in \mathcal{A}$, $(a_1 \wedge a_2) \vee a_3 = (a_1 \vee a_3) \wedge (a_2 \vee a_3)$.
3. For any $\alpha_1 \leq \alpha_2$ and $z \in \mathcal{Z}^\infty$, $\mathrm{Alg}_{\alpha_1}(z) \preceq \mathrm{Alg}_{\alpha_2}(z)$.
4. The loss function $\ell : \mathcal{A} \times \mathcal{V} \to \mathbb{R}^+$ satisfies the following.
    (a) *boundedness*: There exists $c > 0$ such that $\ell(a, v) \leq c$ for all $a \in \mathcal{A}$ and $v \in \mathcal{V}$.
    (b) *monotonicity*: If $a_1 \preceq a_2$, then $\ell(a_1, v) \leq \ell(a_2, v)$ for any $v \in \mathcal{V}$.

The first two statements in Condition F.1 states that there exists a partial order on the action space with "reasonable properties"—namely, that the minimum and maximum of any two elements are well-defined and that the distributive law holds.

Table 3 illustrates what the partial order means in our examples. The monotonicity of the algorithm and loss can be checked on a case-by-case basis. For instance, conformal prediction sets are nested by construction (Vovk et al., 2005). The requirements in Condition F.1 hold in a variety of statistical inference problems, including the examples in Section 3. See also Table 3 for simple examples.

| Example | $\mathcal{A}$ | $a_1 \preceq a_2$ | $a_1 \vee a_2$ | $a_1 \wedge a_2$ |
|---|---|---|---|---|
| Hypothesis testing | $\{0, 1\}$ | $a_1 \leq a_2$ | $a_1$ OR $a_2$ | $a_1$ AND $a_2$ |
| Confidence intervals | $\mathcal{B}(\mathcal{X})$ | $a_1 \supseteq a_2$ | $a_1 \cap a_2$ | $a_1 \cup a_2$ |
| Predictive inference | $\{g : \mathcal{X} \to \mathcal{B}(\mathcal{Y}), \text{ measurable}\}$ | $a_1(x) \supseteq a_2(x) \, \forall x$ | $x \mapsto a_1(x) \cap a_2(x)$ | $x \mapsto a_1(x) \cup a_2(x)$ |

Table 3: Examples of partially ordered action spaces for different inference problems. Here, $\mathcal{B}(\mathcal{X})$ denotes the Borel sigma-algebra of $\mathcal{X}$.

We now present an extended theoretical result for the GESPI procedure with two-sided guardrail (7). Let us consider scenarios in which the algorithm $\mathrm{Alg}_\alpha$ additionally satisfies a tightness guarantee:

$$\alpha - \delta_n \leq \mathbb{E}_{\mathcal{D}_n \overset{\text{iid}}{\sim} P, V \sim \mathcal{T}(P)} \left[ \ell(\mathrm{Alg}_\alpha(\mathcal{D}_n), V) \right] \leq \alpha, \text{ for all } P \in \mathcal{P} \text{ and } n \in \mathbb{N}. \tag{10}$$

The term $\delta_n$ denotes the tightness level which may depend on the sample size $n$. Examples include:

1. The confidence interval $\bar{X} \pm z_{\alpha/2} \frac{\sigma}{\sqrt{n}}$ for the mean of a normal distribution satisfies (10) with $\delta_n = 0$.
2. The split conformal prediction set attains a tightness level of $\delta_n = \frac{1}{n+1}$ (Vovk et al., 2005).
3. Conformal risk control (Angelopoulos et al., 2024) attains a tightness level of $\delta_n = \frac{2B}{n+1}$, where $B$ denotes an upper bound on the loss.

The following theorem extends the results in Theorem 3.3

---

[10]Note, however, that the dominating procedure Alg for a specific $\widetilde{\mathrm{Alg}}$ in Proposition E.1 may not be explicitly known in certain scenarios. Nevertheless, it is likely that well-known methods in various examples correspond to Alg.

**Theorem F.2.** *Suppose that Condition F.1 holds. Then given $\alpha, \varepsilon > 0$, the algorithm $\widetilde{\text{Alg}}$ defined in (7) satisfies*

$$\text{Alg}_\alpha(\mathcal{D}_n) \preceq \widetilde{\text{Alg}}(\mathcal{D}_n, \tilde{\mathcal{D}}_N) \preceq \text{Alg}_{\alpha+\varepsilon}(\mathcal{D}_n),$$

*deterministically. If the algorithm $\text{Alg}_\alpha$ satisfies (10) for $\alpha$ and $\alpha + \varepsilon$, then*

$$\alpha - \min\{\delta_n, c\tau + \delta_{N+n} + c\cdot\tilde{\text{d}}_{\ell,\text{Alg}}(P,Q)\} \leq \mathbb{E}\left[\ell(\widetilde{\text{Alg}}(\mathcal{D}_n, \tilde{\mathcal{D}}_N), V)\right] \leq \alpha + \min\{\varepsilon, c\tau + c\cdot\tilde{\text{d}}_{\ell,\text{Alg}}(P,Q)\}$$

*for all $P, Q \in \mathcal{P}$, where $\tilde{\text{d}}_{\ell,\text{Alg}}(P,Q) = \text{d}_{\text{TV}}(\tilde{P}_{\ell,\text{Alg}}(P,Q), \tilde{P}_{\ell,\text{Alg}}(Q,Q))$, and $\tilde{P}_{\ell,\text{Alg}}(P,Q)$ denotes the distribution of $\ell(\widetilde{\text{Alg}}(\mathcal{D}_n, \tilde{\mathcal{D}}_N), V)$ under $\mathcal{D}_n \overset{\text{iid}}{\sim} P, \tilde{\mathcal{D}}_N \overset{\text{iid}}{\sim} Q$ and $\mathcal{V} \sim \mathcal{T}(P)$. The term $\tau$ is defined as*

$$\tau := 1 - \mathbb{P}_{\mathcal{D}_n \cup \tilde{\mathcal{D}}_N \overset{\text{iid}}{\sim} Q}\left\{\text{Alg}_\alpha(\mathcal{D}_n) \preceq \text{Alg}_\alpha(\mathcal{D}_n \cup \tilde{\mathcal{D}}_N) \preceq \text{Alg}_{\alpha+\varepsilon}(\mathcal{D}_n)\right\}.$$

The first claim of Theorem F.2 simply restates the result of Theorem 3.3. The key strength of this theorem lies in its generality. When it is particularized to specific examples, and sharpened by leveraging additional structure—such as exchangeability in conformal prediction—it can lead to more interpretable and tighter guarantees.

The general bounds in the theorem imply that the risk lies between $\alpha - c\tau - \delta_{N+n}$ and $\alpha + c\tau$ when $P = Q$, i.e., under "perfect" synthetic datapoints. The term $c\tau$, which appears in addition to the total variation distance, represents the potential bias introduced by the guardrail procedures. In other words, there is a tradeoff in using two-sided guardrails—between the bias in risk introduced by the guardrails in the ideal scenario of $P = Q$, and the range of worst-case scenarios controlled by them.

*Remark* F.3. The lower-guardrail procedure $\text{Alg}_\alpha(\mathcal{D}_n)$ can be replaced with other choices, e.g., $\text{Alg}_{\alpha-\varepsilon'}(\mathcal{D}_n)$ for some $\varepsilon' > 0$. However, we consider $\text{Alg}_\alpha(\mathcal{D}_n)$ as the standard choice, since in most relevant settings, synthetic data is desired because the original procedure $\text{Alg}_\alpha(\mathcal{D}_n)$ is not satisfactory. Thus, one typically wants the new procedure to have at least the power or at most the width of the standard method, respectively, in the worst case.

*Remark* F.4. The first claim of Theorem F.2 holds generally for any procedure of the form

$$\widetilde{\text{Alg}}(\mathcal{D}_n, \tilde{\mathcal{D}}_N) = (\text{Alg}'(\mathcal{D}_n; \tilde{\mathcal{D}}_N) \wedge \text{Alg}_{\alpha+\varepsilon}(\mathcal{D}_n)) \vee \text{Alg}_\alpha(\mathcal{D}_n),$$

where $\text{Alg}'$ is any algorithm that takes the real and synthetic data as inputs. In other words, the guardrail bounds still hold regardless of how we use the real and synthetic data—e.g., data-dependent choices, even in the case of double dipping—and regardless of what joint distribution $\tilde{\mathcal{D}}_N$ follows and what guarantee $\text{Alg}'$ satisfies, etc.

## G    EXTENSION: MULTIPLE TESTING

Consider a problem in which one is given a sample $X_1, \ldots, X_n \overset{\text{iid}}{\sim} P$ on $\mathcal{X}$, and the goal is to test the following $m \geq 1$ hypotheses:

$$\mathcal{H}_{0,j} : P \in \mathcal{P}_j, \qquad j \in [m],$$

where $\mathcal{P}_1, \ldots, \mathcal{P}_m \subset \mathcal{P}_{\text{all}}$ are sets of of distributions belonging to a prespecified set $\mathcal{P}_{\text{all}}$ of distributions of interest. Consider controlling the family-wise error rate (FWER), or more generally, the $k$-FWER (Lehmann & Romano, 2005a):

$$\mathbb{P}\left\{\sum_{j=1}^m \mathbb{1}\{\mathcal{H}_{0,j} \text{ is true but rejected}\} > k\right\} \leq \alpha,$$

for some predetermined $k > 0$. This is equivalent to requiring that condition (5) holds under the following setup, for all $J \subset [m]$ (except for the case $J = \emptyset$):

$$\mathcal{Z} = \mathcal{X}, \ \mathcal{A} = \{0, 1\}^m, \ \mathcal{V} = \{0\}, \ \mathcal{T}(P) \equiv \delta_0, \ V = 0,$$

$$\mathcal{P} = \mathcal{P}_J := \bigcap_{j \in J} \mathcal{P}_j \cap \bigcap_{j \notin J} (\mathcal{P}_{\text{all}} \backslash \mathcal{P}_j),$$

$$\ell_J((a_1, \ldots, a_m), 0) = \mathbb{1} \left\{ \sum_{j \in J} a_j > k \right\}.$$

Specifically, this corresponds to the following problem of simultaneously controlling multiple risks:

$$\mathcal{R}_\gamma(\text{Alg}, P) = \mathbb{E}_{\mathcal{D}_n \overset{\text{iid}}{\sim} P, V \sim \mathcal{T}(P)} [\ell_\gamma(\text{Alg}(\mathcal{D}_n), V)] \leq \alpha, \text{ for all } P \in \mathcal{P}_\gamma \text{ and } n \in \mathbb{N}, \ \forall \gamma \in \Gamma$$

for a set $\Gamma$ of indices $\gamma$, each corresponding to a different distribution space $\mathcal{P}_\gamma$ and loss function $\ell_\gamma$.

For the above multiple testing problem with FWER control, Condition F.1 is satisfied under the following ordering, and with all $\ell_\gamma$s:

$$(a_1, a_2, \ldots, a_m) \preceq (b_1, b_2, \ldots, b_m) \iff a_j \leq b_j \ \forall j \in [m].$$

Therefore, for the synthetic-powered procedure defined in (7), the first claim of Theorem F.2 holds, showing that the guardrails also function properly in this setting.

# H  APPLICATIONS OF GESPI

In this section, we provide more detailed discussions of different applications of GESPI, along with more tailored theoretical results. For simplicity, we restrict our discussion to the GESPI procedure with a one-sided guardrail. Throughout this section, we use the notations $\mathcal{B}(\mathcal{X})$ and $\Delta(\mathcal{X})$, whose definitions follow those in Section D.

## H.1  CONFORMAL PREDICTION

Consider a predictive inference problem where we have samples

$$\mathcal{D}_n = ((X_1, Y_1), \ldots, (X_n, Y_n)) \overset{\text{iid}}{\sim} P = P_X \times P_{Y|X},$$

$$\tilde{\mathcal{D}}_N = ((\tilde{X}_1, \tilde{Y}_1), \ldots, (\tilde{X}_N, \tilde{Y}_N)) \overset{\text{iid}}{\sim} Q = Q_X \times Q_{Y|X},$$

and the goal is, given a new input $X_{n+1} \sim P_X$, to construct a prediction set for $Y_{n+1}$, where $Y_{n+1} \mid X_{n+1} \sim P_{Y|X}$. For this problem, split conformal prediction (Vovk et al., 2005; Papadopoulos et al., 2002) provides the following algorithm:

$\text{Alg}_\alpha : (\mathcal{X} \times \mathcal{Y})^\infty \to \mathcal{A} = \{g : \mathcal{X} \to \mathcal{B}(\mathcal{Y}), \text{ measurable}\},$

$\text{Alg}_\alpha((x_i, y_i)_{i \in [n]}) = \left( x \mapsto \left\{ y \in \mathcal{Y} : s(x, y) \leq \hat{q}_\alpha((x_i, y_i)_{i \in [n]}) \right\} \right), \text{ where}$

$\hat{q}_\alpha((x_1, y_1), \ldots, (x_n, y_n)) = ( \text{ the } \lceil (1 - \alpha)(n + 1) \rceil \text{-th smallest element among } s(x_1, y_1), \ldots, s(x_n, y_n) ).$

Here, $s : \mathcal{X} \times \mathcal{Y} \to \mathbb{R}$ is a prespecified nonconformity score function. This procedure attains the following marginal coverage guarantee:

$$\mathbb{E}_{\mathcal{D}_n \overset{\text{iid}}{\sim} P, (X_{n+1}, Y_{n+1}) \sim P} [\ell(\text{Alg}_\alpha(\mathcal{D}_n), (X_{n+1}, Y_{n+1}))] = \mathbb{P}\left\{ Y_{n+1} \notin \widehat{C}_n(X_{n+1}) \right\} \leq \alpha, \text{ for all } P \in \mathcal{P},$$

where $\ell(g, (x, y)) = \mathbb{1}\{y \notin g(x)\}, \widehat{C}_n = \text{Alg}(\mathcal{D}_n)$, and $\mathcal{P} = \Delta(\mathcal{X} \times \mathcal{Y})$.

Now the GESPI procedure for conformal prediction is constructed as

$$\widetilde{\text{Alg}}(\mathcal{D}_n, \tilde{\mathcal{D}}_N) = \text{Alg}_\alpha(\mathcal{D}_n \cup \tilde{\mathcal{D}}_N) \cup \text{Alg}_{\alpha+\varepsilon}(\mathcal{D}_n) = \tilde{C}_{n,N}, \text{ where}$$

$$\tilde{C}_{n,N}(x) = \left\{ y \in \mathcal{Y} : s(x, y) \leq \max\{\hat{q}_{\alpha+\varepsilon}(\mathcal{D}_n), \hat{q}_\alpha(\mathcal{D}_n \cup \tilde{\mathcal{D}}_N)\} \right\}. \tag{11}$$

This extends the method of Bashari et al. (2025a) by using the real samples $\mathcal{D}_n$ not only as a guardrail, but also as part of the main synthetic-boosted procedure. Suppose we predefine another level $\delta \in (0, 1)$ and then set $\varepsilon$ as $\varepsilon = r_\delta/(n+1) - \alpha$, where

$$r_\delta = \min \left\{ r : \sum_{k=1}^{\lceil (1-\alpha)(N+n+1) \rceil} \frac{\binom{k-1}{r-1} \cdot \binom{N+n-k}{n-r}}{\binom{N+n}{n}} \geq 1 - \delta \right\}.$$

Then we have the following guarantee.

**Theorem H.1.** *Let $P_S$ and $Q_S$ denote the distributions of the score $s(Z)$ when $Z \sim P$ and $Z \sim Q$, respectively, and suppose that both $P_S$ and $Q_S$ are continuous. Then the procedure $\tilde{C}_{n,N} = \widetilde{\mathrm{Alg}}(\mathcal{D}_n, \tilde{\mathcal{D}}_N)$ in (11) satisfies*

$$\mathbb{P}\left\{ Y_{n+1} \in \tilde{C}_{n,N}(X_{n+1}) \right\} \geq 1 - \alpha - \min\{\varepsilon, d_{P,Q}\},$$

*where $d_{P,Q} = \frac{1}{n+1} \sum_{r=1}^{n+1} \mathrm{d}_{\mathrm{TV}}(P_{(r)}^{n+1}, Q_{(r)}^{n+1})$, and $P_{(r)}^{n+1}$ and $Q_{(r)}^{n+1}$ denote the distributions of the $r$-th order statistic from $n + 1$ i.i.d. draws from $P_S$ and $Q_S$, respectively.*

*Furthermore, if the scores are all distinct almost surely, then*

$$\mathbb{P}\left\{ Y_{n+1} \in \tilde{C}_{n,N}(X_{n+1}) \right\} \leq 1 - \alpha + \frac{1}{N+n+1} + d_{P,Q} + \mathrm{d}_{\mathrm{TV}}(P_{(t_{\alpha+\varepsilon})}^n, Q_{(t_{\alpha+\varepsilon})}^n) + \delta,$$

*where $t_{\alpha+\varepsilon} = \lceil (1 - \alpha - \varepsilon)(n+1) \rceil$.*

## H.2 CONFORMAL RISK CONTROL

Consider the setting in the previous section H.1, and suppose now that we instead aim for risk control, i.e., we construct a map $h : \mathcal{X} \to \mathcal{Y}'$ with the guarantee

$$\mathbb{E}_{(X_1,Y_1),\ldots,(X_n,Y_n),(X_{n+1},Y_{n+1}) \overset{\mathrm{iid}}{\sim} P} \left[ \tilde{\ell}(h(X_{n+1}), Y_{n+1}) \right] \leq \alpha, \qquad \text{for all distributions } P,$$

for some loss function $\tilde{\ell} : \mathcal{Y}' \times \mathcal{Y} \to [0, B]$. For this problem, Angelopoulos et al. (2024) introduces the following algorithm:

$$\mathrm{Alg}_\alpha : (\mathcal{X} \times \mathcal{Y})^\infty \to \mathcal{H}_\Lambda = \{h_\lambda : \lambda \in \Lambda\} \subset \{g : \mathcal{X} \to \mathcal{B}(\mathcal{Y}), \text{ measurable}\},$$

$$\mathrm{Alg}_\alpha((x_i, y_i)_{i \in [n]}) = h_{\hat{\lambda}_\alpha((x_i,y_i)_{i \in [n]})},$$

where

$$\hat{\lambda}_\alpha((x_i, y_i)_{i \in [n]}) = \inf \left\{ \lambda \in \Lambda : \frac{\sum_{i=1}^n \tilde{\ell}(h_\lambda(X_i), Y_i) + B}{n + 1} \leq \alpha \right\}.$$

Here, $\mathcal{H}_\Lambda$ is a set of measurable functions parameterized by $\lambda \in \Lambda \subset \mathbb{R}$, and the loss $\tilde{\ell}$ and the set $\mathcal{H}_\Lambda$ are chosen so that the function $\lambda \mapsto \tilde{\ell}(h_\lambda(x), y)$ is monotone increasing for any $(x, y) \in \mathcal{X} \times \mathcal{Y}$.

Note that for the set $\mathcal{H}_\Lambda$, the minimum operation $\wedge$ satisfies

$$\tilde{\ell}(h_{\lambda_1 \wedge \lambda_2}(x), y) \leq \min\{\tilde{\ell}(h_{\lambda_1}(x), y), \tilde{\ell}(h_{\lambda_2}(x), y)\}, \text{ for any } \lambda_1, \lambda_2 \in \Lambda \text{ and } x \in X, y \in \mathcal{Y}.$$

Therefore, the GESPI risk control procedure is given as

$$\widetilde{\mathrm{Alg}}(\mathcal{D}_n, \tilde{\mathcal{D}}_N) = h_{\tilde{\lambda}}, \text{ where } \tilde{\lambda} = \hat{\lambda}_\alpha(\mathcal{D}_n \cup \tilde{\mathcal{D}}_N) \wedge \hat{\lambda}_{\alpha+\varepsilon}(\mathcal{D}_n).$$

## H.3 TEST FOR THE MEDIAN

Consider a testing problem where we use real-valued samples $\mathcal{D}_n = (X_1, \ldots, X_n) \overset{\mathrm{iid}}{\sim} P$ to test

$$\mathcal{H}_0 : Q_{1/2}(P) \leq 0.$$

We consider a procedure $\mathrm{Alg}_\alpha : \mathbb{R}^n \to \{0, 1\}$, defined as

$$\mathrm{Alg}_\alpha((x_1, \ldots, x_n)) = \mathbb{1}\left\{ \sum_{i=1}^n \mathbb{1}\{x_i > 0\} > \hat{k}_{n,\alpha} \right\}, \text{ where } \hat{k}_{n,\alpha} = Q_{1-\alpha}\left( \mathrm{Binom}\left( n, \frac{1}{2} \right) \right).$$

$$(12)$$

**Proposition H.2.** *The algorithm* $\text{Alg}_\alpha$ *in* (12) *satisfies*

$$\mathbb{E}_{\mathcal{D}_n \overset{iid}{\sim} P}\left[\text{Alg}_\alpha(\mathcal{D}_n)\right] \leq \alpha, \ \text{for all } P \in \mathcal{P}_{\mathcal{H}_0} = \{ \text{ all distributions on } \mathbb{R} \text{ with non-positive median } \}.$$

Now, given a synthetic data $\tilde{\mathcal{D}}_N = (\tilde{X}_1, \ldots, \tilde{X}_N)$, the GESPI procedure is given as

$$\widetilde{\text{Alg}}(\mathcal{D}_n, \tilde{\mathcal{D}}_N) = \mathbb{1}\left\{ \sum_{i=1}^n \mathbb{1}\left\{X_i > 0\right\} > \hat{k}_{n,\alpha+\varepsilon} \text{ and } \sum_{i=1}^n \mathbb{1}\left\{X_i > 0\right\} + \sum_{j=1}^N \mathbb{1}\left\{\tilde{X}_j > 0\right\} > \hat{k}_{N+n,\alpha} \right\}, \tag{13}$$

and the following results hold.

**Theorem H.3.** *Let* $p = \mathbb{P}_{X \sim P}\{X > 0\}$ *and* $q = \mathbb{P}_{X \sim Q}\{X > 0\}$. *Then the testing procedure* $\widetilde{\text{Alg}}(\mathcal{D}_n, \tilde{\mathcal{D}}_N)$ *in* (13) *satisfies*

$$\mathbb{E}_{\mathcal{D}_n \overset{iid}{\sim} P, \tilde{\mathcal{D}}_N \overset{iid}{\sim} Q}\left[\widetilde{\text{Alg}}(\mathcal{D}_n, \tilde{\mathcal{D}}_N)\right] \leq \alpha + \min\{\varepsilon, d_{P,Q}\}, \ \text{where } d_{P,Q} = \sqrt{\frac{n}{2q(1-q)}} \cdot |p - q|,$$

*for all* $P \in \mathcal{P}_{\mathcal{H}_0}^\varepsilon = \left\{ \text{ distributions } \bar{P} \text{ on } \mathbb{R} : \mathbb{P}_{X \sim \bar{P}}\{X > 0\} \leq \frac{1}{2} - \sqrt{\frac{1}{2n}\varepsilon} \right\}$ *and for any distribution* $Q$.

# I  PROOF OF THEOREMS

## I.1  PROOF OF PROPOSITION E.1

First observe that

$$\sup_{\substack{P \in \mathcal{P} \\ \bar{Q}}} \mathbb{E}_{\mathcal{D}_n \overset{iid}{\sim} P, \tilde{\mathcal{D}}_N \sim \bar{Q}, V \sim \mathcal{T}(P)}\left[\ell(\widetilde{\text{Alg}}(\mathcal{D}_n, \tilde{\mathcal{D}}_N), V)\right]$$

$$= \sup_{\substack{P \in \mathcal{P} \\ \bar{Q}}} \mathbb{E}_{\tilde{\mathcal{D}}_N \sim \bar{Q}}\left[\mathbb{E}_{\mathcal{D}_n \overset{iid}{\sim} P, V \sim \mathcal{T}(P)}\left[\ell(\widetilde{\text{Alg}}(\mathcal{D}_n, \tilde{\mathcal{D}}_N), V) \mid \tilde{D}_N\right]\right]$$

$$= \sup_{\bar{Q}} \mathbb{E}_{\tilde{\mathcal{D}}_N \sim \bar{Q}}\left[\sup_{P \in \mathcal{P}} \mathbb{E}_{\mathcal{D}_n \overset{iid}{\sim} P, V \sim \mathcal{T}(P)}\left[\ell(\widetilde{\text{Alg}}(\mathcal{D}_n, \tilde{\mathcal{D}}_N), V) \mid \tilde{D}_N\right]\right] \leq \sup_{d_N \in \mathcal{Z}^N} h(d_N),$$

where $h(d_N) = \sup_{P \in \mathcal{P}} \mathbb{E}_{\mathcal{D}_n \overset{iid}{\sim} P, V \sim \mathcal{T}(P)}\left[\ell(\widetilde{\text{Alg}}(\mathcal{D}_n, d_N), V)\right]$. Next, for any $\delta > 0$, there exists $d_N^\delta$ such that $h(d_N^\delta) \geq \sup_{d_N \in \mathcal{Z}^N} h(d_N) - \delta$. Since condition (9) holds for any distribution $\bar{Q}$ on $\mathcal{Z}^N$, it also holds for $\bar{Q} = \delta_{d_N^\delta}$ (the point mass on $d_N^\delta$), which implies $h(d_N^\delta) \leq \alpha$. Therefore, defining $\text{Alg}(\mathcal{D}_n) = \widetilde{\text{Alg}}(\mathcal{D}_n, d_N^\delta)$, we have that

$$\sup_{\substack{P \in \mathcal{P} \\ \bar{Q}}} \mathbb{E}_{\mathcal{D}_n \overset{iid}{\sim} P, \tilde{\mathcal{D}}_N \sim \bar{Q}, V \sim \mathcal{T}(P)}\left[\ell(\widetilde{\text{Alg}}(\mathcal{D}_n, \tilde{\mathcal{D}}_N), V)\right] \leq h(d_N^\delta) + \delta$$

$$= \sup_{P \in \mathcal{P}} \mathbb{E}_{\mathcal{D}_n \overset{iid}{\sim} P, V \sim \mathcal{T}(P)}\left[\ell(\text{Alg}(\mathcal{D}_n), V)\right] + \delta \leq \alpha + \delta.$$

This finishes the proof.

## I.2  PROOF OF THEOREM 3.2

First observe that since

$$\ell(a_1 \wedge a_2, v) \leq \min\{\ell(a_1, v), \ell(a_2, v)\}, \text{ for any } a_1, a_2 \in \mathcal{A} \text{ and } v \in \mathcal{V}$$

holds by Condition F.1, we have

$$\mathbb{E}_{\mathcal{D}_n \overset{iid}{\sim} P, \tilde{\mathcal{D}}_N \overset{iid}{\sim} Q, V \sim \mathcal{T}(P)}\left[\ell(\widetilde{\text{Alg}}(\mathcal{D}_n, \tilde{\mathcal{D}}_N), V)\right] \leq \mathbb{E}_{\mathcal{D}_n \overset{iid}{\sim} P, \tilde{\mathcal{D}}_N \overset{iid}{\sim} Q, V \sim \mathcal{T}(P)}\left[\ell(\text{Alg}_{\alpha+\varepsilon}(\mathcal{D}_n), V)\right]$$

$$= \mathbb{E}_{\mathcal{D}_n \overset{iid}{\sim} P, V \sim \mathcal{T}(P)}\left[\ell(\text{Alg}_{\alpha+\varepsilon}(\mathcal{D}_n), V)\right] \leq \alpha + \varepsilon,$$

where the last inequality holds by the assumption that $\mathrm{Alg}_\alpha$ satisfies the guarantee (5). Similarly,

$$\mathbb{E}_{\mathcal{D}_n \overset{\mathrm{iid}}{\sim} P, \tilde{\mathcal{D}}_N \overset{\mathrm{iid}}{\sim} Q, V \sim \mathcal{T}(P)} \left[ \ell(\widetilde{\mathrm{Alg}}(\mathcal{D}_n, \tilde{\mathcal{D}}_N), V) \right] \leq \mathbb{E}_{\mathcal{D}_n \overset{\mathrm{iid}}{\sim} P, \tilde{\mathcal{D}}_N \overset{\mathrm{iid}}{\sim} Q, V \sim \mathcal{T}(P)} \left[ \ell(\mathrm{Alg}_\alpha(\mathcal{D}_n \cup \tilde{\mathcal{D}}_N), V) \right]$$

$$\leq \mathbb{E}_{\mathcal{D}_n \overset{\mathrm{iid}}{\sim} Q, \tilde{\mathcal{D}}_N \overset{\mathrm{iid}}{\sim} Q, V \sim \mathcal{T}(Q)} \left[ \ell(\mathrm{Alg}_\alpha(\mathcal{D}_n \cup \tilde{\mathcal{D}}_N), V) \right] + c \cdot \mathrm{d}_{\ell, \mathrm{Alg}}(P, Q) \leq \alpha + c \cdot \mathrm{d}_{\ell, \mathrm{Alg}}(P, Q).$$

Combining the two results above, we obtain the desired inequality.

### I.3  PROOF OF THEOREM 3.3

The proof of Theorem 3.3 is covered in Section I.4, which provides the proof for the extended statement, Theorem F.2.

### I.4  PROOF OF THEOREM F.2

First, the relation $\mathrm{Alg}_\alpha(\mathcal{D}_n) \preceq \widetilde{\mathrm{Alg}}(\mathcal{D}_n, \tilde{\mathcal{D}}_N)$ follows directly from the first part of Condition F.1. Next, by the second and third condition in Condition F.1, we have

$$\widetilde{\mathrm{Alg}}(\mathcal{D}_n, \tilde{\mathcal{D}}_N) = (\mathrm{Alg}_\alpha(\mathcal{D}_n \cup \tilde{\mathcal{D}}_N) \wedge \mathrm{Alg}_{\alpha+\varepsilon}(\mathcal{D}_n)) \vee \mathrm{Alg}_\alpha(\mathcal{D}_n)$$

$$= (\mathrm{Alg}_\alpha(\mathcal{D}_n \cup \tilde{\mathcal{D}}_N) \vee \mathrm{Alg}_\alpha(\mathcal{D}_n)) \wedge (\mathrm{Alg}_{\alpha+\varepsilon}(\mathcal{D}_n) \vee \mathrm{Alg}_\alpha(\mathcal{D}_n))$$

$$= (\mathrm{Alg}_\alpha(\mathcal{D}_n \cup \tilde{\mathcal{D}}_N) \vee \mathrm{Alg}_\alpha(\mathcal{D}_n)) \wedge \mathrm{Alg}_{\alpha+\varepsilon}(\mathcal{D}_n),$$

which implies $\widetilde{\mathrm{Alg}}(\mathcal{D}_n, \tilde{\mathcal{D}}_N) \preceq \mathrm{Alg}_{\alpha+\varepsilon}(\mathcal{D}_n)$.

Now we prove the second claim. Observe that the first claim implies

$$\alpha - \delta_n \leq \mathbb{E}\left[ \ell(\widetilde{\mathrm{Alg}}(\mathcal{D}_n, \tilde{\mathcal{D}}_N), V) \right] \leq \alpha + \varepsilon,$$

and thus it is sufficient to show that

$$\alpha - c\tau - \delta_{N+n} \leq \mathbb{E}\left[ \ell(\widetilde{\mathrm{Alg}}(\mathcal{D}_n, \tilde{\mathcal{D}}_N), V) \right] \leq \alpha + c\tau,$$

when $P = Q$. Then the final claim follows from the inequality

$$\left| \mathbb{E}_{\substack{\mathcal{D}_n \overset{\mathrm{iid}}{\sim} P, \tilde{\mathcal{D}}_N \sim \bar{Q} \\ V \sim \mathcal{T}(P)}} \left[ \ell(\widetilde{\mathrm{Alg}}(\mathcal{D}_n, \tilde{\mathcal{D}}_N), V) \right] - \mathbb{E}_{\substack{\mathcal{D}_n \overset{\mathrm{iid}}{\sim} Q, \tilde{\mathcal{D}}_N \sim \bar{Q} \\ V \sim \mathcal{T}(P)}} \left[ \ell(\widetilde{\mathrm{Alg}}(\mathcal{D}_n, \tilde{\mathcal{D}}_N), V) \right] \right| \leq c \cdot \tilde{\mathrm{d}}_{\ell, \mathrm{Alg}}(P, Q),$$

which holds by the boundedness of $\ell$ and the properties of the total variation distance.

Let $E$ be the event that $\mathrm{Alg}_\alpha(\mathcal{D}_n) \preceq \mathrm{Alg}_\alpha(\mathcal{D}_n \cup \tilde{\mathcal{D}}_N) \preceq \mathrm{Alg}_{\alpha+\varepsilon}(\mathcal{D}_n)$ holds. Observe that $\widetilde{\mathrm{Alg}}(\mathcal{D}_n, \tilde{\mathcal{D}}_N) = \mathrm{Alg}_\alpha(\mathcal{D}_n \cup \tilde{\mathcal{D}}_N)$ under the event $E$. Therefore, under $P = Q$,

$$\mathbb{E}\left[ \ell(\widetilde{\mathrm{Alg}}(\mathcal{D}_n, \tilde{\mathcal{D}}_N), V) \right] = \mathbb{E}\left[ \ell(\widetilde{\mathrm{Alg}}(\mathcal{D}_n, \tilde{\mathcal{D}}_N), V) \cdot \mathbb{1}\{E\} \right] + \mathbb{E}\left[ \ell(\widetilde{\mathrm{Alg}}(\mathcal{D}_n, \tilde{\mathcal{D}}_N), V) \cdot \mathbb{1}\{E^c\} \right]$$

$$\leq \mathbb{E}\left[ \ell(\mathrm{Alg}_\alpha(\mathcal{D}_n \cup \tilde{\mathcal{D}}_N), V) \right] + c \cdot \mathbb{P}\{E^c\} \leq \alpha + c\tau.$$

Moreover, we have

$$\mathbb{E}\left[ \ell(\widetilde{\mathrm{Alg}}(\mathcal{D}_n, \tilde{\mathcal{D}}_N), V) \right] \geq \mathbb{E}\left[ \ell(\widetilde{\mathrm{Alg}}(\mathcal{D}_n, \tilde{\mathcal{D}}_N), V) \cdot \mathbb{1}\{E\} \right]$$

$$= \mathbb{E}\left[ \ell(\widetilde{\mathrm{Alg}}(\mathcal{D}_n, \tilde{\mathcal{D}}_N), V) \right] - \mathbb{E}\left[ \ell(\widetilde{\mathrm{Alg}}(\mathcal{D}_n, \tilde{\mathcal{D}}_N), V) \cdot \mathbb{1}\{E^c\} \right] \geq \alpha - \delta_{N+n} - c\tau,$$

which completes the proof.

### I.5  PROOF OF THEOREM H.1

Let $S_i = s(X_i, Y_i)$ for $i \in [n+1]$—including the test point—and let $\tilde{S}_j = s(\tilde{X}_i, \tilde{Y}_i)$ for $j \in [N]$. Since the distributions $P_S$ and $Q_S$ are continuous, there are no ties among the scores. Define $R$ as the rank of the test score $S_{n+1}$ among the scores $\{S_1, \ldots, S_{n+1}\}$:

$$R = \sum_{i=1}^{n+1} \mathbb{1}\{S_i \leq S_{n+1}\}.$$

Then we have $S_{n+1} = S_{(R)}$, where $S_{(r)}$ denotes the $r$-th order statistics of $S_1, \ldots, S_{n+1}$. Moreover, by the exchangeability of $S_1, \ldots, S_{n+1}$, we have $R \sim \text{Unif}([n+1])$[11]

Now, we observe

$$\mathbb{P}\left\{Y_{n+1} \in \tilde{C}_{n,N}(X_{n+1})\right\} = \mathbb{P}\left\{S_{n+1} \leq \max\{\hat{q}_{\alpha+\varepsilon}(\mathcal{D}_n), \hat{q}_{\alpha}(\mathcal{D}_n \cup \tilde{\mathcal{D}}_N)\}\right\} \geq \mathbb{P}\left\{S_{n+1} \leq \hat{q}_{\alpha}(\mathcal{D}_n \cup \tilde{\mathcal{D}}_N)\right\}$$

$$= \mathbb{P}\left\{\text{(number of } S_i\text{'s and } \tilde{S}_j\text{'s smaller than } S_{n+1}) \leq \lceil(1-\alpha)(N+n+1)\rceil\right\}$$

$$= \mathbb{P}\left\{\sum_{i=1}^{n} \mathbb{1}\left\{S_i < S_{(R)}\right\} + \sum_{j=1}^{N} \mathbb{1}\left\{\tilde{S}_j < S_{(R)}\right\} \leq \lceil(1-\alpha)(N+n+1)\rceil\right\}$$

$$= \mathbb{P}\left\{R - 1 + \sum_{j=1}^{N} \mathbb{1}\left\{\tilde{S}_j \leq S_{(R)}\right\} \leq \lceil(1-\alpha)(N+n+1)\rceil\right\}$$

$$= \mathbb{E}\left[\mathbb{P}\left\{R - 1 + \sum_{j=1}^{N} \mathbb{1}\left\{\tilde{S}_j \leq S_{(R)}\right\} \leq \lceil(1-\alpha)(N+n+1)\rceil \;\middle|\; R\right\}\right]$$

$$= \frac{1}{n+1}\sum_{r=1}^{n+1} \mathbb{P}\left\{r - 1 + \sum_{j=1}^{N} \mathbb{1}\left\{\tilde{S}_j \leq S_{(r)}\right\} \leq \lceil(1-\alpha)(N+n+1)\rceil \;\middle|\; R = r\right\}$$

$$= \frac{1}{n+1}\sum_{r=1}^{n+1} \mathbb{P}\left\{r - 1 + \sum_{j=1}^{N} \mathbb{1}\left\{\tilde{S}_j \leq S_{(r)}\right\} \leq \lceil(1-\alpha)(N+n+1)\rceil\right\},$$

where the last equality holds since $R$ is independent of $S_{(1)}, \ldots, S_{(n+1)}$,[12] as well as the synthetic scores $(\tilde{S}_j)_{j \in [N]}$. Now denote the event inside the probability by $E_r$, and observe that the probability inside the summation is taken with respect to

$$\tilde{S}_1, \ldots, \tilde{S}_N \overset{\text{iid}}{\sim} Q_S, S_{(r)} \sim P_{(r)}^{n+1}, \text{ with } (\tilde{S}_j)_{j \in [N]} \perp\!\!\!\perp S_{(r)}$$

Therefore, putting everything together, we have

$$\mathbb{P}_{(X_i,Y_i)_{i \in [n+1]} \overset{\text{iid}}{\sim} P, (\tilde{X}_j, \tilde{Y}_j)_{j \in [N]} \overset{\text{iid}}{\sim} Q}\left\{Y_{n+1} \in \tilde{C}_{n,N}(X_{n+1})\right\} \geq \frac{1}{n+1}\sum_{r=1}^{n+1} \mathbb{P}_{\tilde{S}_1,\ldots,\tilde{S}_N \overset{\text{iid}}{\sim} Q_S, S_{(r)} \sim P_{(r)}^{n+1}}\{E_r\}$$

$$\geq \frac{1}{n+1}\sum_{r=1}^{n+1}\left[\mathbb{P}_{\tilde{S}_1,\ldots,\tilde{S}_N \overset{\text{iid}}{\sim} Q_S, S_{(r)} \sim P_{(r)}^{n+1}}\{E_r\} - d_{\text{TV}}(P_{(r)}^{n+1}, Q_{(r)}^{n+1})\right]$$

$$= \frac{1}{n+1}\sum_{r=1}^{n+1} \mathbb{P}_{\tilde{S}_1,\ldots,\tilde{S}_N \overset{\text{iid}}{\sim} Q_S, S_{(r)} \sim Q_{(r)}^{n+1}}\{E_r\} - d_{P,Q}$$

$$= \mathbb{P}_{(X_i,Y_i)_{i \in [n+1]} \overset{\text{iid}}{\sim} Q, (\tilde{X}_j, \tilde{Y}_j)_{j \in [N]} \overset{\text{iid}}{\sim} Q}\left\{S_{n+1} \leq \hat{q}_{\alpha}(\mathcal{D}_n \cup \tilde{\mathcal{D}}_N)\right\} - d_{P,Q}$$

$$\geq 1 - \alpha - d_{P,Q},$$

where the last equality holds by the same steps as derived previously—the distribution $P$ is simply replaced by $Q$. This proves the first claim, since $\mathbb{P}\left\{Y_{n+1} \in \tilde{C}_{n,N}(X_{n+1})\right\} \geq \mathbb{P}\left\{S_{n+1} \leq \hat{q}_{\alpha+\varepsilon}(\mathcal{D}_n)\right\} \geq 1 - \alpha - \varepsilon$.

We now prove the second claim. Observe that

$$\mathbb{P}\left\{Y_{n+1} \in \tilde{C}_{n,N}(X_{n+1})\right\} = \mathbb{P}\left\{S_{n+1} \leq \max\{\hat{q}_{\alpha+\varepsilon}(\mathcal{D}_n), \hat{q}_{\alpha}(\mathcal{D}_n \cup \tilde{\mathcal{D}}_N)\}\right\}$$

$$\leq \mathbb{P}\left\{S_{n+1} \leq \hat{q}_{\alpha}(\mathcal{D}_n \cup \tilde{\mathcal{D}}_N)\right\} + \mathbb{P}\left\{\hat{q}_{\alpha+\varepsilon}(\mathcal{D}_n) > \hat{q}_{\alpha}(\mathcal{D}_n \cup \tilde{\mathcal{D}}_N)\right\}.$$

---

[11]For a finite set $A$, we write $\text{Unif}(A)$ to denote the uniform distribution on $A$.

[12]This follows from $S_{n+1} \mid (S_{(1)}, \ldots, S_{(n+1)}) \sim \frac{1}{n+1}\sum_{i=1}^{n+1} \delta_{S_{(i)}}$, which implies $R \mid (S_{(1)}, \ldots, S_{(n+1)}) \sim \text{Unif}([n+1])$.

applying arguments similar to the one used previously, we have

$$\mathbb{P}_{(X_i,Y_i)_{i\in[n+1]}\overset{\text{iid}}{\sim}P,(\tilde{X}_j,\tilde{Y}_j)_{j\in[N]}\overset{\text{iid}}{\sim}Q}\left\{S_{n+1} \le \hat{q}_\alpha(\mathcal{D}_n \cup \tilde{\mathcal{D}}_N)\right\}$$

$$\le \mathbb{P}_{(X_i,Y_i)_{i\in[n+1]}\overset{\text{iid}}{\sim}Q,(\tilde{X}_j,\tilde{Y}_j)_{j\in[N]}\overset{\text{iid}}{\sim}Q}\left\{S_{n+1} \le \hat{q}_\alpha(\mathcal{D}_n \cup \tilde{\mathcal{D}}_N)\right\} + d_{P,Q}$$

$$\le 1 - \alpha + \frac{1}{N+n+1} + d_{P,Q},$$

where the second inequality applies the result of Vovk et al. (2005).

Next, let $r_{\alpha+\varepsilon} = \lceil(1-\alpha-\varepsilon)(n+1)\rceil$, and let $\bar{S}_{(1)}, \ldots, \bar{S}_{(n)}$ be the order statistics of $S_1, \ldots, S_n$—note that these differ from $S_{(1)}, \ldots, S_{(n+1)}$. We let $\bar{S}_{(n+1)} = +\infty$. We then compute

$$\mathbb{P}\left\{\hat{q}_{\alpha+\varepsilon}(\mathcal{D}_n) > \hat{q}_\alpha(\mathcal{D}_n \cup \tilde{\mathcal{D}}_N)\right\} = \mathbb{P}\left\{\bar{S}_{(r_{\alpha+\varepsilon})} > \hat{q}_\alpha(\mathcal{D}_n \cup \tilde{\mathcal{D}}_N)\right\}$$

$$= \mathbb{P}\left\{\sum_{i=1}^n \mathbb{1}\left\{S_i < \bar{S}_{(r_{\alpha+\varepsilon})}\right\} + 1 + \sum_{j=1}^N \mathbb{1}\left\{\tilde{S}_j \le \bar{S}_{(r_{\alpha+\varepsilon})}\right\} \ge \lceil(1-\alpha)(N+n+1))\rceil\right\}$$

$$= \mathbb{P}\left\{r_{\alpha+\varepsilon} + \sum_{j=1}^N \mathbb{1}\left\{\tilde{S}_j \le \bar{S}_{(r_{\alpha+\varepsilon})}\right\} \ge \lceil(1-\alpha)(N+n+1))\rceil\right\}.$$

Since the event inside the probability depends on $(S_i)_{i\in[n]}$ only through $\bar{S}_{(r_{\alpha+\varepsilon})}$, we have

$$\mathbb{P}_{(X_i,Y_i)_{i\in[n]}\overset{\text{iid}}{\sim}P,(\tilde{X}_j,\tilde{Y}_j)_{j\in[N]}\overset{\text{iid}}{\sim}Q}\left\{\hat{q}_{\alpha+\varepsilon}(\mathcal{D}_n) > \hat{q}_\alpha(\mathcal{D}_n \cup \tilde{\mathcal{D}}_N)\right\}$$

$$\le \mathbb{P}_{(X_i,Y_i)_{i\in[n]}\overset{\text{iid}}{\sim}Q,(\tilde{X}_j,\tilde{Y}_j)_{j\in[N]}\overset{\text{iid}}{\sim}Q}\left\{\hat{q}_{\alpha+\varepsilon}(\mathcal{D}_n) > \hat{q}_\alpha(\mathcal{D}_n \cup \tilde{\mathcal{D}}_N)\right\} + \text{d}_{\text{TV}}(P_{(r_{\alpha+\varepsilon})}^n, Q_{(r_{\alpha+\varepsilon})}^n).$$

Now, let $R_r$ be the rank of $\bar{S}_{(r)}$ in $(S_i)_{i\in[n]} \cup (\tilde{S}_j)_{j\in[N]}$. Then under $\mathcal{D}_n \cup \tilde{\mathcal{D}}_N \overset{\text{iid}}{\sim} Q$,

$$\mathbb{P}\left\{\bar{S}_{(r_{\alpha+\varepsilon})} > \hat{q}_\alpha(\mathcal{D}_n \cup \tilde{\mathcal{D}}_N)\right\} = \mathbb{P}\left\{R_{r_{\alpha+\varepsilon}} > \lceil(1-\alpha)(N+n+1)\rceil\right\}$$

$$= \sum_{k=\lceil(1-\alpha)(N+n+1)\rceil+1}^{N+n} \mathbb{P}\left\{R_{r_{\alpha+\varepsilon}} = k\right\} = \sum_{k=\lceil(1-\alpha)(N+n+1)\rceil+1}^{N+n} \frac{\binom{k-1}{r_{\alpha+\varepsilon}-1} \cdot \binom{N+n-k}{n-r_{\alpha+\varepsilon}}}{\binom{N+n}{n}},$$

which is bounded by $\delta$ under the assumed choice of $\varepsilon$. This completes the proof.

### I.6 PROOF OF PROPOSITION H.2

Fix any $P \in \mathcal{P}$, and let $p = \mathbb{P}\{X > 0\}$. Since $P \in \mathcal{P}$, we have $p < 1/2$. Now let $\tilde{k}_{n,\alpha} = Q_{1-\alpha}(\text{Binom}(n,p))$, and observe that $\tilde{k}_{n,\alpha} \le \hat{k}_{n,\alpha}$. Then

$$\mathbb{E}_{\mathcal{D}_n\overset{\text{iid}}{\sim}P}[\text{Alg}_\alpha(\mathcal{D}_n)] = \mathbb{P}\left\{\sum_{i=1}^n \mathbb{1}\{X_i > 0\} > \hat{k}_{n,\alpha}\right\} \le \mathbb{P}\left\{\sum_{i=1}^n \mathbb{1}\{X_i > 0\} > \tilde{k}_{n,\alpha}\right\} \le \alpha,$$

since $\sum_{i=1}^n \mathbb{1}\{X_i > 0\} \sim \text{Binom}(n,p)$.

### I.7 PROOF OF THEOREM H.3

Fix any $P \in \mathcal{P}_{\mathcal{H}_0}^\varepsilon$ and $Q$. We first have

$$\mathbb{E}_{\mathcal{D}_n\overset{\text{iid}}{\sim}P,\tilde{\mathcal{D}}_N\overset{\text{iid}}{\sim}Q}\left[\widetilde{\text{Alg}}(\mathcal{D}_n, \tilde{\mathcal{D}}_N)\right] \le \alpha + \varepsilon$$

by Theorem 3.2. The claim holds trivially if $Q \notin \mathcal{P}_{\mathcal{H}_0}$, since

$$d_{P,Q} = \sqrt{\frac{n}{2q(1-q)}} \cdot (q-p) \ge \sqrt{2n} \cdot \left(\frac{1}{2} - \left(\frac{1}{2} - \sqrt{\frac{1}{2n}}\varepsilon\right)\right) = \varepsilon.$$

in that case, since $q(1-q) \leq 1/4$. Thus, we now focus on the case $Q \in \mathcal{P}_{\mathcal{H}_0}$.

Since the bound $\alpha + \varepsilon$ follows directly from the guardrail component $\text{Alg}(\mathcal{D}_n)$, it suffices to show that $\mathbb{E}\left[\widetilde{\text{Alg}}(\mathcal{D}_n, \tilde{\mathcal{D}}_N)\right] \leq \alpha + d_{P,Q}$ holds. Let $W_n = \sum_{i=1}^{n} \mathbb{1}\{X_i\} > 0$ and $\widetilde{W}_N = \sum_{j=1}^{N} \mathbb{1}\{\tilde{X}_j > 0\}$. Since the rejection event depends only on $W_n \sim \text{Binom}(n,p)$ and $\widetilde{W}_N \sim \text{Binom}(N,q)$,

$$\mathbb{E}_{\mathcal{D}_n \overset{\text{iid}}{\sim} P, \tilde{\mathcal{D}}_N \overset{\text{iid}}{\sim} Q}\left[\widetilde{\text{Alg}}(\mathcal{D}_n, \tilde{\mathcal{D}}_N)\right] \leq \mathbb{E}_{W_n \sim \text{Binom}(n,p), \widetilde{W}_N \sim \text{Binom}(N,q)}\left[W_n + \widetilde{W}_N > \hat{k}_{N+n,\alpha}\right]$$

$$\leq \mathbb{E}_{W_n \sim \text{Binom}(n,q), \widetilde{W}_N \sim \text{Binom}(N,q)}\left[W_n + \widetilde{W}_N > \hat{k}_{N+n,\alpha}\right] + d_{\text{TV}}(\text{Binom}(n,p), \text{Binom}(n,q))$$

$$\leq \alpha + d_{\text{TV}}(\text{Binom}(n,p), \text{Binom}(n,q)),$$

where the second inequality holds since we are assuming $Q \in \mathcal{P}_{\mathcal{H}_0}$. Then observe

$$d_{\text{TV}}(\text{Binom}(n,p), \text{Binom}(n,q)) \leq \sqrt{\frac{1}{2}d_{\text{TV}}(\text{Binom}(n,p), \text{Binom}(n,q))}$$

$$= \sqrt{\frac{n}{2}\left(p\log\frac{p}{q} + (1-p)\log\frac{1-p}{1-q}\right)} \leq \sqrt{\frac{n}{2}\left(p \cdot \left(\frac{p}{q} - 1\right) + (1-p) \cdot \left(\frac{1-p}{1-q} - 1\right)\right)}$$

$$= \sqrt{\frac{n}{2q(1-q)}} \cdot |p - q|.$$

This completes the proof.