# OpenReview forum: "Statistical Inference Leveraging Synthetic Data with Distribution-Free Guarantees"
_ICLR.cc/2026/Conference — ICLR 2026 Conference Withdrawn Submission_

### Official Review · Reviewer_yUKm · 2025-10-26

**Soundness:** 2
**Presentation:** 3
**Contribution:** 1
**Rating:** 0
**Confidence:** 4

**Summary:**

The paper proposes to combine synthetic and real data for statistical inference problems while controlling the risk to a given level. Potential applications include conformal prediction, one-sided hypothesis testing, outlier detection and multiple hypothesis testing. The method, called GESPI, targets a confidence level $\alpha$, and includes an error term $\epsilon$. GESPI uses the underlying statistical inference method on the real data with confidence levels $\alpha$ and $\alpha + \epsilon$, and on the combination of real and synthetic data with confidence level $\alpha$. These results from the three runs are combined in such a way that the result is valid for confidence level $\alpha + \epsilon$ regardless of the distribution of the synthetic data, and has higher power than the test only using real data with confidence level $\alpha$ if the synthetic data is similar enough to the real data. The paper compares GESPI against two baselines: only using real data, and only using synthetic data, both with confidence level $\alpha$. Across many applications, the results show that GESPI has higher power than either baseline.

**Strengths:**

The paper tackles an important problem: data are scarce in many important applications, which limits the effectiveness of reliable statistical procedures. Two of the applications presented in the paper, protein folding prediction and LLM performance comparison, are very good examples of these types of problems. The writing of the paper is generally clear, and the proposed method is original.

**Weaknesses:**

The critical weakness of GESPI is the error term $\epsilon$: GESPI only has the distribution-free guarantee advertised in the abstract and introduction for a confidence level $\alpha + \epsilon$. This means that GESPI should be compared to using only the real data at confidence level $\alpha + \epsilon$. However, the formulation of GESPI shows that it would lose this comparison. This can be seen by looking at the result of GESPI in each of Sections 3.1-3.3: in each case, GESPI cannot return a better result than the underlying inference method with confidence level $\alpha + \epsilon$. For example, for conformal prediction in Section 3.1, GESPI cannot return a smaller confidence set than the underlying procedure only using real data at $\alpha + \epsilon$, which is pointed out in footnote 2. The paper even proves this more generally in Proposition E.1. It is true that the actual type 1 error rate of GESPI may be lower than $\alpha + \epsilon$, but this cannot be known without making assumptions on the synthetic data and giving up the distribution-free guarantee.

This baselines chosen for the experiments conceal this weakness. The baseline using only real data is calibrated to $\alpha$, not $\alpha + \epsilon$, so it is not surprising that it has less power. The proper baseline to compare with would be calibrated to $\alpha + \epsilon$. In addition, the baseline using only synthetic data doesn't make sense. It should use the real data in addition to the synthetic data.

Minor points:
- Figure 2: it is not clear what is the difference between panels (1) through (4) before page 8. This should be in the caption.
- Line 413: shouldn't the null be $p \leq 0.5$?

**Questions:**

No additional questions.

---

> ### Author Response · Authors · 2025-11-20
> **Rebuttal by Authors (1/2)**
>
> We suspect there has been a big misunderstanding. To clarify, **there is nothing to conceal, and there is no ``win or lose'' situation when compared to the worst-case guardrail** (using the real data at level $\alpha + \varepsilon$). GESPI is not competing with the worst-case guardrail---which is instead an integral part of our method. In fact, Theorem 3.3 formally shows that the GESPI procedure is deterministically sandwiched between the base and the worst-case guardrail. Deterministically, GESPI’s power is always capped by the worst-case guardrail and can never surpass it (in terms of prediction set size, statistical power, etc.). This is the reason we initially did not include the guardrail baseline.
>
> Therefore, we respectfully disagree with the reviewer’s assertion that ``...the formulation of GESPI shows that it would lose this comparison,'' as well as with their interpretation of $\varepsilon$ as a weakness.
>
> We can certainly compare GESPI with the method that uses real data at error level $\alpha + \varepsilon$, and we include all experiments with this guardrail method in the supporting document in this [link](https://tinyurl.com/rebuttal-exp). However, that method **does not leverage the synthetic data, and therefore still suffers from the low real sample size, even if it is run at a higher level**. Our experiments show that for this reason, this method leads to a large variability (even though the average risk is controlled at  $\alpha + \varepsilon$, the variance can be much higher than that of our GESPI method). This shows that this method is not uniformly preferable to our method.
>
> To further illustrate this point, we added a new experiment on class-conditional coverage in image classification. As shown in Figure S1 in this [link](https://tinyurl.com/rebuttal-exp), for the *Magpie* class, the variance of the guardrail method is extremely large due to the small sample size, leading to realized coverage as low as 60\% even though the target level is 95\%. In contrast, GESPI achieves coverage close to 95\% with far lower variability, making it clearly preferable to the guardrail approach in this case.
>
> Perhaps the source of confusion stems from the change of perspective we present in this work. We thank the reviewer for the opportunity to clarify this point. Our work introduces a paradigm shift in statistical inference, enabling the safe use of additional synthetic data. Consider a practitioner with limited real data, abundant synthetic data, and a risk-controlling algorithm.
>
> The key question is: **how can one perform statistical inference in this setting, with the goal of controlling the risk at level $\alpha$**?
> 1. The first option is to completely ignore the synth. data, even when it may be informative, and rely solely on the limited real data. This approach can lead to overly conservative inference, often prompting practitioners to increase the risk level from $\alpha$ to $\alpha'$ to obtain more informative results---thereby achieving a risk level of $\alpha'$ instead of the desired $\alpha$.
> 2. The second option is to blindly trust the synth. data as if they were real, pooling them together with the limited real data. However, this approach lacks any form of distribution-free risk-control guarantee and results in uncontrolled risk.
>
> Both alternatives are insufficient. **GESPI offers a principled hybrid solution that combines the strengths of both**: providing the practitioner with the ability to control the worst-case outcome while also benefiting from high-quality synthetic data, even when their quality is unknown a-priori.
>
> Importantly, this benefit would not be possible without $\varepsilon$, showing that it is not a weakness but rather a strength: $\varepsilon$ has a clear statistical interpretation, providing the practitioner with a new dimension to safely leverage synthetic data.
> > Regarding the comment on ``GESPI only has the distribution-free guarantee'':
>
> While it is true that under distribution-free guarantees we can only ensure risk control at level $\alpha + \varepsilon$, **in practice, when the synthetic data are of high quality, GESPI achieves a risk level close to the nominal level $\alpha$**. This is supported both theoretically (Theorems 3.2 and F.2) and empirically. Importantly, GESPI does not require prior knowledge of the quality of the synth. data---it adapts automatically to it.
>
> The new experiment on class-conditional coverage with ImageNet (Figure S1 in this [link](https://tinyurl.com/rebuttal-exp)) further illustrates this point. As shown, when the synth. data are of low quality (e.g., the Siberian husky class), GESPI performs on par with the guardrail. As the quality of the synth. data improves, GESPI effectively leverages it, achieving coverage close to the nominal $1 - \alpha$ level, with reduced variance.
>
> ### Given this clarification, we hope the reviewer will reassess our work, and we are open to discussion about the new perspectives that our work offers.

---

> > ### Comment · Reviewer_yUKm · 2025-11-22
> >
> > Thank you for the response and the non-shortened link. After reading your responses and the linked document, I'm still not convinced that GESPI is useful over just using the real data.
> >
> > I agree with you that neither ignoring synthetic data or blindly trusting it is a good solution. This means that any proposed solution should demonstrate some benefit compared to the naive solutions. This is where the $\epsilon$ of GESPI becomes a weakness: the distribution-free risk control guarantee of GESPI is for a level of $\alpha + \epsilon$. This means that GESPI should be compared with other methods with a distribution-free guarantee of $\alpha + \epsilon$. And, as you have said, GESPI cannot have higher power than just using real data with the same guarantee.
> >
> > If a practitioner is willing to give up the distribution-free guarantee and instead rely on empirically demonstrated risk, then GESPI may be beneficial over just using real data. But this needs to be demonstrated, and the current experiments do not include a proper comparison, since they do not calibrate each method to the same empirical risk. And in the cases where GESPI and Guardrail have very similar empirical coverage or risk in your new results, they also have very similar power. So based on these cases, it appears GESPI does not bring clear benefit when controlling empirical risk either.

---

> > > ### Author Response · Authors · 2025-11-25
> > > **Response to Reviewer yUKm (1/2)**
> > >
> > > We value the reviewer’s opinion and thank them for the comments. If we understand the reviewer correctly, the suggestion is to tune $\alpha'$ such that GESPI and OnlyReal attain the same empirical risk and then compare them. Of course, as we are certain the reviewer knows, this is not feasible in practice, since it requires knowing the empirical risk.
> > >
> > > Still, for the sake of this discussion, we include such a plot here:  https://anonymous.4open.science/api/repo/gespi-rebuttal-9E46/file/exp_rebuttal_2.pdf?v=6bf9b83c
> > > Figure S18 presents the class-conditional coverage experiment in image classification, comparing GESPI and OnlyReal when they achieve the same mean empirical risk. Figure S19 shows the same setting but for the one-sided version of GESPI. Since both GESPI and OnlyReal use the same machine-learning model and the same prediction-set construction base algorithm, and obtain the same mean empirical risk, the meaningful quantity to examine is the variance.
> > >
> > > The variance reflects the scenario in which a practitioner would run the procedure multiple times with different realizations of the data. Accordingly, each point in the figures represents the holdout-data-conditional risk observed from a single run using one data realization.
> > >
> > > We observe that GESPI obtains lower variance than OnlyReal. In addition, GESPI attains coverage levels that OnlyReal cannot due to the small sample size. **The achieved variance reduction demonstrates another benefit of GESPI**, which goes beyond the theoretical guarantees on risk control and power. When synthetic data are of high quality, GESPI yields smaller variance and therefore higher stability. This variance reduction is important in practice, as a practitioner has access to only a single dataset, and lower variance ensures that the holdout-data-conditional risk is more likely to be close to the target level. Therefore, controlling the empirical risk in expectation alone does not fully capture the procedure’s practical performance.
> > >
> > > All in all, this experiment provides evidence that GESPI is useful over using only the limited real data, addressing the reviewer’s comment that ``I'm still not convinced that GESPI is useful over just using the real data.'' Using only real data suffers from small sample size, whereas GESPI **can effectively increase the effective sample size when synthetic data are of high quality**.
> > >
> > > Note that the synthetic data used in this experiment are not perfect, but are of high quality. To illustrate this, Figure S20 presents the performance of OnlySynth and Synth+Real across varying miscoverage levels $\alpha$. These methods are intended only to illustrate the quality of the synthetic data and do not provide any distribution-free risk control guarantees. As shown,
> > > **both methods consistently achieve coverage slightly below the target $1-\alpha$**, indicating that the synthetic data alone does not suffice here. Nevertheless, as evidenced by Figures S18 and S19, the synthetic data are still of high quality, as GESPI achieves lower variance than OnlyReal.
> > >
> > > Moreover, this phenomenon of variance reduction occurs across our experiments in the supporting document, where GESPI shows lower variance than Guardrail (e.g., Figures S1, S6-S10). Although the two methods do not attain exactly the same empirical risk in these experiments---which is not feasible to know in practice---they offer the same distribution-free risk control guarantees. Thus, the reduced variance highlights a practical advantage of using GESPI.
> > >
> > > We agree with the reviewer that this is an insightful comparison and will include it in the paper. At the same time, as mentioned at the beginning, **this experiment is not feasible in practice, since a practitioner cannot tune $\alpha'$ without access to the true (unknown) empirical risk**. Therefore, we view this analysis as strictly pedagogical: it is unclear how such a comparison would translate into a usable procedure.
> > >
> > > As the reviewer mentioned ``I agree with you that neither ignoring synthetic data or blindly trusting it is a good solution.'' We are on the same page. GESPI provides a framework that combines the strengths of both approaches: benefiting from high-quality synthetic data to improve sample efficiency, while maintaining distribution-free guarantees regardless of the quality of the synthetic data.

---

> > > > ### Author Response · Authors · 2025-11-25
> > > > **Response to Reviewer yUKm (2/2)**
> > > >
> > > > > ``This means that GESPI should be compared with other methods with a distribution-free guarantee of $\alpha+\epsilon$.''
> > > >
> > > > We already include comparison to the guardrail baseline, which uses only the real data at level $\alpha+\varepsilon$. There are no other methods to apply, so it is unclear to us what additional methods the reviewer is referring to.
> > > >
> > > > > ``If a practitioner is willing to give up the distribution-free guarantee and instead rely on empirically demonstrated risk''
> > > >
> > > > The practitioner does **not** need to give up the distribution-free guarantee. On the contrary, GESPI provides a distribution-free guarantee at level $\alpha+\varepsilon$, as formally stated in Theorems 3.2 and F.2. **In addition to this guarantee**, GESPI enables the user to safely benefit from synthetic data: when the synthetic data are of high quality, the observed risk can be close to $\alpha$ while increasing the sample efficiency, without needing to know this a priori.
> > > >
> > > > It appears that the reviewer’s comment focuses only on the worst-case scenario and overlooks the possibility that synthetic data may be of high quality. This is a valuable perspective, as ensuring worst-case validity is paramount; we agree that it is important to have it. We will emphasize this perspective even more in our paper.
> > > >
> > > > At the same time, **GESPI is specifically designed to address the full spectrum of data qualities from worst-case to best-case**. When the synthetic data are of low quality, GESPI still guarantees distribution-free risk control, and when the synthetic data are informative, it can effectively benefit from a larger sample size.
> > > >
> > > > We thank the reviewer for their engagement and the opportunity to clarify the advantages of GESPI over using only limited real data. We believe this discussion better addresses the reviewer’s concern, showing that GESPI is not merely a worst-case guardrail, but can provide benefits by effectively leveraging high-quality synthetic data.

---

> ### Author Response · Authors · 2025-11-20
> **Rebuttal by Authors (2/2)**
>
> > Regrading the baselines
>
> As explained above, we are not competing with the worst-case guardrail; on the contrary, it defines the maximum power that GESPI can achieve. That said, power alone does not tell the full story and does not make the guardrail necessarily preferable over GESPI, as illustrated above. When the synthetic data are of high quality, we expect GESPI to achieve a risk level close to the nominal $\alpha$, while its power remains capped by the guardrail.
>
> For completeness, we include the worst-case guardrail as a baseline in all experiments in the supporting document (see [link](https://tinyurl.com/rebuttal-exp)), alongside the pooled real and synth baseline (`Synth+Real`). The `OnlySynth` baseline is included solely to illustrate the quality of the synthetic data; it is not intended as a competing method and, like `Synth+Real`, does not provide any risk-control guarantees.
> > Figure 2: it is not clear what is the difference between panels (1) through (4) before page 8. This should be in the caption.
>
> Thank you for the comment. We will include a more detailed explanation in the caption.
> > Line 413: shouldn't the null be $p\leq 0.5$?
>
> Thanks for pointing this out. We agree that it will be more clear to formulate the null hypothesis as $p\leq 0.5$, in order to explicitly include in our hypothesis testing model the scenario where the new model is worse than the old one. However, using $p = 0.5$ leads to the exact same algorithms and conclusions since the binomial testing problem has a monotone likelihood ratio property; thus, the tests for the two hypotheses coincide. Thank you for pointing this out. We will update this in the paper.

---

### Official Review · Reviewer_3sMY · 2025-10-27

**Soundness:** 3
**Presentation:** 4
**Contribution:** 3
**Rating:** 6
**Confidence:** 3

**Summary:**

The authors propose GESPI, a general procedure for combining real and synthetic data to perform statistical inference (e.g., predictive inference, hypothesis testing) while controlling risk at a specified level. The procedure relies upon two “guardrails” to ensure validity: a risk level-$\alpha$ algorithm, which ensures the algorithm resulting from pooling the real and synthetic data is at least as useful as when using the real data only; and a risk level-($\alpha + \epsilon$) algorithm, which ensures that the risk of the pooled algorithm is controlled at level $\alpha + \epsilon$.

The authors effectively demonstrated that GESPI can be applied to many inference problems, including conformal inference and risk control, one-sided hypothesis testing, multiple hypothesis testing, and outlier detection (Sections 3.1-3.3). They subsequently generalize their framework to general statistical inference problems (Section 3.4).

In Section 3.4.1, the authors prove that the GESPI procedure controls the risk at level $\alpha + \epsilon$, and they give an upper bound on the risk of the algorithm that depends on the (unknown) closeness of the real data distribution to the synthetic data distribution. Moreover, they prove that when the risk level-$\alpha$ algorithm is used as a lower guardrail, the GESPI algorithm will always be at least as useful as this algorithm.

Finally, in Section 4 (and Appendix Section C), the authors demonstrate the varying degrees to which GESPI improves upon the real-only and the synthetic-only algorithms.  Their applications include conformal risk control for protein structure prediction (with AlphaFold2), hypothesis testing for the win-rate of two LLMs on the AIME25 dataset, conformal outlier detection on benchmark datasets, and hypothesis testing for mechanistic interpretability in a vision transformer.

**Strengths:**

- The problem of leveraging synthetic data to perform better statistical inference is well-motivated and timely.
- The paper is well organized and could be understood by a reader unfamiliar with the related literature e.g., synthetic-power predictive inference, conformal prediction.
- Unlike recent works [1] that leverage unlabeled data, GESPI does not assume that the distribution of covariates is the same for the labeled and unlabeled data.
- The general formulation of the GESPI framework (Section 3.4) is elegant, and the main results of the paper (Theorems 3.2 and 3.3) are well-explained.
- The experimental results for conformal risk control in protein structure prediction and outlier detection are compelling. For the outlier detection example, I would be interested in seeing how the performance of GESPI changes as a function of the size and contamination rate of the reference set.
- The manuscript includes detailed appendices explaining experimental setup and providing proofs of theoretical results.

[1] Angelopoulos et al., “Prediction-powered inference”. Science, 2023.

**Weaknesses:**

- The authors claim (p. 2)  that when synthetic and real data are well-aligned, GESPI “mimics the effect of applying the base method to a larger real dataset.” I disagree with this claim.

In particular, suppose one would like to test $H_0: \mu = 0$ against $H_1: \mu > 1$ in the normal family $P_{\mu} = N(\mu, 1)$. Then according to eq. (3), the power of GESPI under any alternative is at most $P_{\mu}(Z > z_{1-\alpha - \epsilon} - \sqrt{n} \mu)$. However, if the synthetic data is also _i.i.d._ from $P_{\mu}$, then the uniformly most powerful level-$\alpha$ test achieves power $P_{\mu}(Z > z_{1-\alpha} - \sqrt{n + N} \mu)$. As $N \to \infty$ (the number of synthetic data points increases to infinity), the UMP test has power $\to 1$, whereas that of GESPI remains constant. More succinctly, the power of GESPI is limited by the real data sample size.

I understand that this is a consequence of the fact that GESPI is a distribution-free method–as pointed out in Appendix C.1–but the authors should be more upfront about this limitation in the main text.

- In the testing experiments (Section 4.2), it’d be helpful to see a comparison to the power of the level-$(\alpha + \epsilon)$ OnlyReal test. It is clear that the GESPI test controls Type I error approximately at level $\alpha$ (and at worst $\alpha + \epsilon$), but does it obtain power close to that of the level-$(\alpha + \epsilon)$ test?

- The paper could benefit from more specifics about how the general GESPI procedure relates to Synthetic-Powered Predictive Inference procedure from Bashari et al., 2025 [2], either in the main text or in Appendix A.

[2] Bashari et al., “Synthetic-powered predictive inference”. arXiv preprint arXiv:2505.13432, 2025.

**Questions:**

- If a practitioner would like to apply GESPI, how would you advise that they choose $\epsilon$? Can $\epsilon$ potentially be chosen as a function of the data (e.g., make $\epsilon$ large if the empirical distributions of the real and synthetic data look similar)?
- Related to the normal testing example above, I’d imagine one could quantify the suboptimality of GESPI compared to the pooled (UMP) test against local alternatives when $n, N \to \infty$ and $N / n \to \gamma$ while $\epsilon$ remains fixed. Can you give a formal result of this flavor?

---

> ### Author Response · Authors · 2025-11-20
> **Rebuttal by Authors (1/2)**
>
> Thank you for your thoughtful review and encouraging feedback. We greatly appreciate your support. **If you believe that our paper has value, it would be great if you could support it in the discussion among the reviewers**. Below, we address your comments in detail.
> > **C1**: For the outlier detection example, I would be interested in seeing how the performance of GESPI changes as a function of the size and contamination rate of the reference set.
>
> We have conducted additional outlier detection experiments in response to the reviewer’s request. These experiments are available in the supporting document ([link](https://tinyurl.com/rebuttal-exp)).
>
> Figure S8 shows performance as a function of the contamination rate for FWER control (a) and type I error control (b). As the contamination rate increases, the overall power of all methods decreases because the model is trained on a contaminated set. Consistent with the main manuscript, GESPI achieves near-oracle power while controlling the error rate close to $\alpha$. OnlyReal controls the error rate but has lower power due to the small sample size. Notably, GESPI exhibits smaller variance than the guardrail (real data at level $\alpha+\varepsilon$); for example, at 7\% contamination, the guardrail achieves FWER as high as $\approx 0.7$ (target 0.15), while GESPI maintains much lower variance. Similar trends hold for type I error.
>
> Figure S9 shows performance as a function of the real sample size $n$ for FWER (a) and type I error (b). For very small $n$ (60 for FWER, 20 for type I), OnlyReal and the guardrail achieve the same power, leaving little room for leveraging the synthetic data; thus, GESPI performs similarly. As $n$ increases, GESPI achieves higher power than OnlyReal while obtaining error close to the target level $\alpha$. For large enough $n$, all methods achieve comparable power. For type I error, this occurs around $n \approx 60$, whereas for FWER, which is a multiple hypothesis testing problem and more challenging, it occurs around $n \approx 150$ in this setting.
> > **W1**: ``The authors claim that when synth. and real data are well-aligned, GESPI “mimics the effect of […] a larger real dataset.’’ I disagree with this claim.''
>
> We thank the reviewer for catching this. This was an unfortunate inaccurate phrasing, where, in fact, there are a few important words missing: the accurate phrasing is ``GESPI mimics the effect of applying the base method to a larger real dataset but capped by the performance of the worst-case guardrail.'' With that said, this is formally stated in Theorem 3.3. We thank the reviewer for bringing this to our attention and will revise the text accordingly.
> > **W2**: ``it’d be helpful to see a comparison to […] the level-$(\alpha+\epsilon)$ OnlyReal.''
>
> This is a great point. We include the guardrail baseline---using the real data at level $\alpha+\varepsilon$ (`Guardrail`)---across all experiments in the supporting document (see [link](https://tinyurl.com/rebuttal-exp)). We can see that across all experiments, GESPI achieves power close to the guardrail, including in two newly added experiments beyond those mentioned in $C1$: (i) class-conditional coverage in image classification on ImageNet with FLUX-generated images, and (ii) hypothesis testing for hyperparameter selection for K-way majority vote in the large reasoning model.
>
> Specifically, GESPI achieves near-oracle power in the outlier detection experiments (Figures S5 and S6), similar rejection rates in the large reasoning model comparison (Figures S3 and S4), close abstention rates in the protein structure prediction experiments (Figure S7), and comparable prediction set sizes in the conformal prediction experiments (Figure S1). As expected, these metrics are not exactly identical to the guardrail, as GESPI achieves risk close to $\alpha$ rather than $\alpha+\varepsilon$. This is further illustrated in the hypothesis testing with simulated data experiments, which cover all possible scenarios---null and alternative, low- and high-quality synthetic data---demonstrating the performance of all methods across these settings.
>
> That said, power alone does not tell the full story. The guardrail method **does not leverage the synthetic data, and therefore still suffers from the low real sample size, even if it is run at a higher level**. As a result, this method leads to a large variability (even though the average risk is controlled at  $\alpha + \varepsilon$, the variance can be much higher than that of our GESPI method). This is illustrated in the new experiment (i): as shown in Figure S1 in this [link](https://tinyurl.com/rebuttal-exp), for the *Magpie* class, the variance of the guardrail method is extremely large due to the small sample size, leading to realized coverage as low as 60\% even though the target level is 95\%. In contrast, GESPI achieves coverage close to 95\% with far lower variability, making it clearly preferable to the guardrail approach in this case.

---

> ### Author Response · Authors · 2025-11-20
> **Rebuttal by Authors (2/2)**
>
> > **W3**: The paper could benefit from more specifics about how the general GESPI procedure relates to Synthetic-Powered Predictive Inference procedure from Bashari et al., 2025 [2], either in the main text or in Appendix A.
>
> We thank the reviewer for this suggestion. We will include a more detailed discussion of how GESPI relates to SPI in the main manuscript if the page limit allows; otherwise, we will provide this discussion in Appendix A.
>
> > **Q1**: If a practitioner would like to apply GESPI, how would you advise that they choose $\epsilon$? Can $\epsilon$ potentially be chosen as a function of the data (e.g., make $\epsilon$ large if the empirical distributions of the real and synthetic data look similar)?
>
>
> We suggest **viewing the user-specified $\varepsilon$ as the additional worst-case risk the user is willing to tolerate when the synthetic data are of poor quality**. That is, $\varepsilon$ allows the user to better control the trade-off between the worst-case risk (low quality synthetic data) versus best-case gain (high quality synthetic data).
>
> Choosing $\varepsilon$ in a fully data-driven manner would essentially require knowing the quality of the synthetic data---a priori knowledge that is unrealistic, and estimating it is challenging, especially with limited real data.
>
> This is precisely the challenge that GESPI addresses: **the design choice of introducing $\varepsilon$ effectively bypasses the need to formally quantify or know a priori the quality of the synthetic data**, with the premise that if the synthetic data are of high quality, the risk will be controlled near the level $\alpha$ while still benefiting from the larger synthetic sample size.
>
> That said, **one practical approach to decide whether to use the synthetic data or not could be to apply a goodness-of-fit test**. Although it may seem that this requires data-splitting, in practice, the test can be applied as a part of the synth.-powered algorithm. Specifically, the test compares the empirical risk of the synth. data to that of the real data. If the two are significantly different, so that a suitable null hypothesis is rejected, this indicates that the synthetic data differ substantially from the real data and should not be used. In this case, the synth.-powered algorithm simply applies the risk-controlling method on the real data at level $\alpha$. This deterministically produces the same result as the base method and, practically, yields the same performance.
>
> This discussion also raises an interesting question of how $\varepsilon$ could be chosen in a more data-adaptive way, rather than in a binary use-or-not manner. Conceptually, this can be achieved by using a risk-controlling algorithm that is based on e-values. The advantage of e-values is that they allow observing the data first and then setting the error level. We believe this is a very interesting future direction.
> > **Q2**: Related to the normal testing example above, I’d imagine one could quantify the suboptimality of GESPI compared to the pooled (UMP) test against local alternatives when $n,N\rightarrow\infty$ and $N/n\rightarrow \gamma$ while $\epsilon$ remains fixed. Can you give a formal result of this flavor?
>
> Quantifying this gap is indeed an interesting question, and we thank the reviewer for raising it. We are making efforts to include a formal theoretical result of this nature during the rebuttal, and in any case, we plan to include it in the paper. Interestingly, our Theorem 3.3 already supports the reviewer’s intuition, as it formally states that the power of GESPI is capped by the guardrail. We are currently working on precisely quantifying this gap and hope to provide an updated result soon.

---

> ### Comment · Reviewer_3sMY · 2025-11-22
> **Response to Authors**
>
> I thank the authors for their thorough rebuttal, particularly their inclusion of comparisons to the risk level-$(\alpha + \epsilon)$ guardrail across all experiments, their comparison of GESPI to SPI (Figure S3), and their discussion of how the risk level $\epsilon$ of the guardrail can be chosen via a goodness-of-fit test (by wrapping the test in the synth.-powered algorithm). Adding these results/discussion would greatly improve the practicality of the authors' manuscript, and I encourage them to do so.
>
> As to the authors' point about the variability of GESPI versus the risk level-$(\alpha + \epsilon)$ guardrail, it is not clear to me that this is an important metric to consider. For example, in Figure S5, my understanding is that each of the points represents the power and Type I error estimated across 50 trials, where each trial corresponds to randomly sampling $n$ AIME questions and $N$ synthetic questions. In other words, the Type I error is just a sample mean of $\text{Bern}(\alpha)$ random variables. And so if GESPI controls the Type I error at level $\alpha$, the variance of these sample means should be smaller than the guardrail. In other words, how is this observation different from the authors' previous observation that GESPI empirically controls the risk at level smaller than $\alpha + \epsilon$? Clarifying the conceptual distinction between these two points would be helpful.

---

> > ### Author Response · Authors · 2025-11-25
> > **Response to Reviewer 3sMY**
> >
> > Thank you for your thoughtful engagement and encouraging feedback. We appreciate the reviewer for going beyond our responses and carefully reviewing the supporting document. We will include all relevant discussion and results in the revised manuscript.
> >
> > Regarding methods that control risk at level $\alpha$ versus $\alpha+\varepsilon$, we agree with the reviewer that variance is smaller at the lower risk level (assuming $\alpha+\varepsilon \leq 0.5$). However, GESPI can further obtain lower variance in some cases, not only due to the lower empirical risk level, but also by leveraging additional synthetic data when these data are of high quality. This advantage also holds when comparing to OnlyReal, which uses only real data at level $\alpha$ and suffers from the limited sample size.
> >
> > The interpretation of the variance in the specific hypothesis testing experiment mentioned by the reviewer is indeed different. Here, the empirical type I error is a Bernoulli trial with probability $\alpha$, as the reviewer correctly noted. In this setting, the variance depends on the number of trials rather than the sample size. Thus, the observed variability in type I error is not the variance reduction benefit we mentioned during the rebuttal phase for GESPI. In this experiment, only the mean of the type I error and the comparison in power are informative, as power depends on the sample size. We thank the reviewer for pointing this out and will clarify it in the revised version.
> >
> > In contrast, for all other applications considered in this paper---conformal prediction, outlier detection, etc.---the variance of the empirical risk does depend on the sample size. In these cases, variance reduction is particularly important in practice, as a practitioner has access to only a single dataset, and lower variance ensures that the holdout-data-conditional risk is more likely to be close to the target level. Thus, controlling the empirical risk in expectation alone does not fully capture the procedure’s practical performance.
> >
> > For example, in conformal prediction, the holdout-data-conditional coverage follows a beta distribution (assuming infinite test points), where the variance is approximately $\alpha\cdot(1-\alpha) / (n+2)$, which shrinks with the sample size $n$. Hence, when the synthetic data are of high quality, **GESPI offers an additional benefit over using only the real data**: it not only achieves coverage close to the nominal level $1-\alpha$ with higher power than using only real data at level $\alpha$, but also **further reduces variance by leveraging additional synthetic data, resulting in higher stability**.
> >
> > We mentioned the comparison to the guardrail since both the guardrail and GESPI provide the same distribution-free guarantees at level $\alpha+\varepsilon$, and we wanted to clearly illustrate that GESPI is preferable to naively using only the real data at level $\alpha+\varepsilon$.
> >
> > We greatly appreciate the reviewer’s positive evaluation of our work and the thoughtful discussions and suggestions that strengthen it. If the reviewer feels that our work deserves acceptance at ICLR, we would be grateful if they could consider increasing their score.

---

> ### Comment · Reviewer_3sMY · 2025-11-27
> **Response to Authors**
>
> I thank the reviewer for answering my question regarding the settings in which the increased sample size resulting from using synthetic data could reduce the variance of the empirical risk.
>
> I find the author's example of split conformal prediction, wherein the size of the calibration dataset (i.e., the dataset used to compute the conformal prediction set) increases with the amount of synthetic data, to be convincing. Moreover, this claim is supported by the authors' additional experiments on conformal prediction for image classification (Figure S1).
>
> My primary concern with the manuscript in its current form is that, as other reviewers also noted, the key strengths of GESPI are not presented with sufficient clarity or supported consistently by the experiments. As pointed out by the authors, the main reason to use GESPI over the risk level $(\alpha + \epsilon)$ controlling procedure, in addition to controlling the risk at a potentially smaller level, is that the variability in the empirical risk will be smaller. Despite being knowledgeable about conformal prediction, this was not evident to me upon reading the manuscript, wherein no comparisons are provided to the level-$(\alpha + \epsilon)$ procedure and many of the examples given (in the main text and appendix) are in the hypothesis testing setting wherein this comparison does not apply.
>
> I will keep my score of 6 and recommend that the authors revise their manuscript to better highlight the contributions of their work.

---

> > ### Author Response · Authors · 2025-11-27
> > **Response to Reviewer 3sMY**
> >
> > Thank you for the constructive feedback and for emphasizing the need to better highlight GESPI’s strengths.
> >
> > In the revised manuscript, we will (i) clarify the additional variance-reduction benefits of GESPI in conformal prediction, conformal risk control, and outlier detection (single and multiple hypothesis testing), (ii) add a dedicated section on the experiments with conformal prediction, including a detailed explanation of the variance reduction, (iii) include explicit comparisons to the level-$(\alpha+\varepsilon)$ guardrail across all experiments as provided in the supporting document, and (iv) explicitly mention the evaluation metrics in the experiments section (section 4).
> >
> > We appreciate your comments and believe these revisions will make the contributions clearer.

---

### Official Review · Reviewer_MhsE · 2025-10-31

**Soundness:** 2
**Presentation:** 3
**Contribution:** 2
**Rating:** 2
**Confidence:** 3

**Summary:**

This paper proposes GESPI, a general framework for leveraging synthetic data in statistical inference with guaranteed error-rate control. The method wraps around a base inference algorithm. It runs the base method three times: (1) standard, (2) a more relaxed guardrail version using only real data, and (3) pooled real+synthetic. It then aggregates the outputs to ensure the final decision does not exceed an error-rate bound, even if the synthetic data is misaligned. Experiments span risk-controlled protein structure prediction, model comparison, out-of-distribution detection, and interpretability.

**Strengths:**

1. It addresses an important and timely problem: using synthetic data safely.
2. The proposed framework is general, applies to various inference tasks, with finite-sample guarantees providing rigor and confidence in decision-making.

**Weaknesses:**

1. The core idea resembles Synthetic-Powered Predictive Inference (SPI) (Bashari et al. 2025) with a wrapper reformulation. The conceptual advance over existing selective-use-of-synthetic-data methods is modest.
2. The applications largely show numerical improvements but do not demonstrate meaningful decision changes attributable to GESPI. For example, in AlphaFold experiments, why should abstaining on slightly fewer residues matter biologically?
3. Lack of baseline: each application should have more baselines. For example, the hypothesis testing should have PPI and PPI++ as baseline. Similar for other applications.
4. The synthetic data in the experiments is unusually high-quality (e.g., AlphaFold MSAs), biasing results toward success scenarios. In practice, the synthetic data may not be of such high quality

**Questions:**

1. Can you provide one clear decision-making example where the guardrail version changes the outcome in a scientifically significant way?
2. How will the other baseline perform in those tasks?

---

> ### Author Response · Authors · 2025-11-20
> **Rebuttal by Authors (1/2)**
>
> We thank the reviewer for the feedback. Below, we address each comment in detail.
> > **W1**: ``The core idea resembles Synthetic-Powered Predictive Inference (SPI) (Bashari et al. 2025) with a wrapper reformulation. The conceptual advance over existing selective-use-of-synthetic-data methods is modest.''
>
> We respectfully disagree with the reviewer. Our work introduces two key contributions that go well beyond a reformulation of SPI as a wrapper. We kindly refer the reviewer to our response to **Comment 1** in our **General comment**, posted at the top of the OpenReview page.
> > **W2**: ``The applications largely show numerical improvements but do not demonstrate meaningful decision changes attributable to GESPI. For example, in AlphaFold experiments, why should abstaining on slightly fewer residues matter biologically?''
>
> Abstaining on slightly fewer residues **matters biologically because it increases the number of residues considered reliable, providing more informative sites for downstream experimental or computational decisions**. For example, in protein-ligand drug discovery, filtering out too many residues can remove viable residues in potential binding sites, reducing the pool of actionable candidates. Moreover, knowing that certain residues are reliably predicted can influence the downstream decisions about what experiments to do and thus meaningfully save many researcher-hours.
>
> To provide some scientific context, consider the following: Bryant et al. (2024)[1] shows that pLDDT (the confidence score used for risk control in our experiments) correlates with ligand pose prediction success, enabling researchers to prioritize candidates for experimental validation.
>
> Applying risk-control in the protein structure prediction application moves beyond heuristic pLDDT thresholds by selecting a calibrated threshold with risk-control guarantees. Without such risk control, two issues arise: false positives---low-confidence residues mistakenly included---waste experimental resources, and false negatives---overly conservative filtering---can leave too few candidates to pursue, reducing statistical power. Thus, our method **enables reducing wasted experimental resources and increasing statistical power in detecting true discoveries**, both of which have very real consequences.
>
> Moreover, Tejera-Nevado et al. (2024)[2] show that pLDDT and related confidence metrics are critical for distinguishing reliable from uncertain protein regions across multiple prediction models, and Harmalkar et al. (2024)[3] demonstrates that confidence-based filtering improves protein-protein docking accuracy, highlighting the broader relevance of confidence thresholds for downstream tasks.
>
> [1] Bryant et al., ``Structure prediction of protein-ligand complexes from sequence information with Umol,'' Nature Communications, 2024.
>
> [2] Tejera-Nevado et al., ``Unlocking the power of AI models: exploring protein folding prediction through comparative analysis,'' Frontiers in Molecular Biosciences, 2024.
>
> [3] Ameya Harmalkar, Sergey Lyskov, Jeffrey J Gray ``Reliable protein-protein docking with AlphaFold, Rosetta, and replica-exchange,'' eLife, 2024.
>
> > **W3**: ``Lack of baseline: each application should have more baselines. For example, the hypothesis testing should have PPI and PPI++ as baseline. Similar for other applications.''
>
> The methods **PPI and PPI++ assume that the covariates ($X$) of the labeled and unlabeled data are i.i.d., which is fundamentally different from the setting considered in our work**. For example, in the hypothesis or multiple hypothesis testing problems, this assumption implies that the synthetic data come from the same distribution as the real data. As a result, these methods are not applicable in our setting.
>
> At the same time, although it is impossible to compare to them, **PPI and PPI++ can, in fact, be used within our GESPI framework as the synthetic-powered algorithm**. In this case, validity is guaranteed regardless of whether the i.i.d. covariate assumption holds. This combination of PPI and GESPI presents an exciting research direction for future work.

---

> ### Author Response · Authors · 2025-11-20
> **Rebuttal by Authors (2/2)**
>
> > **W4**: ``The synthetic data in the experiments is unusually high-quality (e.g., AlphaFold MSAs), biasing results toward success scenarios. In practice, the synthetic data may not be of such high quality''
>
> Thank you for raising this point. For the AlphaFold experiments, **the opposite is true: the synthetic data (AlphaFold MSAs) consist unlabeled protein sequences, and the real experimental structures are unavailable, so these synthetic data are far from high quality**. A similar point can be made about our example in mechanistic interpretability: our experiments show that the distribution of activations is in fact very different for the synthetic FLUX-based generated data than for the real one.
>
> As emphasized throughout the paper, GESPI automatically adapts to the quality of the synthetic data. When the synthetic data are of high quality, GESPI benefits from the synthetic data and achieves risk near the nominal level $\alpha$; otherwise, GESPI defaults to the performance of the guardrail. This behavior is further illustrated in the simulated experiments (Appendix C.1).
>
> That said, to illustrate this point on real data, we have added new experiments on class-conditional coverage for ImageNet with FLUX-generated synthetic data, Figure S1 in the supporting document in this [link](https://tinyurl.com/rebuttal-exp). We observe that when the synthetic data are of low quality (e.g., the Siberian husky class), GESPI defaults to the performance of the guardrail. As the quality of the synthetic data improves, GESPI effectively leverages it, achieving coverage close to the nominal $1-\alpha$ level. Interestingly, while the synthetic husky images appear visually high-quality (Figure S2), they do not provide high-quality information for statistical purposes, further emphasizing the importance of safely leveraging synthetic data.
> > **Q1**: ``Can you provide one clear decision-making example where the guardrail version changes the outcome in a scientifically significant way?''
>
> See W2.
> > **Q2**: ``How will the other baseline perform in those tasks?''
>
> As explained in $W3$, the other baselines (PPI and PPI++) rely on assumptions that do not hold in our setting and are therefore not applicable. To the best of our knowledge, there is no method that provides distribution-free guarantees without any assumptions on the synthetic data.

---

> > ### Comment · Reviewer_MhsE · 2025-11-27
> >
> > I thank the authors for the detailed response. I am raising my score to 4 to reflect that the authors have addressed most of my concern. However, I agree with other reviewers that the benefits of GESPI over only using real data is not very clearly demonstrated. I would encourage the authors to more clearly to show this point. For example, the PPI++ shows the inference results is valid and more efficient than only using real data.

---

> > > ### Author Response · Authors · 2025-11-27
> > > **Response to Reviewer MhsE**
> > >
> > > We thank the reviewer for raising their score. We are glad to hear that we have addressed most of their concerns.
> > >
> > > We would like to clarify that our work focuses on a particularly challenging setting in which **no assumptions are made regarding the synthetic data or their quality**. In contrast, methods such as PPI and PPI++ rely on additional assumptions on the synthetic data---i.e., assuming the $X$'s of the synthetic and real data are drawn from the same distribution. We do not make such assumptions, and naturally, methods that rely on stronger assumptions can achieve larger performance gains in practice. When using generated AI data or auxiliary data, such assumptions do not hold, and **our work is designed to handle this more general setting**. It is also possible that the empirical improvement the reviewer observed in the context of PPI is supported by asymptotic guarantees, which in practice may lead to larger gains compared to guarantees that hold in finite samples. For these reasons, we find this comparison not relevant, yet we understand why it may be confusing at first.
> > >
> > > The fair comparison is to methods that use only real data, as there are no existing methods that utilize synthetic data without additional assumptions for the problems considered in this work, as we do in our setting. As described at length in the paper and in the rebuttal, our experiments show significant improvement in performance when the synthetic data are of high quality.
> > >
> > > We thank the reviewer for the comments, and we will clarify these points, as well as the metrics in the comparisons, and further emphasize the practical importance of such performance gains.

---

### Official Review · Reviewer_FjoW · 2025-11-01

**Soundness:** 2
**Presentation:** 2
**Contribution:** 2
**Rating:** 2
**Confidence:** 5

**Summary:**

This paper proposes GEPSI, a general framework for incorporating synthetic data into statistical inference while maintaining distribution-free, finite-sample guarantees.
The method applies to any inference procedure three times and aggregates the outputs to ensure that the overall error rate never exceeds \alpha+\epsilon.
While the framework is conceptually interesting, the contribution is somewhat incremental relative to prior work on synthetic-powered inference (e.g., SPI) and conformal prediction.
More importantly, theoretically, should the proposed method maintain the error rate at \alpha instead of \alpha+\epsilon?
The intuition is that, if one method has a relaxed error rate control, it should of course have a greater power.
In addition, the empirical studies are illustrative but limited in scope and depth, and the practical impact of the proposed method remains unclear.

**Strengths:**

The method is general and easy to implement, acting as a wrapper for existing inference procedures.

Finite-sample, distribution-free guarantees are theoretically appealing and well-articulated.

**Weaknesses:**

Incremental contribution: The proposed framework closely parallels earlier work such as Synthetic-Powered Predictive Inference (SPI). GESPI mainly reformulates SPI in a more general setting but adds limited new theoretical or methodological insight.

Weak experimental evidence: The experiments are small-scale and largely illustrative. The improvements over the baseline (using real data only) are modest, and the synthetic data setups (e.g., AlphaFold MSA sequences or Olympiad math questions) feel somewhat ad hoc.

Limited novelty in theory: The theoretical guarantees mostly extend standard monotonicity and total variation arguments; there is little fundamentally new mathematical insight.

Parameter sensitivity: The \epsilon parameter plays a critical role in balancing power and validity, yet there is no principled or data-driven way to choose it.

Lack of strong baselines: The comparisons omit relevant existing methods for combining heterogeneous datasets, such as transfer learning, domain adaptation, or causal data fusion approaches.

Unclear practical relevance: It is not evident that practitioners would find GESPI advantageous, given the need to rerun inference procedures multiple times and the modest empirical gains.

**Questions:**

How does GESPI differ fundamentally from SPI beyond the reformulation to a more general inference framework?

Could you provide theoretical or empirical comparisons with domain adaptation or semi-supervised learning approaches that also combine datasets from different distributions?

How sensitive is performance to \epsilon, and could it be tuned automatically rather than chosen heuristically?

Can the method be applied meaningfully at large scale, where repeated runs of the base inference algorithm may be infeasible?

How can one assess when synthetic data are sufficiently aligned to expect benefits from GESPI?

Should the proposed method maintain the error rate at \alpha instead of \alpha+\epsilon? Otherwise, it is not a fair comparison with the baseline which controls the error rate at \alpha.

In all the numerical studies, should you compare with the method Real+Synth? Why OnlySynth?

---

> ### Author Response · Authors · 2025-11-20
> **Rebuttal by Authors (1/2)**
>
> We thank the reviewer for the feedback. Below, we address each comment in detail.
> > **W1**: ``Incremental contribution’’
>
> We respectfully disagree with the reviewer. Our work introduces two key contributions that go well beyond a reformulation of SPI as a wrapper. We kindly refer the reviewer to our response to **Comment 1** in our **General comment**, posted at the top of the OpenReview page.
> > **W2-1**: ``Weak experimental evidence: The experiments are small-scale and largely illustrative. The improvements over the baseline (using real data only) are modes.''
>
> We respectfully disagree with the reviewer and kindly refer the reviewer to our response to **Comment 2** in our **General comment**, posted at the top of the OpenReview page.
> > **W2-2**: ``Weak experimental evidence: The synthetic data setups (e.g., AlphaFold MSA sequences or Olympiad math questions) feel somewhat ad hoc.''
>
> **The synthetic data setups reflect natural and domain-driven choices**.  For the large reasoning model comparison, it is natural to construct synthetic data from math questions that resemble the real questions of interest (the limited AIME25 dataset). For the AlphaFold experiments, obtaining additional relevant protein sequences is non-trivial; thus, reusing the sequences that AlphaFold2 itself retrieved for structure prediction is a practically sound and reasonable choice.
>
> Moreover, GESPI allows such use of real-data-dependent synthetic data, which can be adapted or selected based on the real data. This can potentially lead to better alignment with the real distribution and thus be more useful within the GESPI framework.
>
> > **W3**: ``Limited novelty in theory''
>
> We propose a **novel theoretical formulation for leveraging synthetic data in the context of statistical decision theory**, which is based on controlling a general loss function. This allows us to tackle a broad range of statistical problems in a unified framework, including hypothesis testing, multiple testing, confidence intervals, etc. This perspective enables us to develop our powerful new method.
>
> We agree that TV-based bounds are present in the literature on conformal prediction, but **in the context of synthetic data, no such bounds exist for the broad class of problems addressed in this work**. To the best of our knowledge, SPI is the only prior method that provides distribution-free guarantees in this setting, and it is explicitly tailored to conformal prediction. Even within conformal prediction, our theoretical guarantees differ from those in SPI (see Theorem H.1 versus Theorems 3.3 and 3.5 in SPI). In addition, we present theoretical results on the power of GESPI---results that do not exist for SPI.
> > **W4**: ``Parameter sensitivity''
>
> We suggest **viewing the user-specified $\varepsilon$ as the additional worst-case risk the user is willing to tolerate when the synthetic data are of poor quality**. That is, $\varepsilon$ allows the user to better control the trade-off between the worst-case risk (low quality synthetic data) versus best-case gain (high quality synthetic data).
>
> Choosing $\varepsilon$ in a fully data-driven manner would essentially require knowing the quality of the synthetic data. To illustrate this, consider a thought experiment: suppose we knew that the synthetic data were of poor quality; in that case, we would set $\varepsilon = 0$, as the synthetic data would be useless. Analogously, if we knew that the synthetic data were perfectly aligned with the real data, we would choose a larger $\varepsilon$, since we would know that the risk would be controlled at level $\alpha$. Of course, this thought experiment assumes an unrealistic scenario in which we know a priori whether the synthetic data are perfect.
>
> This is precisely the challenge that GESPI addresses: **the design choice of introducing $\varepsilon$ effectively bypasses the need to formally quantify or know a priori the quality of the synthetic data**, with the premise that if the synthetic data are of high quality, the risk will be controlled near the level $\alpha$ while still benefiting from the larger synthetic sample size.
>
> That said, **one practical approach to decide whether to use the synthetic data or not could be to apply a goodness-of-fit test**.  Although it may seem that this requires data-splitting, in practice, the test can be applied as a part of the synth.-powered algorithm. Specifically, the test compares the empirical risk of the synthetic data to that of the real data. If the two are significantly different, so that a suitable null hypothesis is rejected, this indicates that the synthetic data differ substantially from the real data and should not be used. In this case, the synth.-powered algorithm simply applies the risk-controlling method on the real data at level $\alpha$. This deterministically produces the same result as the base method and, practically, yields the same performance.

---

> ### Author Response · Authors · 2025-11-20
> **Rebuttal by Authors (2/2)**
>
> > **W5**: ``Lack of strong baselines''
>
> As far as we know, **there is no suitable method in transfer learning or domain adaptation that offers distribution-free guarantees without data assumptions**. Prior works place various restrictions on the problem that do not hold in the general setup we study and are therefore not applicable. However, if there is any specific method that the reviewer believes performs the same goal as we do and should be compared with, we would appreciate it if they could refer us to it.
> > **W6**: ``Unclear practical relevance: [...]given the need to rerun inference procedures multiple times and the modest empirical gains.''
>
> To clarify, **the runtime of the statistical inference procedure is completely negligible compared to the computational cost of running machine learning models**. Therefore, the fact that GESPI requires running the inference procedure three times has essentially no impact on practical relevance. In fact, the opposite is true: by leveraging synthetic data, GESPI strengthens the underlying statistical test, making it more useful and practically relevant.
>
> For example, in the hypothesis-testing experiments for large reasoning model comparison, the runtime of the binomial test is effectively zero relative to the cost of running the reasoning models on the dataset. Thus, running the test three times incurs no meaningful overhead. For this reason, we respectfully disagree with the reviewer, and **we believe we have provided ample evidence that practitioners would find it highly useful**.
>
> Moreover, there are situations where GESPI runs the inference method only once. For example, for conformal methods, calibration amounts to sorting arrays of size $n$ and $N+n$ only once, then computing a calibrated threshold, which is fixed and applied to the entire test set. The additional computation required by GESPI in this setting is likewise negligible.
> > **Q1**: How does GESPI differ fundamentally from SPI?
>
> See W1.
> > **Q2**: Could you provide theoretical or empirical comparisons with domain adaptation or SSL?
>
> See W5.
> > **Q3**: How sensitive is performance to $\epsilon$ [...]?
>
> In turn, $\varepsilon$ defines the maximum power that GESPI can achieve (in terms of prediction set size, etc.), as formally stated in Theorem 3.3. As discussed in W4, **$\varepsilon$ represents the additional worst-case risk the user is willing to tolerate when the synthetic data are of poor quality**. Thus, the choice of $\varepsilon$ naturally depends on the application and the practitioner’s preferences. At the same time, one can test if the synthetic data are far off the real data; if so, $\varepsilon$ can be set to zero, as described in W4.
> > **Q4**: Can the method be applied meaningfully at large scale?
>
> See W6.
> > **Q5**: How can one assess when synth. data are sufficiently aligned[…]?
>
> Assessing the quality of synthetic data is inherently difficult. In principle, one would need a holdout set of labeled real and synthetic data to measure the discrepancy between the risk evaluated on the real data and that evaluated on the synthetic data. As mentioned in W4, one can test whether the synthetic data significantly differ from the real data, which can help detect extreme cases where the synthetic data are clearly not useful, though it does not guarantee that they are sufficiently well-aligned.
>
> **GESPI bypasses this challenge by setting $\varepsilon$ a priori, without requiring explicit knowledge of whether the synthetic data are sufficiently aligned**. It adaptively leverages the synthetic data when they are well aligned, while guaranteeing that the risk is always controlled at level $\alpha + \varepsilon$ in the worst case.
> > **Q6**: Should the proposed method maintain the error rate at $\alpha$ instead of $\alpha+\epsilon$? Otherwise, it is not a fair comparison with the baseline which controls the error rate at $\alpha$.
>
> The goal is to control the error rate at level $\alpha$. However, to safely leverage the synthetic data, in the worst case, GESPI controls the error rate at level $\alpha + \varepsilon$. Regarding the comparison with the worst-case guardrail---using the real data at level $\alpha+\varepsilon$---we kindly refer the reviewer to our response to **Comment 3** in our **General comment**, posted at the top of the OpenReview page.
> > **Q7**: Should you compare with the method Real+Synth? Why OnlySynth?
>
> The `OnlySynth` baseline is included solely to illustrate the quality of the synthetic data, and it is not truly a competing baseline since it does not provide any risk-control guarantees. That said, we also include the pooled real and synthetic baseline `Synth+Real`; see the supporting document in this [link](https://tinyurl.com/rebuttal-exp)), which, similar to OnlySynth, does not offer any form of risk control guarantees.
>
> As shown in this document, `Synth+Real` performs very similarly to `OnlySynth` across most scenarios, which is expected, especially in cases where $N\gg n$.

---

> > ### Comment · Reviewer_FjoW · 2025-11-26
> >
> > I thank the authors to spend time and energy to address my questions and comments. Some of them become clearer now. I now understand better the difference with the previous paper and the scope of the numerical studies presented in this paper. I still have concerns about the following items.
> >
> > 1. For a testing problem, one usually fix the type I error \alpha and then compare the power of different testing procedures. It is reasonable to compare different procedures which control the same type I error rate. It is still quite confusing why the proposed method can control a higher type I error but its competitors cannot. It is definitely reasonable if some procedure is allowed to make type I error "more", then it of course will have a "greater" power. I know the authors explained this, but I feel this is still quite confusing.
> >
> > 2. What is the intuition behind the finding that, regardless of the quality of the synthetic data, one can always increase the performance compared to the method that does not use synthetic data? It still seems counterintuitive to me.
> >
> > 3. The comparison to Synth+Real is a lot more critical, than the comparison to OnlySynth. When the synthetic data has a very poor quality, OnlySynth is expected to perform poor; however, the method Synth+Real should still perform well. The answer to my last question is still not satisfactory. When the synthetic data has a good quality and when N>>n, of course these two methods should perform similarly. However, what is the performance when the synthetic has a poor quality?

---

> > > ### Author Response · Authors · 2025-11-26
> > > **Response to Reviewer FjoW (1/2)**
> > >
> > > We thank the reviewer for their engagement and the opportunity to provide further clarification. We are glad that our explanations were helpful, and we address the remaining points below.
> > >
> > > > Q1.
> > >
> > > Ideally, as the reviewer said, one would want a procedure that leverages synthetic data and always controls the type I error at level $\alpha$. However, this is not possible without any prior knowledge of the distribution of synthetic data. In this work, we suggest a **new perspective on how to leverage synthetic data and provide new risk control guarantees that safely account for the unknown quality of the synthetic data**, effectively providing a ``guardrail'' against worst-case scenarios.
> > >
> > > Importantly, there are currently no other methods that provide this same guarantee. This is a key contribution of our work. The overall message in the experiments is not to claim that GESPI dominates or strictly improves existing methods with the ``same guarantee"---since such methods do not exist---, but rather to illustrate the benefits of leveraging large synthetic data while ensuring that the risk never exceeds the guardrail bound $\alpha+\varepsilon$.
> > >
> > > GESPI does not sacrifice type I error to gain higher power. Instead, it benefits from a larger sample size---e.g., to improve power in the hypothesis testing---when possible, while safely ensuring that the type I error does not exceed the predefined level. For example, in Figures S4-S7 in the supporting document ([link](https://tinyurl.com/rebuttal-exp)), we observe that GESPI obtains higher power than OnlyReal while still achieving type I error close to the target level $\alpha$---specifically, below that of the guardrail method.
> > >
> > > Regarding the comment, ``It is still quite confusing why the proposed method can control a higher type I error but its competitors cannot.''
> > >
> > > We note that, as explained above, there is no other method that provides the same guarantee as GESPI. When considering the worst-case scenario, GESPI guarantees distribution-free risk control at level $\alpha+\varepsilon$. As shown in the supporting document (see [link](https://tinyurl.com/rebuttal-exp)), we already compared GESPI to the guardrail method, which uses only the limited real data at level $\alpha+\varepsilon$. Thus, we indeed compare GESPI to a method that controls the risk at level $\alpha+\varepsilon$, even though the goal is to control the risk at level $\alpha$. Unlike the guardrail, GESPI can approach the target level $\alpha$ when the synthetic data are of high quality.
> > >
> > > Moreover, the guardrail method **does not leverage the synthetic data, and therefore still suffers from the low real sample size, even if it is run at a higher level $\alpha+\varepsilon$**. In contrast, **GESPI can effectively enhance the sample efficiency---and thus reduce variance or improve power---by leveraging the synthetic data when are of high quality**.
> > >
> > > Beyond risk and power, another important dimension of performance is the variance of each method. GESPI can reduce variance when high-quality synthetic data are available, which is particularly important for applications such as conformal prediction, conformal risk control, and outlier detection. This is demonstrated across our experiments (Figures S6–S10 in the supporting document), where GESPI shows lower variance than the guardrail. Both methods offer the same distribution-free risk control guarantees (in the worst case for GESPI); therefore, the reduced variance highlights another practical advantage of using GESPI.

---

> > > ### Author Response · Authors · 2025-11-26
> > > **Response to Reviewer FjoW (2/2)**
> > >
> > > > Q2. What is the intuition behind the finding that, regardless of the quality of the synthetic data, one can always increase the performance compared to the method that does not use synthetic data? It still seems counterintuitive to me.
> > >
> > > This statement is not what we claim. Indeed, one cannot always improve performance compared to using only real data. If the synthetic data are of low quality and provide no useful information, then there is simply nothing to gain by using them.
> > >
> > > What we claim, and a more accurate statement, is that, if the user is willing to control the risk in the worst case at level $\alpha+\varepsilon$, then GESPI offers a principled framework to safely utilize synthetic data.
> > >
> > > Crucially, the quality of the synthetic data is unknown a priori. In fact, **GESPI adapts automatically to synthetic data quality**: when the synthetic data are of high quality, GESPI enhances sample efficiency and achieves risk close to the desired level $\alpha$; when the synthetic data are of low quality and not useful, GESPI defaults to using only real data while still guaranteeing valid distribution-free risk control at level $\alpha+\varepsilon$ in the worst case.
> > >
> > > Perhaps the confusion arises from Theorem 3.3. This theorem formally states that GESPI is sandwiched between the base and guardrail baselines: it ensures that GESPI never performs worse than the base (in terms of statistical power, prediction set size, etc.) and is capped by the guardrail. However, this does not guarantee that GESPI will strictly outperform the base, since power alone is not the only relevant metric---it must be considered together with the type I error.
> > >
> > > > Q3. The comparison to Synth+Real is a lot more critical, than the comparison to OnlySynth. When the synthetic data has a very poor quality, OnlySynth is expected to perform poor; however, the method Synth+Real should still perform well. The answer to my last question is still not satisfactory. When the synthetic data has a good quality and when N>>n, of course these two methods should perform similarly. However, what is the performance when the synthetic has a poor quality?
> > >
> > > We first emphasize that **neither OnlySynth nor Synth+Real are competing methods to GESPI**, as they do **not** provide any form of (distribution-free) risk control guarantees. These methods are shown only to illustrate the quality of the synthetic or pooled dataset, and as requested, we have already added Synth+Real results in the supporting document to all our experiments.
> > >
> > > Regarding "When the synthetic data has a very poor quality, OnlySynth is expected to perform poor; however, the method Synth+Real should still perform well.'' and the question ``what is the performance when the synthetic has a poor quality?''
> > >
> > > This is not quite right---for the following reason. **If the synthetic data are of low quality, pooling them with limited real data results in an overall low quality dataset, leading to poor performance**. This effect is clearly illustrated in our experiments in the supporting document (see [link](https://tinyurl.com/rebuttal-exp)); for example, in Figure S1, the synthetic data for the *Siberian Husky* class are of low quality, as the target coverage is 95\% but the observed coverage of OnlySynth and Synth+Real is only 60\%, demonstrating that Synth+Real also performs poorly in this case.
> > >
> > > The only scenario in which Synth+Real could perform well despite poor synthetic data is when $n\gg N$---that is, when there is abundant real data and only a small amount of synthetic data. In such a setting, the pooled distribution is effectively dominated by the real data. However, when the real dataset is already sufficiently large, **there is no reason to use synthetic data at all**. This is not the setting we consider in this paper, which focuses on scenarios where real data are limited.
> > >
> > > We thank the reviewer for their engagement and the opportunity to provide further clarification. We believe this discussion addresses the reviewer’s remaining concerns.

---

### Official Review · Reviewer_SedH · 2025-11-01

**Soundness:** 4
**Presentation:** 4
**Contribution:** 4
**Rating:** 8
**Confidence:** 2

**Summary:**

This work proposes General Synthetic-Powered Predictive Inference (GESPI) that can leverage synthetic data samples for statistical inference, while maintaining "guardrails" against synthetic data that differs (in quality, but probably also distribution) from the limited real data.

**Strengths:**

The proposed work is interesting and timely. Its approach is general and not tailored to any particular statistical inference model like prior works. The paper is well-written with a neat exposition that makes it easy to understand for even a general ML audience.

**Weaknesses:**

I do not have any concerns about this work. Some questions I had about the theory are below. Comments related to writing are under "questions."

**Q1. Effect of $n$** and $N$: How does the number of real samples $n$ and the number of synthetic samples $N$ (or their ratio) affect the guarantees in Theorems 3.2 and 3.3?

**Q2. Presence of low quality samples in real data**: If there were a few low quality samples in the real dataset and the synthetic data matched the distribution of good quality real samples, wouldn't GESPI essentially throw out some of the benefits from good quality synthetic data? I understand that this is not the exact scenario for which GESPI was designed, but I would like to know what its behavior would be.

**Q3. Comparison with SPI**: Is it possible to compare GESPI with SPI in conformal prediction experiments on the datasets in this work or the SPI paper?

**Questions:**

Comments related to writing:

**C1.** Line 284 says "the synthetic-leveraging procedure...improves upon the standard procedure..." Can the authors clarify what "improve" means here in formal terms?

---

> ### Author Response · Authors · 2025-11-20
> **Rebuttal by Authors**
>
> Thank you for your thoughtful review and encouraging feedback. We greatly appreciate your support. **If you believe that our paper has value, it would be great if you could support it in the discussion among the reviewers**. Below, we address your comments in detail.
> > **Q1**: Effect of $n$ and $N$
>
> Theorem 3.3 does not explicitly show the dependence on the sample sizes $n$ and $N$. Similarly, Theorem 3.2 establishes the validity of GESPI---i.e., that the algorithm controls the risk---without explicitly showing dependence on the sample sizes. These theorems are meant to validate the algorithm itself, not to quantify the effect of the sample sizes $n$ and $N$; the benefit of using synthetic data is illustrated empirically in our experiments.
>
> To theoretically capture such sample-size effects, additional assumptions on the base risk-controlling algorithm are required. Indeed, Theorem F.2, which extends Theorem 3.2 to the two-sided guardrail, explicitly depends on the sample sizes. Under the assumption that the base risk-controlling algorithm is near-tight and satisfies
> $$
> \alpha - \delta_n \leq \mathbb{E} [\ell(\text{Alg}_\alpha(\mathcal{D}_n,V)]\leq \alpha,
> $$
> where $\delta_n \rightarrow 0$ as $n\rightarrow \infty$ (for example, in conformal prediction $\delta_n=1/(n+1)$). This shows that the tightness of our guarantees depends on how large the sample size is.
>
> In Theorem F.2, the quantities $\delta_n$, $\delta_{N+n}$, and $\tau$ all converge to zero as both $n$ and $N$ increase. If $n$ is fixed and only $N \rightarrow \infty$, then $\delta_{N+n} \rightarrow 0$, whereas $\delta_n$ and $\tau$ do not.
> > **Q2**: Presence of low quality samples in real data
>
> We thank the reviewer for raising this interesting question and will add a discussion of it in the paper.
>
> If we understand the reviewer correctly, the question concerns the situation where applying the risk-controlling method on the real data at level $\alpha$ does not control the risk of the good-quality samples. For example, in conformal prediction, this would mean that the resulting prediction sets are too small and do not achieve coverage at the desired target level. This scenario is outside the scope of our setting, where the real data are considered the gold standard and we aim to control the risk with respect to this distribution.
>
> If GESPI is applied in this scenario using the two-sided guardrail, the risk would still not be controlled for the good-quality samples, because by design GESPI cannot construct sets larger than the base method---which is arguably the desired behavior in standard situations. In contrast, the one-sided version of GESPI would fully leverage the good synthetic data; for instance, in the conformal prediction example, the one-sided version of GESPI will output the synth.-powered prediction set, which benefits from the larger sample size of the synthetic data, and controls the risk at level $\alpha$ for the good-quality real samples (assuming the synthetic data match the good-quality real data distribution).
>
> As a remark, in some cases low-quality samples may lead to conservative risk control (e.g., label noise). For example, in conformal prediction, this results in prediction sets that are large and possibly conservative. In this scenario, both the two-sided and one-sided versions of GESPI could fully utilize the synthetic data if $\varepsilon$ is sufficiently large.
> > **Q3**: Comparison with SPI
>
> SPI is tailored to conformal prediction, and it is not clear how/if that method could be applied to the applications considered in this paper. Therefore, we cannot compare SPI with GESPI on those tasks. Nevertheless, focusing on the conformal prediction setting, GESPI is more sample-efficient than SPI, as it can leverage the pooled real and synthetic data (although demonstrating this is not the main focus of our work).
>
> To illustrate this, we conducted a new experiment in the conformal prediction setting. We follow the experimental setup from SPI for class-conditional coverage in image classification and compare GESPI and SPI as a function of the number of synthetic samples $N$. Following Figure S3 in the supporting document in this [link](https://tinyurl.com/rebuttal-exp), for small values of $N$, GESPI is more sample-efficient, achieving coverage closer to $1-\alpha$ and producing smaller prediction sets. While for sufficiently large $N$, the two methods become comparable.
> > **C1**: In line 284, can the authors clarify what "improve" means here in formal terms?
>
> Assuming the synthetic data are of high quality, by ``improve'' we mean that the effective sample size relative to the standard (base) procedure would be enhanced. In turn, this can potentially lead to (1) an empirical risk closer to the target level $\alpha$, resulting in less conservative risk control with potentially smaller variance and higher stability (Theorems 3.2 and F.2), and (2) higher statistical power (e.g., smaller prediction set size, etc.; Theorem 3.3).

---

> > ### Comment · Reviewer_SedH · 2025-11-26
> > **Response from reviewer SedH**
> >
> > Thank you for your response. My questions were adequately answered. I keep my score of 8.

---

### Author Response · Authors · 2025-11-20
**General Comment (Part 2)**

> ### **Comment 3** (Reviewer yUKm, FjoW): authors try to conceal the ``weakness'' of increasing the error by $\varepsilon$.

**Response**: To clarify, **there is nothing to conceal compared to the worst-case guardrail** (using the real data at level $\alpha + \varepsilon$); and there is no ``win or lose'' situation. GESPI is **not competing with the worst-case guardrail---which is instead an integral part of our method**. In fact, Theorem 3.3 formally shows that the GESPI procedure is deterministically sandwiched between the base and the worst-case guardrail. Deterministically, GESPI’s power is always capped by the worst-case guardrail, which is further discussed in the example presented after the theorem.

 We can certainly compare GESPI with the method that uses real data at error level $\alpha + \varepsilon$, and we include all experiments with this guardrail method in the supporting document in this [link](https://tinyurl.com/rebuttal-exp). However, that method **does not leverage the synthetic data, and therefore still suffers from the low real sample size, even if it is run at a higher level**. Our experiments show that for this reason, this method leads to a large variability (even though the average risk is controlled at $\alpha + \varepsilon$, the variance can be much higher than that of our GESPI method). This shows that this method is not uniformly preferable to our method.

To further illustrate this point, we added a new experiment on class-conditional coverage in image classification. As shown in Figure S1 in this [link](https://tinyurl.com/rebuttal-exp), for the *Magpie* class, the variance of the guardrail method is extremely large due to the small sample size, leading to realized coverage as low as 60\% even though the target level is 95\%. In contrast, GESPI achieves coverage close to 95\% with far lower variability, making it clearly preferable to the guardrail approach in this case.

**Clarification**: We regret not stating earlier in the manuscript that the power of GESPI is capped by the worst-case guardrail. We will ensure it is incorporated in the abstract, introduction, and contributions. As in the standard literature on statistical inference (e.g., hypothesis testing and confidence intervals), we viewed the validity of GESPI as its core underlying theoretical property. Thus, we emphasized that the worst-case guardrail ensures validity, and the base method ensures that the power of GESPI is never worse than the base method.

> ### **Comment 4** (Reviewers FjoW, MhsE, yUKm): lack of baselines.

**Response**: To the best of our knowledge, **there is no other method in the literature that controls risk under arbitrary synthetic data**. The natural ``baselines'' are `OnlyReal` and ` OnlySynth`, where the latter lacks any risk control guarantees and is included solely to illustrate how the unknown quality of the synthetic data can affect the performance (either controlling or not controlling the risk depending on the situation).

We did not include the worst-case guardrail in the initial manuscript because, as stated in Theorem 3.3, GESPI’s performance is deterministically sandwiched between the base method and the worst-case guardrail, and is in turn capped by the guardrail. Similarly, the pooled real and synthetic baseline was not included initially; like `OnlySynth`, this approach lacks risk control guarantees. These are not really baselines in the classic sense that they are competing with our method, but rather components of our method (our method uses both of them!), and one can think of this comparison more as an ablation study.

**Clarification**: To fully address the reviewers' requests, we include in the supporting document in this [link](https://tinyurl.com/rebuttal-exp) all possible baselines: `OnlyReal`---the base inference method applied to the real data at level $\alpha$; `Guardrail`---the same as OnlyReal but applied at level $\alpha+\varepsilon$; `OnlySynth`---the same inference method applied to the synthetic data at level $\alpha$; this method does not have error rate control guarantees; and `Synth+Real`---similar to OnlySynth but using the pooled real and synthetic data, which likewise lacks error-rate control guarantees.

---

> ### Comment · Reviewer_yUKm · 2025-11-20
>
> Can you post your link(s) without a URL shortener so the domain is clearly visible?

---

> > ### Author Response · Authors · 2025-11-20
> >
> > Here is the link to the supporting document with the full URL: https://anonymous.4open.science/api/repo/gespi-rebuttal-9E46/file/exp_rebuttal.pdf?v=2800fc21

---

### Author Response · Authors · 2025-11-20
**General Comment (Part 1)**

We thank the reviewers for their feedback and for taking the time to review our work. We address all comments separately in a point-by-point manner. But unfortunately, there are some critical misunderstandings that are at the core of our method, as well as factually incorrect comments that we would first like to clarify generally below.

> ### **Comment 1** (Reviewers FjoW, MhsE): minor novelty compared to previous SPI work [1].

[1] Bashari et al. (2025) ``Synthetic-Powered Predictive Inference.''

**Response**: Our work introduces two key contributions that go well beyond a reformulation of SPI as a wrapper. First, **we propose a formulation that is universally applicable to any loss function**, and thus allows us to tackle a broad variety of statistical problems in a unified way. This perspective is in line with one of the most commonly studied perspectives in statistical work, statistical decision theory, a field that formulates statistical problems in terms of loss functions, and which has been widely studied in the literature. Our method does enable using synthetic data with rigorous statistical guarantees in a wide variety of settings, which has not been previously possible.

Second, **we introduce the new GESPI procedure, which enables using synthetic data with risk-based safety guarantees, and which was not possible before our work**.  This method is not an extension of SPI, but rather a brand new perspective on how statistical inference can safely leverage synthetic data. GESPI is accompanied by new theoretical results, including theoretical results on the power of the method---results that do not exist for SPI. While we clearly give credit to SPI as an inspiration, the scope of GESPI is considerably broader (and the method is different!).

Importantly, SPI is tailored to conformal prediction, and after reading that paper, it was not clear at all how/if that method could be adapted to the broader setting we consider. The SPI method is based on opening windows in the space of scores of synthetic data and mapping the real scores to the synthetic scores. Clearly, the SPI window-based mapping approach is not applicable to the broader setting of loss control that we study here. Even in the conformal prediction setting, our method and theoretical guarantees differ from those in SPI (see Theorem H.1 compared with Theorems 3.3 and 3.5 in SPI).

Finally, **GESPI allows greater methodological flexibility than SPI**. In particular, we can utilize the synthetic data $\tilde{\mathcal{D}}_N$ pooled with the real data $\mathcal{D}_n$, and we can apply any algorithm that takes $(\tilde{\mathcal{D}}_N, \mathcal{D}_n)$ as inputs, while the guardrail bounds still hold; which was not previously possible. This flexibility allows the practitioners to filter, modify, or adapt the synthetic data using data-dependent choices without compromising validity. In contrast, SPI is restricted to applying only the split conformal prediction algorithm solely on the synthetic data.


> ### **Comment 2** (Reviewer FjoW): small-scale experiments with modest improvement compared to OnlyReal.

**Response**: **We included six different experimental settings covering a broad range of domains**: conformal risk control for protein structure prediction, hypothesis testing for large reasoning model comparisons, outlier detection with type I error and FWER control across three different benchmarks, hypothesis testing for mechanistic interpretability of vision transformers, and comprehensive simulated data experiments for hypothesis testing.

Thus, we believe that **the scale and diversity of our experiments match or exceed those in papers accepted to ICLR or related venues in past years**. In response to the reviewers’ comments (which were not related to experimental scale), we added two further experimental settings (see all experiments are in the supporting document in this [link](https://tinyurl.com/rebuttal-exp)): class-conditional coverage for image classification on ImageNet with FLUX-generated images, and hypothesis testing for hyper-parameter selection for the K-majority vote of a large reasoning model.

Further, **our experiments strongly support the claim that the improvements can be significant**, especially when synthetic data is moderate to high quality. For this reason, we disagree with the claim that the improvements are modest. In particular, the gains in the outlier-detection experiments and the protein-folding experiments are substantial (see Figures 5 and 3). Specifically, for the outlier detection, we have the same power as the oracle procedure! To summarize, we believe that our experiments convincingly demonstrate the benefits of using our method, which allows practitioners to leverage synthetic data in a safe way.

---

### Note · Authors · 2025-12-01

**Comment:**

We thank the reviewers for their input and engagement.
The main point raised in the rebuttal phase was the importance of emphasizing the additional variance-reduction benefit of GESPI when the synthetic data are of high quality. We agree that it is important to further highlight this aspect. GESPI achieves risk closer to the target level $\alpha$ and improves sample efficiency in this setting. This enhanced sample efficiency can not only lead to higher power (while still achieving risk at level $\alpha$), but also reduce the variance of the empirical risk. This provides an additional and practically important benefit in applications such as conformal prediction, conformal risk control, and outlier detection (both single and multiple testing).

Unfortunately, although we believe the discussion was heading in a positive direction, we are unable to continue the discussion with the reviewers and therefore withdraw our submission.

**Withdrawal Confirmation:**

I have read and agree with the venue's withdrawal policy on behalf of myself and my co-authors.